# On Minimax Estimation of Parameters in Softmax-Contaminated Mixture of Experts

**Fanqi Yan**[*,1]    **Huy Nguyen**[*,2]    **Dung Le**[*,2]    **Pedram Akbarian**[3]
**Nhat Ho**[2]    **Alessandro Rinaldo**[2]

[1] Department of Computer Science,  [2] Department of Statistics and Data Sciences,
[3] Department of Electrical and Computer Engineering,   The University of Texas at Austin

`{fanqi.yan, huynm, quangdung0110, akbarian, minhnhat}@utexas.edu`,
`alessandro.rinaldo@austin.utexas.edu`

## Abstract

The softmax-contaminated mixture of experts (MoE) model is deployed when a large-scale pre-trained model, which plays the role of a fixed expert, is fine-tuned for learning downstream tasks by including a new contamination part, or prompt, functioning as a new, trainable expert. Despite its popularity and relevance, the theoretical properties of the softmax-contaminated MoE have remained unexplored in the literature. In the paper, we study the convergence rates of the maximum likelihood estimator of gating and prompt parameters in order to gain insights into the statistical properties and potential challenges of fine-tuning with a new prompt. We find that the estimability of these parameters is compromised when the prompt acquires overlapping knowledge with the pre-trained model, in the sense that we make precise by formulating a novel analytic notion of distinguishability. Under distinguishability of the pre-trained and prompt models, we derive minimax optimal estimation rates for all the gating and prompt parameters. By contrast, when the distinguishability condition is violated, these estimation rates become significantly slower due to their dependence on the prompt convergence rate to the pre-trained model. Finally, we empirically corroborate our theoretical findings through several numerical experiments.

## 1 Introduction

Mixture of experts (MoE) [14, 16] has emerged as a statistical machine learning model that aggregates the power of multiple sub-models. This model consists of two primary components: expert function (or, simply, expert) and a gating network. Experts can be, for example, a feed-forward network (FFN) [33, 4], a classifier [2, 27], or a regression model [7, 17]. The gating network softly divides the input space into multiple regions where the opinions of some experts are deemed to be more trustworthy than others. This is done by dynamically allocating higher input-dependent weights instead of constant weights to the various experts, making MoE more flexible and adaptive than traditional mixture models [25]. As a consequence, MoE has been leveraged in a wide range of fields, including natural language processing [5, 15, 10, 8, 21, 33], computer vision [32, 24], speech recognition [36, 37], multimodal learning [11, 38, 28], continual learning [20, 22], and reinforcement learning [1, 3].

Unlike these applications where all experts are trainable, parameter-efficient fine-tuning methods such as prefix tuning [23, 19, 18] can be interpreted as a mixture of a frozen or pre-trained expert and

---

[*]Co-first authors.

39th Conference on Neural Information Processing Systems (NeurIPS 2025).

a trainable prompt expert responsible for learning downstream or more specialized tasks, which we refer to as *contaminated MoE* throughout this paper. Despite the empirical success of this fine-tuning approach, there is a very limited theoretical understanding of their properties and limitations in the literature. To the best of our knowledge, contaminated MoE has only been previously studied in [35] to characterize expert structures achieving the optimal parameter estimation rates. However, the analysis in that work is conducted under a simplified setting where the gating (mixture weight) is independent of the input value, which is a very impractical assumption. To close this gap, we undertake a thorough theoretical analysis of the more commonly used *softmax-contaminated MoE* model, specified in equation (1) below, a contaminated MoE model whose gating function takes the form of a soft-maxed linear network. We analyze the issue of identifiability and the convergence properties of the maximum likelihood estimator of the prompt parameters to shed light on the understanding of prompt behavior in prefix tuning methods. A main take-away of our analysis is the potential for the prompt to be exceedingly similar to – and thus to acquire the same knowledge as – the pre-trained model, a situation greatly impacting the estimability of the prompt parameter. To overcome this issue, in Definition 1 we formulate analytical properties of the pre-trained and prompt models, which we refer to as *distinguishability,* that are guaranteed to rule out excessive overlap between the models and ensure good estimation rates. We make the following contributions.

*(i) Distinguishability of the prompt model from the pre-trained model.* In Section 2, we propose a novel notion of distinguishability between the pre-trained and prompt models and then illustrate its properties.

*(ii) When the distinguishability condition is satisfied,* we show in Section 3.1 that the prompt does not converge to the pre-trained model – intuitively, these two models have distinct expertise. In fact, we demonstrate that the convergence rates of the MLE of all the prompt and gating parameters are of parametric order in the sample size $n$, that is, $\widetilde{\mathcal{O}}(n^{-1/2})$. Furthermore, we establish minimax lower bounds on the estimation errors with matching rates, thus showing that the convergence rate of MLE is minimax optimal.

*(iii) When the distinguishability condition is violated,* the prompt will converge to the pre-trained model, that is, both models employ the same expert structure and thus will gain similar expertise. In Section 3.2, we show that, under this setting, the estimation rates for prompt and gating parameters are negatively affected by the prompt convergence to the pre-trained model and, therefore, become substantially slower than the parametric rate $\widetilde{\mathcal{O}}(n^{-1/2})$. We confirm that these slower rates are tight by deriving matching minimax lower bounds. See Table 1 for a summary of our results.

Lastly, in Section 4, we carry out several numerical experiments to empirically justify our theoretical results, and then conclude the paper in Section 5. Rigorous proofs are provided in the Appendices.

A major technical innovation in our contribution that sets it apart from existing theoretical analyses of MoE models is the fact that we let the parameters of the prompt model to vary with the sample size $n$, thus potentially allowing for a more challenging estimation task as the sample size increases. This approach is necessary to carry out a minimax analysis.

**Notation.** For any $n \in \mathbb{N}$, we let $[n] := \{1, 2, \ldots, n\}$. For a vector $u$ we denote with $\|u\|$ its Euclidean norm value. Given any two positive sequences $(a_n)_{n \geq 1}$ and $(b_n)_{n \geq 1}$, we write $a_n = \mathcal{O}(b_n)$ or $a_n \lesssim b_n$ if $a_n \leq Cb_n$ for all $n \in \mathbb{N}$ and some $C > 0$. We further write $a_n = \widetilde{\mathcal{O}}(b_n)$ to denote $a_n \lesssim b_n \mathrm{polylog}(b_n)$, where $\mathrm{polylog}(b_n)$ indicate any term that is polylogarithmic in $b_n$. Lastly, for any two densities $p$ and $q$ (dominated by the Lebesgue measure), their squared Hellinger distance is computed as $d_H^2(p, q) := \frac{1}{2} \int [\sqrt{p(x)} - \sqrt{q(x)}]^2 dx$, while the total variation distance is given by $d_V(p, q) := \frac{1}{2} \int |p(x) - q(x)| dx$.

Table 1: Summary of parameter estimation rates in the softmax-contaminated MoE model. Notice that the rates are in expectation. For the notation, please refer to equations (1) and (2). In addition, we also denote $\Delta \eta^* := \eta^* - \eta_0$ and $\Delta \nu^* := \nu^* - \nu_0$.

| **Setting** | $\mathbf{\lvert \exp(\widehat{\tau}_n) - \exp(\tau^*) \rvert}$ | $\mathbf{\lVert \widehat{\beta}_n - \beta^* \rVert}$ | $\mathbf{\lVert \widehat{\eta}_n - \eta^* \rVert}$ | $\mathbf{\lvert \widehat{\nu}_n - \nu^* \rvert}$ |
|---|---|---|---|---|
| Distinguishable | $\widetilde{\mathcal{O}}(n^{-\frac{1}{2}})$ | $\widetilde{\mathcal{O}}(n^{-\frac{1}{2}})$ | | |
| Non-distinguishable | $\widetilde{\mathcal{O}}(n^{-\frac{1}{2}} \cdot \lVert (\Delta\eta^*, \Delta\nu^*) \rVert^{-2})$ | $\widetilde{\mathcal{O}}(n^{-\frac{1}{2}} \cdot \lVert (\Delta\eta^*, \Delta\nu^*) \rVert^{-1})$ | | |

## 2 Preliminaries

In this section, we begin with setting up the problem, followed by a discussion on related works in Section 2.1. Then, in Section 2.2, we introduce the distinguishability condition and provide an investigation into the fundamental properties of the softmax-contaminated MoE, including the model identifiability and the model convergence.

### 2.1 Problem Setup

**Problem setting.** Suppose that $(X_1, Y_1), (X_2, Y_2), \ldots, (X_n, Y_n) \in \mathcal{X} \times \mathcal{Y} \subset \mathbb{R}^d \times \mathbb{R}$ are i.i.d. samples of covariate-response pairs of size $n$. We assume that the input covariates $X_1, X_2, \ldots, X_n$ are drown in an i.i.d. manner from some known continuous probability distribution on $\mathbb{R}^d$ and that the responses are generated according to a softmax-contaminated MoE model, which postulates that the conditional density function of the response given the covariates is given by

$$p_{G_*}(y|x) := \frac{1}{1 + \exp((\beta^*)^\top x + \tau^*)} \cdot f_0(y|h_0(x, \eta_0), \nu_0)$$
$$+ \frac{\exp((\beta^*)^\top x + \tau^*)}{1 + \exp((\beta^*)^\top x + \tau^*)} \cdot f(y|h(x, \eta^*), \nu^*). \tag{1}$$

Above, the pre-trained model corresponds to as a *fixed* and known conditional probability density function $f_0(\cdot|h_0(\cdot, \eta_0), \nu_0)$, parametrized by the pre-trained mean expert function $x \mapsto h_0(x, \eta_0)$ and variance $\nu_0$. Meanwhile, the prompt model, denoted as $f(\cdot|h(\cdot, \eta^*), \nu^*)$ is modeled as an unknown Gaussian density function with the prompt mean expert $x \mapsto h(x, \eta^*)$ and variance $\nu^*$. We collect all the unknown parameters of the prompt model into the vector $G_* = (\beta^*, \tau^*, \eta^*, \nu^*)$, belonging to some *parameter space* $\Xi \subseteq \mathbb{R}^d \times \mathbb{R} \times \mathbb{R}^q \times \mathbb{R}_+$. Note that we allow the values of these parameters to vary with the sample size $n$. However, for notational convenience, we suppress the dependence of $G_*$ on $n$ throughout the paper. In addition, it should also be noted that the "probabilistic" MoE model (1) can be related to "deterministic" MoE models used in deep learning [33] by taking the expectation of the response given the covariate, that is,

$$\mathbb{E}[Y|X] = \frac{1}{1 + \exp((\beta^*)^\top x + \tau^*)} \cdot h_0(x, \eta_0) + \frac{\exp((\beta^*)^\top x + \tau^*)}{1 + \exp((\beta^*)^\top x + \tau^*)} \cdot h(x, \eta^*).$$

**Maximum likelihood estimation (MLE).** We utilize the maximum likelihood method [34] to estimate the unknown parameters $G_* = (\beta^*, \tau^*, \eta^*, \nu^*)$ of the softmax-contaminated MoE model (1) as follows:

$$\widehat{G}_n := (\widehat{\beta}_n, \widehat{\tau}_n, \widehat{\eta}_n, \widehat{\nu}_n) \in \arg\max_{G \in \Xi} \sum_{i=1}^n \log(p_G(Y_i|X_i)). \tag{2}$$

For the sake of theory, we assume that the input space $\mathcal{X}$ is bounded, whereas the parameter space $\Xi$ is compact. In addition, we assume that the prompt expert function $x \mapsto h(x, \eta)$ is differentiable with respect to $\eta \in \mathbb{R}^q$ for almost all $x \in \mathcal{X}$. Note that these assumptions are mild and have been used in previous works [13, 30, 35].

**Related work.** Mendes et al. [26] considered an MoE model where each expert was formulated as a polynomial regression model. Their objective was to address the trade-off between the number of experts and the expert size to obtain the optimal parameter estimation rates. Next, Ho et al. [13] took into account the parameter estimation problem for Gaussian MoE models with input-free gating. They demonstrated that when expert functions satisfied an algebraic independence condition, the convergence rates of MLE were optimal of parametric order on the sample size. Conversely, if the expert functions are not algebraic independent, then the parameter estimation rates became inversely proportional to the number of fitted experts. These results were then extended to more practical settings of input-dependent gatings, including softmax gating [31] and sigmoid gating [29], revealing that the latter was more sample-efficient than former in terms of expert estimation.

It was not until 2024 that Nguyen et al. [30] investigated a contaminated MoE where a frozen pre-trained model was fine-tuned by a mixture of prompts rather than a single prompt model. However, they imposed two unrealistic assumptions on their model of interest: they equipped the contaminated MoE with input-free gating and kept the ground-truth parameters unchanged with the sample size.

Then, Yan et al. [35] overcame the second limitation by allowing ground-truth parameters to hinge on the sample size as in the case of traditional mixture models [6], while the first limitation remained unsolved. Therefore, in this work, our goal is to completely address both limitations by studying the softmax-contaminated MoE in equation (1).

**Challenges.** There are three fundamental challenges of our analysis compared to previous work.

*1. Uniform convergence rates.* We allow ground-truth parameters $G_*$ to change with sample size $n$, which is challenging yet closer to practice than the settings in previous works on MoE [31, 29], where $G_*$ does not change with $n$. Thus, the convergence rates of parameter estimations in our work are uniform rather than point-wise as in those works.

*2. Minimax lower bounds.* We determine minimax lower bounds under both distinguishable and non-distinguishable settings. Based on these lower bounds, we can claim that our derived convergence rates are optimal. However, no minimax lower bounds are provided in [31, 29].

*3. Input-dependent gating.* The latest work on understanding the contaminated MoE model is [35], but it considers input-free gating in the analysis. On the other hand, in this paper, we take into account softmax gating, which hinges upon the input value. This input-dependence yields several challenges on the convergence of density estimation and parameter estimation.

## 2.2 Fundamental Properties of the Softmax-Contaminated MoE

As mentioned above, when the prompt's learned skills overlap with those of the pre-trained model, estimating the prompt parameters becomes challenging due to potential non-identifiability. To capture that issue accurately, we introduce an analytic condition called distinguishability in Definition 1.

**Definition 1** (Distinguishability). *We say that $f_0$ is distinguishable from $f$ if the following hold: for any distinct pairs of parameters $(\eta_1, \nu_1), (\eta_2, \nu_2) \in \Theta$, if there exist measurable real-valued functions $x \in \mathcal{X} \mapsto b_0(x)$, $x \in \mathcal{X} \mapsto b_1(x)$, and $x \in \mathcal{X} \mapsto \{c_\alpha(x)\}_{0 \leq |\alpha| \leq 1}$, where $\alpha = (\alpha_1, \alpha_2) \in \mathbb{N}^q \times \mathbb{N}$ with $|\alpha| = |\alpha_1| + \alpha_2 \leq 1$ such that*

$$b_0(x) \cdot f_0(y|h_0(x, \eta_0), \nu_0) + b_1(x) \cdot f(y|h(x, \eta_1), \nu_1)$$
$$+ \sum_{0 \leq |\alpha| \leq 1} c_\alpha(x) \cdot \frac{\partial^{|\alpha|} f}{\partial \eta^{\alpha_1} \partial \nu^{\alpha_2}}(y|h(x, \eta_2), \nu_2) = 0,$$

*for almost every $(x, y) \in \mathcal{X} \times \mathcal{Y}$, then it must be the case that*

$$b_0(x) = b_1(x) = 0, \quad c_\alpha(x) = 0 \quad \text{for all } 0 \leq |\alpha| \leq 1, \quad \text{for almost every } x.$$

To help understand the notion of distinguishability better, in our next result we characterize the class of pre-trained models distinguishable from the prompt $f$. The proof can be found in Appendix B.1.

**Proposition 1.** *If a pre-trained model $f_0$ does not belong to the family of Gaussian densities, then $f_0$ is distinguishable from the prompt model $f$ in the sense of Definition 1.*

On the other hand, if $f_0$ belongs to the family of Gaussian distributions and the pre-trained expert shares the same structure as the prompt expert, that is, $h_0 = h$, then the above condition is violated. It should be noted that the distinguishability condition ensures that the prompt does not acquire overlapping knowledge with the pre-trained model since the equation $f_0(y|h(x, \eta_0), \nu_0) = f(y|h(x, \eta), \nu)$ cannot hold for almost all $(x, y) \in \mathcal{X} \times \mathcal{Y}$. Moreover, we illustrate in the following proposition that the distinguishability condition also implies that the softmax-contaminated MoE is identifiable.

**Proposition 2** (Identifiability). *Let $G, G'$ be two components in $\Xi$. Suppose that $f$ is distinguishable from $f_0$, then if the identifiability equation $p_G(y|x) = p_{G'}(y|x)$ holds for almost all $(x, y) \in \mathcal{X} \times \mathcal{Y}$, then we obtain $G = G'$.*

The proof of Proposition 2 is provided in Appendix B.2. Given the consistency of the softmax-contaminated MoE, we continue to investigate the convergence behavior of density estimation under this model in Proposition 3 whose proof can be found in Appendix B.3. We conclude this section with a consistency guarantee for the contaminate density itself, which under mild tail conditions on $f_0$, can be estimated at a parametric rate in the Hellinger distance, regardless of the distinguishability between $f_0$ and $f$. Below and throughout the paper, $\mathbb{E}_{p_{G_*, n}}$ denotes the expectation operator with respect

to the joint distribution of the data $(X_1, Y_1), \ldots, (X_n, Y_n)$ and assuming the softmax-contaminated MoE model (1) parametrized by $G^* \in \Xi$, i.e. $Y_i | X_i \sim p_{G_*}$ for all $i$. Instead, $\mathbb{E}_X$ indicates the expectation with respect to the input distribution.

**Proposition 3** (Model Convergence). *Suppose that the pre-trained model $f_0$ is bounded and, for some $p > 0$,*

$$\mathbb{E}_X \left[ -\log f_0(y | h_0(X, \eta_0), \nu_0) \right] \gtrsim y^p, \quad \text{for almost every } y \in \mathcal{Y}. \tag{3}$$

*Then, for the MLE $\widehat{G}_n$ defined in equation (2), it holds, for almost all $x \in \mathcal{X}$,*

$$\sup_{G_* \in \Xi} \mathbb{E}_{p_{G_*, n}} \left[ \mathbb{E}_X \left[ d_H \left( p_{\widehat{G}_n}(\cdot | X), p_{G_*}(\cdot | X) \right) \right] \right] \lesssim \sqrt{\log(n)/n}. \tag{4}$$

The above result shows that the density estimator $p_{\widehat{G}_n}$ converges to the true density $p_{G_*}$ under the Hellinger distance at the near-parametric rate of order $\widetilde{\mathcal{O}}(n^{-1/2})$. To extract from this result a convergence guarantee for the MLE $\widehat{G}_n$ itself, we follow a by-now-standard approach in the latest analysis of MoEs; see, e.g., [31]. The main idea is that, if one can exhibit a loss function among parameters, say $D(\widehat{G}_n, G_*)$, such that $\mathbb{E}_{p_{G_*, n}}[D(\widehat{G}_n, G_*)] \lesssim \mathbb{E}_{p_{G_*, n}} \left[ \mathbb{E}_X \left[ d_H \left( p_{\widehat{G}_n}(\cdot | X), p_{G_*}(\cdot | X) \right) \right] \right]$, then convergence of $\widehat{G}_n$ in the expected $D(\cdot, \cdot)$ loss, as well potentially information on the rate of convergence, will follow. See Appendix A for further details. Throughout the rest of the paper, we assume that the tail condition (3) on $f_0$ and the distribution of $X$ used in Proposition 3 is in effect.

## 3 Convergence Analysis of Parameter Estimation

In this section, we present various convergence rates for the MLE estimator of the model prompt and gating parameters. In Sections 3.1 and 3.2 we provide separate minimax analyses, depending on whether the distinguishability condition of Definition 1 holds or not, respectively.

### 3.1 Distinguishable Setting

To start with, we consider a scenario in which the pre-trained model $f_0$ is distinguishable from the prompt model $f$. Recall that given the density estimation rate in Proposition 3, we need to construct a loss function between the MLE $\widehat{G}_n$ and the ground-truth parameters $G_*$, which should be bounded by the Hellinger distance between the two corresponding densities, in order to capture the parameter estimation rates. Tailored to the distinguishable setting, we measure the discrepancy between two arbitrary parameters $G$ and $G_*$ in $\Xi$ via the loss

$$D_1(G, G_*) = |\exp(\tau) - \exp(\tau^*)| + \left( \exp(\tau) + \exp(\tau^*) \right) \| (\beta, \eta, \nu) - (\beta^*, \eta^*, \nu^*) \|. \tag{5}$$

We are ready to determine the convergence behavior of the MLE under distinguishable settings.

**Theorem 1.** *Suppose that the pre-trained model $f_0$ is distinguishable from the prompt model $f$. For almost every $x \in \mathcal{X}$, and for any $\eta \in \mathbb{R}^q$, we assume that the Jacobian of the prompt expert function does not vanish, i.e., $\frac{\partial h}{\partial \eta}(x, \eta) \neq 0$. Then, there exists a positive constant $C_1$ that depends on $\Xi$ and $f_0$ such that the Hellinger lower bound $\mathbb{E}_X [d_H(p_G(\cdot | X), p_{G_*}(\cdot | X))] \geq C_1 D_1(G, G_*)$ holds for all parameters $G \in \Xi$. As a result, we obtain*

$$\sup_{G_* \in \Xi} \mathbb{E}_{p_{G_*, n}} \left[ |\exp(\widehat{\tau}_n) - \exp(\tau^*)|^2 \right] \lesssim \log(n)/n, \tag{6}$$

$$\sup_{G_* \in \Xi} \mathbb{E}_{p_{G_*, n}} \left[ \exp^2(\tau^*) \| (\widehat{\beta}_n, \widehat{\eta}_n, \widehat{\nu}_n) - (\beta^*, \eta^*, \nu^*) \|^2 \right] \lesssim \log(n)/n. \tag{7}$$

The proof of Theorem 1 is deferred to Appendix A.1. The bound in equation (6) reveals that the gating parameter estimator $\exp(\widehat{\tau}_n)$ converges to its ground-truth counterpart $\exp(\tau^*)$ at a rate of order $\widetilde{\mathcal{O}}(n^{-1/2})$. Analogously, looking at the bound in equation (7), since the terms $\exp(\tau^*)$ cannot go to zero due to the compactness of the parameter space $\Xi$, it follows that the convergence rates of the parameter estimators $\widehat{\beta}_n, \widehat{\eta}_n$, and $\widehat{\nu}_n$ to $\beta^*, \eta^*$ and $\nu^*$ are also of order $\widetilde{\mathcal{O}}(n^{-1/2})$. Meanwhile, in the contaminated MoE with input-free gating in [35], the estimation rates for prompt parameters

$\eta^*, \nu^*$ are slower than $\widetilde{\mathcal{O}}(n^{-1/2})$ as they depend on the convergence rate of the gating parameter to zero. Therefore, replacing the input-free gating with the softmax gating in the contaminated MoE helps reduce the sample complexity of parameter estimation.

Given the near-parametric convergence rates in Theorem 1, it is natural to wonder if they are optimal. To answer this question in the affirmative, below we derive minimax lower bounds.

**Theorem 2.** *If the pre-trained model $f_0$ is distinguishable from the prompt model $f$, then the following minimax lower bounds hold for any $0 < r < 1$:*

$$\inf_{\overline{G}_n \in \Xi} \sup_{G \in \Xi} \mathbb{E}_{p_{G,n}} \left( |\exp(\overline{\tau}_n) - \exp(\tau)|^2 \right) \gtrsim n^{-1/r},$$

$$\inf_{\overline{G}_n \in \Xi} \sup_{G \in \Xi} \mathbb{E}_{p_{G,n}} \left( \exp^2(\tau) \|(\overline{\beta}_n, \overline{\eta}_n, \overline{\nu}_n) - (\beta, \eta, \nu)\|^2 \right) \gtrsim n^{-1/r},$$

*where the infimum is over all estimators $\overline{G}_n := (\overline{\beta}_n, \overline{\tau}_n, \overline{\eta}_n, \overline{\nu}_n)$ taking values in $\Xi$.*

The proof of Theorem 2 can be found in Appendix A.2. The above minimax lower bounds imply that, under distinguishability, the convergence rates of the MLE, of order $\widetilde{\mathcal{O}}(n^{-1/2})$ is nearly minimax optimal, save for a logarithmic factor.

## 3.2 Non-distinguishable Setting

We now turn to the much subtler case in which the distinguishability condition is violated. Since we assume a Gaussian prompt, it follows from Proposition 2 that the pre-trained model $f_0$ necessarily belongs to the family of Gaussian densities. Furthermore, if the pre-trained and prompt model use the same expert function, i.e. $h_0 = h$, then $f_0$ is not distinguishable from the prompt model $f$. We will thus focus on this challenging scenario.

Under this setting, the prompt model may converge to the pre-trained model. In particular, if the pair of prompt parameters $(\eta^*, \nu^*)$ converge to the pair of pre-trained parameters $(\eta_0, \nu_0)$ as $n \to \infty$, then it follows that $f(\cdot|h(\cdot, \eta^*), \nu^*)$ converges to $f_0(\cdot|h(\cdot, \eta_0), \nu_0)$, indicating that the prompt learns the same expertise as the pre-trained model. Therefore, it becomes difficult for the gating network to assign higher weight to either the pre-trained model or the prompt than the other as they have similar expertise. As a result, one may expect the estimation rates of the gating parameters to be substantially slower. To formalize these setttings precisely, we need to pay more attention to the expert structure.

It should be noted that a key step in obtaining the MLE convergence rates in Theorem 1 is to decompose the density discrepancy $p_{\widehat{G}_n} - p_{G_*}$ into a combination of linearly independent terms through an appropriate Taylor series expansion of the function $g(y|x; \beta, \eta, \nu) := \exp(\beta^\top x) \cdot f(y|h(x, \eta), \nu)$ with respect to its parameters $\beta, \eta, \nu$. This process involves, in particular, higher derivatives of the expert function $h$ with respect to $\eta$, which may not be algebraically independent. To ensure the linear independence of the terms in the Taylor expansion, we formulate a *strong identifiability* condition that is indeed sufficient for these purposes.

**Definition 2** (Strong Identifiability)**.** *The expert function $x \mapsto h(x, \eta)$ is strongly identifiable if it is twice differentiable with respect to $\eta \in \mathbb{R}^q$ for almost all $x \in \mathcal{X}$, and if, for any fixed $\beta \in \mathbb{R}^d$ and $\eta \in \mathbb{R}^q$, each of the following sets of real-valued functions (of $x$) consists of linearly independent functions over $\mathbb{R}$. For notational simplicity, we write $h(\cdot)$ in place of $h(\cdot, \eta)$ below.*

1. *The first-order gating independence set:*

$$\left\{ \frac{\partial h}{\partial \eta^{(u)}}, \ \exp(\beta^\top x) \frac{\partial h}{\partial \eta^{(u)}} \right\}_{u \in [q]}.$$

2. *The gradient product independence set:*

$$\left\{ 1, \ x^{(w)}, \ \exp(\beta^\top x), \ \frac{\partial h}{\partial \eta^{(u)}} \frac{\partial h}{\partial \eta^{(v)}}, \ \exp(\beta^\top x) \frac{\partial h}{\partial \eta^{(u)}} \frac{\partial h}{\partial \eta^{(v)}} \right\}_{u,v \in [q], \ w \in [d]}.$$

3. *The mixed and second-order independence set:*

$$\left\{ \frac{\partial h}{\partial \eta^{(u)}}, \ \exp(\beta^\top x) \frac{\partial h}{\partial \eta^{(u)}}, \ x^{(w)} \frac{\partial h}{\partial \eta^{(u)}}, \ \frac{\partial^2 h}{\partial \eta^{(u)} \partial \eta^{(v)}}, \ \exp(\beta^\top x) \frac{\partial^2 h}{\partial \eta^{(u)} \partial \eta^{(v)}} \right\}_{u,v \in [q], \ w \in [d]}.$$

Here, the *First-order gating independence* condition guarantees that changes in $h$ with respect to $\eta$ remain distinguishable, even after modulation by the gating weights $\exp(\beta^\top X)$. This is a minimal requirement to ensure that the expert and gating mechanisms interact in a structurally non-degenerate way. The *Gradient product independence* condition guarantees that the products of directional derivatives of $h$ are distinguishable from each other (even under modulation by gating terms) and cannot be expressed as a linear combination of basic functions. This prevents higher-order interactions among gradients from collapsing into lower-order structures. Finally, the *Mixed and second-order independence* condition is stronger than the first-order one. It rules out first-order interactions between expert and gating parameters of the form $\partial h/\partial \eta^{(w)} = x^{(w)} \cdot \partial h/\partial \eta^{(v)}$, which would imply $\partial g/\partial \eta^{(w)} = \partial^2 g/(\partial \beta^{(w)} \partial \eta^{(v)})$. It also requires that second-order derivatives remain linearly independent, even accounting for the effect of the gating function. This guarantees that both first- and second-order directional changes in $h$ convey distinct, non-redundant information, and that higher-order structure in $h$ cannot be reduced to or absorbed by lower-order terms. This is essential when handling second-order Taylor expansions of the model.

**Examples.** The expert functions $h(x,\eta) = \mathrm{GELU}(\eta^\top x)$, $h(x,\eta) = \mathrm{sigmoid}(\eta^\top x)$, and $h(x,\eta) = \tanh(\eta^\top x)$ satisfy the strong identifiability condition, as their nonlinearities avoid degeneracies. In contrast, $h(x,\eta) = \mathrm{ReLU}(\eta^\top x)$ fails the second-order independence condition, as the second-order derivatives vanish almost everywhere. Another failure case arises when $h(x,\eta) = \sigma(a^\top x + b)$, where $\eta = (a, b)$ and $\sigma$ is any scalar activation function. This leads to $\partial h/\partial a = x \cdot \partial h/\partial b$, directly violating Condition 3.

To determine the convergence rates for the MLE in these settings, we construct the following loss function between parameters $G$ and $G^*$, carefully tailored to the non-distinguishable setting:

$$
\begin{aligned}
D_2(G, G_*) : = & \exp(\tau)\|(\Delta\eta, \Delta\nu)\|^2 + \exp(\tau^*)\|(\Delta\eta^*, \Delta\nu^*)\|^2 \\
& - \min\{\exp(\tau), \exp(\tau^*)\} \left(\|(\Delta\eta, \Delta\nu)\|^2 + \|(\Delta\eta^*, \Delta\nu^*)\|^2\right) \\
& + \left(\exp(\tau)\|(\Delta\eta, \Delta\nu)\| + \exp(\tau^*)\|(\Delta\eta^*, \Delta\nu^*)\|\right) \times \|(\beta, \eta, \nu) - (\beta^*, \eta^*, \nu^*)\|,
\end{aligned}
$$

where we denote $(\Delta\eta, \Delta\nu) = (\eta - \eta_0, \nu - \nu_0)$ and $(\Delta\eta^*, \Delta\nu^*) = (\eta^* - \eta_0, \nu^* - \nu_0)$.

**Theorem 3.** *Suppose that $f_0$ belongs to the family of Gaussian densities and $h_0 = h$. Then, there exists a positive constant $C_2$ that depends on $\Xi, \eta_0, \nu_0$ such that $\mathbb{E}_X \left[d_H(p_G(\cdot|X), p_{G_*}(\cdot|X))\right] \geq C_2 D_2(G, G_*)$ holds for all parameters $G$. As a result, we obtain*

$$
\sup_{G_* \in \Xi(l_n)} \mathbb{E}_{p_{G_*,n}}\left[\|(\Delta\eta^*, \Delta\nu^*)\|^4 \times |\exp(\widehat{\tau}_n) - \exp(\tau^*)|^2\right] \lesssim \log(n)/n, \tag{8}
$$

$$
\sup_{G_* \in \Xi(l_n)} \mathbb{E}_{p_{G_*,n}}\left[\exp^2(\tau^*)\|(\Delta\eta^*, \Delta\nu^*)\|^2 \times \|(\widehat{\beta}_n, \widehat{\eta}_n, \widehat{\nu}_n) - (\beta^*, \eta^*, \nu^*)\|^2\right] \lesssim \log(n)/n, \tag{9}
$$

*for any sequence $(l_n)_{n \geq 1}$ such that $l_n/\log n \to \infty$ as $n \to \infty$ where we denote*

$$
\Xi(l_n) := \left\{G = (\tau, \beta, \eta, \nu) \in \Xi : \frac{l_n}{\min\limits_{1 \leq i \leq q, 1 \leq j \leq d,} \left\{|\eta^{(i)}|^2, |\nu|^2, |\beta^{(j)}|^2\right\} \sqrt{n}} \leq \exp(\tau)\right\}.
$$

The proof of Theorem 3 is in Appendix A.3. Note that under the setting of Theorem 3, the softmax-contaminated MoE model is not identifiable, that is, the equation $p_G(y|x) = p_{G_*}(y|x)$ for almost all $(x, y)$ does not imply $G = G_*$. For that reason, we restrict the parameter space to the set $\Xi(l_n)$ to guarantee the consistency of the MLE. Compared to Theorem 1, the above rates exhibit differ in several aspects.

*(i)* From equation (8), we observe that the convergence rate of $\exp(\widehat{\tau}_n)$ to $\exp(\tau^*)$ becomes slower than the parametric order $\widetilde{\mathcal{O}}(n^{-1/2})$ as they depend on the vanishing rate of $(\Delta\eta^*, \Delta\nu^*)$ to zero. For example, if the pair of prompt parameters $(\eta^*, \nu^*)$ approach $(\eta_0, \nu_0)$ at the rate of $\widetilde{\mathcal{O}}(n^{-1/8})$, then the bound (8) implies that $\exp(\widehat{\tau}_n)$ goes to $\exp(\tau^*)$ at the rate of $\widetilde{\mathcal{O}}(n^{-1/4})$. This toy example is indeed confirmed by our numerical experiments in the next section.

*(ii)* Likewise, the convergence rates of the estimators $(\widehat{\beta}_n, \widehat{\eta}_n, \widehat{\nu}_n)$ are also impacted by the convergence rates of the prompt parameters and therefore slower than $\widetilde{\mathcal{O}}(n^{-1/2})$. For example, if

$(\Delta\eta^*, \Delta\nu^*)$ go to zero at the rate of $\widetilde{\mathcal{O}}(n^{-1/8})$, then the bound (9) indicates that $\widehat{\beta}_n, \widehat{\eta}_n, \widehat{\nu}_n$ converges to $\beta^*, \eta^*, \nu^*$ at the rate of $\widetilde{\mathcal{O}}(n^{-3/8})$, respectively. Again, in our numerical experiments below we empirically verify this behavior.

In our final result, whose proof can be found in Appendix A.4, we show that the slower converge rates for the MLE under non-distinguishability are in fact essentially minimax optimal.

**Theorem 4.** *Suppose that $f_0$ belongs to the family of Gaussian densities and $h_0 = h$. Then, the minimax lower bounds*

$$\inf_{\overline{G}_n} \sup_{G \in \Xi(l_n)} \mathbb{E}_{p_{G,n}} \left[ \|(\Delta\eta, \Delta\nu)\|^4 \times \| \exp(\overline{\tau}_n) - \exp(\tau)\|^2 \right] \gtrsim n^{-1/r},$$

$$\inf_{\overline{G}_n} \sup_{G \in \Xi(l_n)} \mathbb{E}_{p_{G,n}} \left[ \exp^2(\tau) \|(\Delta\eta, \Delta\nu)\|^2 \times \|(\overline{\beta}_n, \overline{\eta}_n, \overline{\nu}_n) - (\beta, \eta, \nu)\|^2 \right] \gtrsim n^{-1/r},$$

*hold for any sequence $(l_n)_{n\geq 1}$ and any $0 < r < 1,$ , where the infimum is over all estimators $\overline{G}_n$ taking values in $\Xi$.*

### 3.3 Practical Implications

There are two important practical implications for the design of a contaminated MoE model from our theoretical results.

*1. Softmax gating is more sample-efficient than input-free gating.* We observe that softmax gating yields faster convergence rates of prompt parameter estimation in contaminated MoE than input-free gating in [35]. In particular, when using input-free gating, Table 2 reveals that the rates for estimating expert parameters and variance depend on the convergence rate of the gating parameter to zero. By contrast, when using softmax gating, estimation rates for expert parameters and variance become significantly faster as the previous rate dependence disappears. Therefore, our theories encourage the use of softmax gating over input-free gating when tuning contaminated-MoE-based models.

*2. Prompt models should have different expertise from pre-trained models.* It can be seen from Table 2 that when the prompt model acquires overlapping knowledge with the pre-trained model (non-distinguishable setting), the convergence rates of parameter estimation are slower than when these models have distinct knowledge (distinguishable setting). Thus, our theories advocate using prompt models with different expertise from the pre-trained model.

Table 2: Comparison of parameter estimation rates in input-free-contaminated MoE [35] and softmax-contaminated MoE (Ours). Below, we consider gating parameters $\exp(\beta_0^*)$, expert parameters $\eta^*$, and variance $\nu^*$. In addition, $\lambda^*$ denotes the constant weight in input-free-contaminated MoE.

| **Distinguishable Setting** | | |
| --- | --- | --- |
| | Gating parameters | Expert parameters and Variance |
| **Input-free gating [35]** | $\widetilde{\mathcal{O}}(n^{-1/2})$ | $\widetilde{\mathcal{O}}(n^{-1/2}(\lambda^*)^{-1})$ |
| **Softmax gating (Ours)** | $\widetilde{\mathcal{O}}(n^{-1/2})$ | $\widetilde{\mathcal{O}}(n^{-1/2})$ |
| **Non-distinguishable Setting** | | |
| | Gating parameters | Expert parameters and Variance |
| **Input-free gating [35]** | $\widetilde{\mathcal{O}}(n^{-\frac{1}{2}} \cdot \|(\Delta\eta^*, \Delta\nu^*)\|^{-2})$ | $\widetilde{\mathcal{O}}(n^{-\frac{1}{2}} \cdot \|(\Delta\eta^*, \Delta\nu^*)\|^{-1}(\lambda^*)^{-1})$ |
| **Softmax gating (Ours)** | $\widetilde{\mathcal{O}}(n^{-\frac{1}{2}} \cdot \|(\Delta\eta^*, \Delta\nu^*)\|^{-2})$ | $\widetilde{\mathcal{O}}(n^{-\frac{1}{2}} \cdot \|(\Delta\eta^*, \Delta\nu^*)\|^{-1})$ |

## 4 Numerical Experiments

In this section, we present several numerical experiments to verify our theoretical findings.

**Experimental setup.** Recall that, in the distinguishable setting, the pre-trained model $f_0$ does not belong to the Gaussian density family. Thus, we let $f_0$ be the density of a Laplace distribution, with mean function $h_0(x, \eta_0) = \tanh(\eta_0^\top x)$ and variance $\nu_0$. Here, $\eta_0$ is a $d$-dimensional vector defined as $e_1 := (1, 0, \ldots, 0)$, and $\nu_0 = 0.001$. Meanwhile, the prompt $f$ is formulated as a Gaussian density,

with the same $\tanh$ mean function but a different parameter $\eta^*$—i.e., $h(x, \eta^*) = \tanh((\eta^*)^\top x)$—and variance $\nu^*$.

On the other hand, in the non-distinguishable setting , both $f$ and $f_0$ belong to the Gaussian density family, and $h$ and $h_0$ are expert functions of the same form (albeit parameterized by different values of $\eta_0$ and $\eta^*$). As in the previous case, we let the expert function be the $\tanh$ function: in the pre-trained model, the expert is $h(x, \eta_0) = \tanh(\eta_0^\top x)$, and in the prompt model, it is $h(x, \eta^*) = \tanh((\eta^*)^\top x)$.

**Synthetic data generation.** We create synthetic datasets following the model outlined in equation (1). Specifically, we generate data pairs $\{(X_i, Y_i)\}_{i=1}^n \in \mathcal{X} \times \mathcal{Y} \subset \mathbb{R}^d \times \mathbb{R}$ by first drawing each covariate $X_i$ independently from a standard Gaussian distribution, for $i = 1, \ldots, n$, and consistently set $d = 8$ across all trials. The responses $Y_i$ are drawn from the density $p_{G_*}(y|x)$, where $G_* = (\beta^*, \tau^*, \eta^*, \nu^*)$:

(a) In the distinguishable setting, we let $\beta^* = 1/\sqrt{d} \cdot \mathbf{1}_d, \tau^* = 1, \eta^* = -e_1 = -\eta_0$ and $\nu^* = \nu_0 = 0.001$.

(b) In the non-distinguishable setting, we examine two cases to study the MLE convergence behavior as either $\eta^*$ or $\nu^*$ varies with $n$: in the first, $\eta^*$ is an $\mathcal{O}(n^{-1/8})$ perturbation of $\eta_0$ with $\nu^*$ fixed at $\nu_0$; in the second, $\eta^* = -\eta_0$ while $\nu^*$ is perturbed around $\nu_0$ at the same rate. In detail, we set:

   (i) In the first case, $\beta^* = 1/\sqrt{d} \cdot \mathbf{1}_d$, $\tau^* = 1$, $\eta^* = e_1(1 + n^{-1/8}) = \eta_0(1 + n^{-1/8})$, and $\nu^* = \nu_0 = 0.001$.

   (ii) In the second case, $\beta^* = 1/\sqrt{d} \cdot \mathbf{1}_d$, $\tau^* = 1$, $\eta^* = -e_1 = -\eta_0$, and $\nu^* = 0.001(1 + n^{-1/8}) = \nu_0(1 + n^{-1/8})$.

**Training procedure.** We conduct 40 experiments and, for each of them, consider 20 different sample sizes $n$, ranging from $10^3$ to $10^5$. In computing the MLEs, the initialization is set relatively close to the true parameter values to mitigate potential optimization instabilities. We use an EM algorithm [16] to compute the MLE, employing an off-the-shelf BFGS optimizer for the M-step due to the absence of a universal closed-form solution. All the numerical experiments are performed on a MacBook Air with an Apple M4 chip.

**Results.** The experimental results are presented in Figure 1 and Figure 2, where the x-axis displays varying sample sizes $n$, and the y-axis shows the parameter estimation error. We now present a detailed analysis of the results shown in each figure:

(a) Figure 1 displays the results for Theorem 1. We observe that the convergence rates of $(\widehat{\beta}_n, \widehat{\tau}_n, \widehat{\eta}_n, \widehat{\nu}_n)$ are $\mathcal{O}(n^{-0.45}), \mathcal{O}(n^{-0.52}), \mathcal{O}(n^{-0.50}), \mathcal{O}(n^{-0.54})$, respectively, aligning with the theoretical rates of order $\mathcal{O}(n^{-1/2})$ in Theorem 1.

(b) On the other hand, Figure 2 illustrates the parameter estimation errors for the simulations conducted in the non-distinguishable setting as Theorem 3.

   (i) In the first case, $\eta^*$ converges to $\eta_0$ at the rate of $\mathcal{O}(n^{-1/8})$, while $\nu^*$ remains fixed, Figure 2a shows that the convergence rate of $\exp(\widehat{\tau}_n)$ to $\exp(\tau^*)$ is $\mathcal{O}(n^{-0.23})$, which is consistent with the expected rate of $\mathcal{O}(n^{-1/4})$. The convergence rates for $\widehat{\beta}_n, \widehat{\eta}_n$, and $\widehat{\nu}_n$ are $\mathcal{O}(n^{-0.37})$, $\mathcal{O}(n^{-0.39})$, and $\mathcal{O}(n^{-0.35})$, respectively, all of which are approximately $\mathcal{O}(n^{-0.375})$, as they hinge on the vanishing rate $\mathcal{O}(n^{-3/8})$. These empirical rates are consistent with the theoretical rates in Theorem 3.

   (ii) In the alternative setting, $\eta^*$ is held fixed, while $\nu^*$ converges to $\nu_0$ at the rate of $\mathcal{O}(n^{-1/8})$. Figure 2b reveals that the convergence rate of $\exp(\widehat{\tau}_n)$ to $\exp(\tau^*)$ is of order $\mathcal{O}(n^{-0.22})$, again close to $\mathcal{O}(n^{-1/4})$. Meanwhile, the MLEs $\widehat{\beta}_n, \widehat{\eta}_n$, and $\widehat{\nu}_n$ still empirically converge to $\beta^*, \eta^*$, and $\nu^*$ at rates of $\mathcal{O}(n^{-0.39})$, $\mathcal{O}(n^{-0.37})$, and $\mathcal{O}(n^{-0.39})$, respectively, which align well with the theoretical rates $\widetilde{\mathcal{O}}(n^{-3/8})$. This observation is consistent with the theoretical convergence rates in Theorem 3.

## 5 Conclusion

In this paper, we characterize the convergence behavior of maximum likelihood estimators for parameters in the softmax-contaminated MoE model formulated as a mixture of a frozen pre-trained

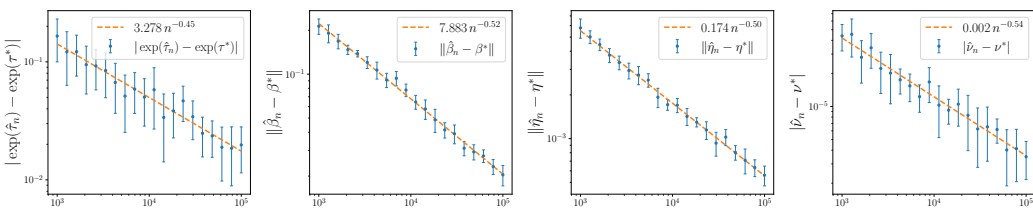

Figure 1: (**Distinguishable Setting:** $f_0$ is the density of a Laplace distribution.) Log-log graphs depicting the empirical convergence rates of the MLE $(\widehat{\beta}_n, \widehat{\tau}_n, \widehat{\eta}_n, \widehat{\nu}_n)$ to the ground-truth values $(\beta^*, \tau^*, \eta^*, \nu^*)$. The blue lines display the parameter estimation errors, while the orange dashed dotted lines are the fitted lines, highlighting the empirical MLE convergence rates.

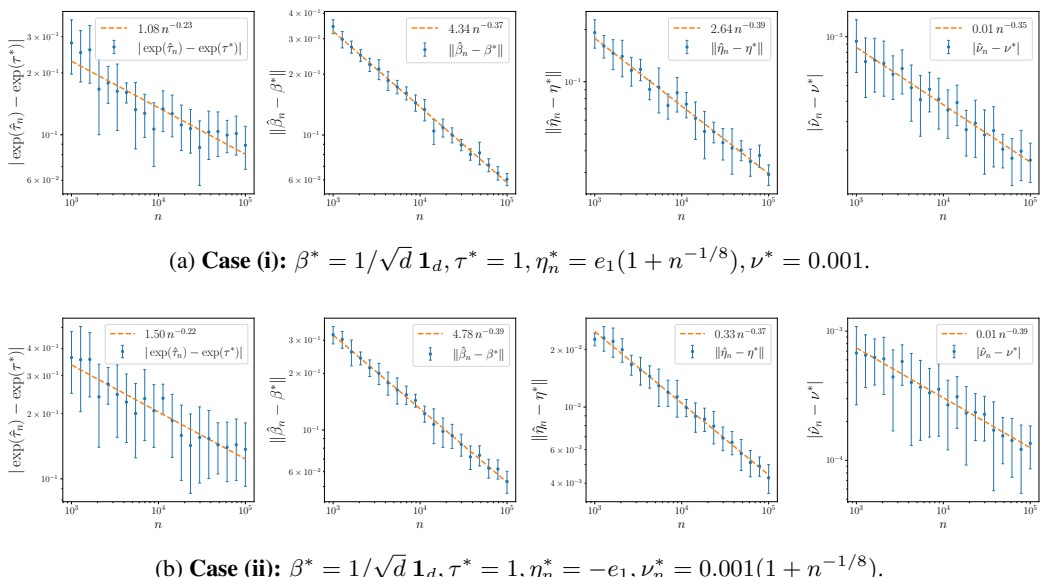

(a) **Case (i):** $\beta^* = 1/\sqrt{d}\,\mathbf{1}_d, \tau^* = 1, \eta_n^* = e_1(1 + n^{-1/8}), \nu^* = 0.001$.

(b) **Case (ii):** $\beta^* = 1/\sqrt{d}\,\mathbf{1}_d, \tau^* = 1, \eta_n^* = -e_1, \nu_n^* = 0.001(1 + n^{-1/8})$.

Figure 2: (**Non-distinguishable Setting:** $f_0$ is a Gaussian density.) Log-log graphs depicting the empirical convergence rates of the MLE $(\widehat{\beta}_n, \widehat{\tau}_n, \widehat{\eta}_n, \widehat{\nu}_n)$ to the ground-truth values $(\beta^*, \tau^*, \eta^*, \nu^*)$. The blue lines display the parameter estimation errors, while the orange dashed dotted lines are the fitted lines, highlighting the empirical MLE convergence rates. Figure 2a and Figure 2b illustrates results for Case (i) and Case (ii), respectively.

model and a trainable prompt model. To capture the challenge in which the prompt model admits the same expertise as the pre-trained model, we propose a novel analytic distinguishability condition and divide our analysis based on that condition. When the distinguishability condition is satisfied, we obtain minimax optimal parameter estimation rates of parametric order in the sample size, which are faster than those under the contaminated MoE with input-free gating. Conversely, when the distinguishability condition is violated, these rates become substantially slower than the parametric rates as they hinge on the convergence rates of prompt parameters to pre-trained parameters.

Based on our theoretical analysis, we make the following observations. First, the softmax gating helps to improve the sample efficiency for estimating the parameters in the contaminated MoE compared to the input-free gating. Second, the convergence rates for parameter estimation will be negatively affected if the prompt model acquires overlapping knowledge with the pre-trained model, thereby increasing the sample complexity of parameter estimation.

In future work, we plan to consider a more challenging setting of the contaminated MoE where the pre-trained model is fine-tuned by multiple prompt models rather than a single prompt as in the current setting. Furthermore, we can also generalize the analysis to the scenario where the prompt models belong to various families of distributions, rather than being restricted to Gaussian distributions.

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

# Supplementary Material for "On Minimax Estimation of Parameters in Softmax-Contaminated Mixture of Experts"

In this supplementary material, we provide the theoretical proofs omitted from the main text. Appendix A presents the proofs of our main results, including the theorems on convergence rates for parameter estimation and the minimax lower bounds stated in Section 3. Proofs of auxiliary results concerning the fundamental properties of the softmax-contaminated MoE model, introduced in Section 2, are deferred to Appendix B.

## A  Proof of Main Results

In this section, we present the proofs of the MLE rate theorem and the minimax lower bound theorem from Section 3, covering both distinguishable and non-distinguishable settings.

### A.1   Proof of Theorem 1

We begin by proving Theorem 1 under the distinguishable setting.

*Proof of Theorem 1.* Let $\overline{G} = (\bar{\beta}, \bar{\tau}, \bar{\eta}, \bar{\nu})$, we need to demonstrate that

$$\lim_{\varepsilon \to 0} \inf_{G, G_*} \left\{ \frac{\mathbb{E}_X[d_V(p_G(\cdot|X), p_{G_*}(\cdot|X))]}{D_1(G, G_*)} : D_1(G, \overline{G}) \vee D_1(G_*, \overline{G}) \leq \varepsilon \right\} > 0.$$

Using the argument with Fatou's lemma as in Theorem 3.1, [12], it is sufficient to show that

$$\lim_{\varepsilon \to 0} \inf_{G, G_*} \left\{ \frac{\|p_G - p_{G_*}\|_\infty}{D_1(G, G_*)} : D_1(G, \overline{G}) \vee D_1(G_*, \overline{G}) \leq \varepsilon \right\} > 0.$$

Assume by contrary that the above claim is not true. Then, there exist two sequences $G_n = (\beta_n, \tau_n, \eta_n, \nu_n)$ and $G_{*,n} = (\beta_n^*, \tau_n^*, \eta_n^*, \nu_n^*)$, such that when $n$ tends to infinity, we get

$$\begin{cases} D_1(G_n, \overline{G}) \to 0, \\ D_1(G_{*,n}, \overline{G}) \to 0, \\ \|p_{G_n} - p_{G_{*,n}}\|_\infty / D_1(G_n, G_{*,n}) \to 0. \end{cases}$$

In this proof, we will take into account only the most challenging setting of $(\beta_n, \eta_n, \nu_n)$ and $(\beta_n^*, \eta_n^*, \nu_n^*)$ when they converge to the same limit point $(\beta', \eta', \nu')$, where $(\beta', \eta', \nu')$ is not necessarily equal to $(\bar{\beta}, \bar{\eta}, \bar{\nu})$.

**Step 1: Density Decomposition.** Subsequently, we consider $Q_n(Y|X) = [1 + \exp((\beta_n)^\top X + \tau_n)] \cdot [p_{G_n}(Y|X) - p_{G_{*,n}}(Y|X)]$, which can decomposed as

$$Q_n(Y|X) = \exp(\tau_n) \left[ \exp((\beta_n)^\top X) f(Y|h(X, \eta_n), \nu_n) - \exp((\beta_n^*)^\top X) f(Y|h(X, \eta_n^*)), \nu_n^*) \right] := \mathrm{I}_n$$
$$- \exp(\tau_n) \left[ \exp((\beta_n)^\top X) - \exp((\beta_n^*)^\top X) \right] p_{G_{*,n}}(Y|X) := \mathrm{II}_n$$
$$+ \left[ \exp(\tau_n) - \exp(\tau_n^*) \right] \exp((\beta_n^*)^\top X) \left[ f(Y|h(X, \eta_n^*), \nu_n^*) - p_{G_{*,n}}(Y|X) \right]$$

Based on the first order Taylor expansion, $\mathrm{I}_n$ and $\mathrm{II}_n$ could be denoted as

$$\mathrm{I}_n = \exp(\tau_n) \sum_{|\alpha|=1} \frac{1}{2^{\alpha_3} \alpha!} (\beta_n - \beta_n^*)^{\alpha_1} (\eta_n - \eta_n^*)^{\alpha_2} (\nu_n - \nu_n^*)^{\alpha_3}$$

$$\cdot X^{\alpha_1} \exp((\beta_n^*)^\top X) \cdot \frac{\partial^{|\alpha_2| + 2\alpha_3} f}{\partial h^{|\alpha_2| + 2\alpha_3}} (Y|h(X, \eta_n^*), \nu_n^*) \frac{\partial^{\alpha_2} h}{\partial^{\alpha_2} \eta} (X, \eta_n^*) + R_1(Y|X)$$

$$= \exp(\tau_n) \sum_{2|\ell_1| + \ell_2 = 1}^{2} \sum_{\alpha \in \mathcal{I}_{\ell_1, \ell_2}} \frac{1}{2^{\alpha_4} \alpha!} (\beta_n - \beta_n^*)^{\alpha_1} (\eta_n - \eta_n^*)^{\alpha_2} (\nu_n - \nu_n^*)^{\alpha_3}$$

$$\cdot X^{\ell_1} \exp((\beta_n^*)^\top X) \cdot \frac{\partial^{\ell_2} f}{\partial h^{\ell_2}} (Y|h(X, \eta_n^*), \nu_n^*) \frac{\partial^{\alpha_2} h}{\partial^{\alpha_2} \eta} (X, \eta_n^*) + R_1(Y|X) \qquad (10)$$

where $\ell_1 = \alpha_1, \ell_2 = |\alpha_2| + 2\alpha_3$, and

$$\mathcal{I}_{\ell_1,\ell_2} := \left\{ \alpha = (\alpha_i)_{i=1}^3 \in \mathbb{N}^d \times \mathbb{N}^q \times \mathbb{N} : \alpha_1 = \ell_1, 2\alpha_3 = \ell_2 - |\alpha_2| \right\}, \tag{11}$$

for all $(\ell_1, \ell_2) \in \mathbb{N}^d \times \mathbb{N}$ such that $1 \leq 2|\ell_1| + \ell_2 \leq 2$.

Similarly, $\mathrm{II}_n$ can be expressed as:

$$\mathrm{II}_n = -\exp(\tau_n) \sum_{|\gamma|=1} \frac{1}{\gamma!} (\beta_n - \beta_n^*)^\gamma X^\gamma \exp((\beta_n^*)^\top X) p_{G_{*,n}}(Y|X) + R_2(Y|X). \tag{12}$$

Here $R_p(Y|X)/D_1(G_n, G_{*,n}) \to 0$ as $n \to \infty$, where $R_p(X, Y), p \in [2]$ are Taylor remainders . Consequently, $Q_n$ can be expressed as:

$$Q_n = \sum_{2|\ell_1|+\ell_2=0}^{2} T_{\ell_1,\ell_2}^n \cdot X^{\ell_1} \exp((\beta_n^*)^\top X) \frac{\partial^{\alpha_2} h}{\partial^{\alpha_2} \eta}(X, \eta_n^*) \frac{\partial^{\ell_2} f}{\partial h^{\ell_2}}(Y|h(X, \eta_n^*), \nu_n^*)$$

$$+ \sum_{|\gamma|=0}^{1} S_\gamma^n \cdot X^\gamma \exp((\beta_n^*)^\top X) p_{G_{*,n}}(Y|X), \tag{13}$$

with coefficients $T_{\ell_1,\ell_2}^n$ and $S_\gamma^n$ are defined for any $0 \leq 2|\ell_1| + \ell_2 \leq 2$ , and $0 \leq |\gamma| \leq 1$ as:

$$T_{\ell_1,\ell_2}^n = \begin{cases} \exp(\tau_n) \sum_{\alpha \in \mathcal{I}_{\ell_1,\ell_2}} \frac{1}{2^{\alpha_3} \alpha!} (\beta_n - \beta_n^*)^{\alpha_1} (\eta_n - \eta_n^*)^{\alpha_2} (\nu_n - \nu_n^*)^{\alpha_3}, & (\ell_1, \ell_2) \neq (0_d, 0), \\ \exp(\tau_n) - \exp(\tau_n^*), & (\ell_1, \ell_2) = (0_d, 0); \end{cases}$$

and

$$S_\gamma^n = \begin{cases} -\exp(\tau_n) \frac{1}{\gamma!} (\beta_n - \beta_n^*)^\gamma, & |\gamma| \neq 0, \\ -\exp(\tau_n) + \exp(\tau_n^*), & |\gamma| = 0. \end{cases}$$

where $Q_n$ can be viewed as linear combinations of elements of the set $\mathcal{H}_1$ defined as

$$\mathcal{H}_1 = \left\{ X^{\ell_1} \exp((\beta_n^*)^\top X) \frac{\partial^{\alpha_2} h}{\partial \eta^{\alpha_2}}(X, \eta_n^*) \frac{\partial^{\ell_2} f}{\partial h^{\ell_2}}(Y|h(X, \eta_n^*), \nu_n^*), X^\gamma \exp((\beta_n^*)^\top X) p_{G_{*,n}}(Y|X) \right\}. \tag{14}$$

**Step 2: Non-vanishing coefficients.** In this step, we will use a contradiction argument to demonstrate that not all the coefficients in the set

$$\mathcal{S}_1 = \left\{ \frac{T_{\ell_1,\ell_2}^n}{D_{1n}}, \frac{S_\gamma^n}{D_{1n}} : 0 \leq 2|\ell_1| + \ell_2 \leq 2, 0 \leq |\gamma| \leq 1 \right\} \tag{15}$$

vanish as $n \to \infty$ where $D_{1n} := D_1(G_n, G_{*,n})$. Specifically, suppose that all these coefficients converge to zero, when $n \to \infty$, then we get,

$$\frac{|\exp(\tau_n) - \exp(\tau_n^*)|}{D_{1n}} = \frac{|T_{0_d,0}^n(j)|}{D_{1n}} \to 0, \tag{16}$$

Similarly, by analyzing the limits of $T_{\ell_1,\ell_2}^n/D_{1n}$ s.t. $1 \leq 2|\ell_1| + \ell_2 \leq 2$, we conclude that:

$$\frac{\exp(\tau_n)(\beta_n - \beta_n^*)^{(u)}}{D_{1n}} \to 0, \frac{\exp(\tau_n)(\eta_n - \eta_n^*)^{(v)}}{D_{1n}} \to 0, \frac{\exp(\tau_n)(\nu_n - \nu_n^*)}{D_{1n}} \to 0,$$

as $n \to \infty$ for all $u \in [d], v \in [q]$. Given that our parameter lies in a compact set, there exists a positive constant $C$ such that $|\exp(\tau_n^*)/\exp(\tau_n)| \leq C$. Thus, we have

$$\frac{\exp(\tau_n^*)(\beta_n - \beta_n^*)^{(u)}}{D_{1n}} \to 0, \frac{\exp(\tau_n^*)(\eta_n - \eta_n^*)^{(v)}}{D_{1n}} \to 0, \frac{\exp(\tau_n^*)(\nu_n - \nu_n^*)}{D_{1n}} \to 0,$$

The limits imply that

$$(\exp(\tau_n) + \exp(\tau_n^*))\|(\beta_n, \eta_n, \nu_n) - (\beta_n^*, \eta_n^*, \nu_n^*)\|/D_{1n} \to 0. \tag{17}$$

Combining the results in equations (16) and (17) with the formulation of $D_{1n}$, we deduce that

$$1 = [|\exp(\tau_n) - \exp(\tau_n^*)| + (\exp(\tau_n) + \exp(\tau_n^*))\|(\beta_n, \eta_n, \nu_n) - (\beta_n^*, \eta_n^*, \nu_n^*)\|]/D_{1n} \to 0,$$

which is a contradiction. Thus, not all the coefficients in the set $\mathcal{S}_1$ tend to 0 as $n \to \infty$.

**Step 3 - Application of Fatou's lemma.** Let us denote by $m_n$ the maximum of the absolute values of those coefficients. It follows from the previous result that $1/m_n \not\to \infty$. Then $|T^n_{\ell_1,\ell_2}|/(m_n D_{1n})$ and $|S^n_\gamma|/(m_n D_{1n})$ remain bounded, we can consider subsequences of these terms, ensuring that: $|T^n_{\ell_1,\ell_2}|/m_n D_{1n} \to \eta_{\ell_1,\ell_2}$, $|S^n_\gamma|/m_n D_{1n} \to \omega_\gamma$, as $n \to \infty$ for all $0 \le 2|\ell_1| + \ell_2 \le 2, 0 \le |\gamma| \le 1$. Here, at least one among $\eta_{\ell_1,\ell_2}(j)$ and $\omega_\gamma(j)$ is different from zero. By applying the Fatou's lemma, we get

$$\lim_{n\to\infty} \frac{\mathbb{E}_X[d_V(p_{G_n}(\cdot|X), p_{G_*}(\cdot|X))]}{m_n D_{1n}} \ge \int \liminf_{n\to\infty} \frac{|p_{G_n}(Y|X) - p_{G_*}(Y|X)|}{2m_n D_{1n}} d(X,Y) \tag{18}$$

Under the given assumption, the left-hand side of the equation (18) is zero. Consequently, the integrand on the right-hand side of the equation (18) must also be zero almost surely with respect to $(X,Y)$. This results in:

$$\sum_{2|\ell_1|+\ell_2=0}^{2} \eta_{\ell_1,\ell_2} \cdot X^{\ell_1} \exp((\beta^*)^\top X) \frac{\partial^{\alpha_2} h}{\partial \eta^{\alpha_2}}(X, \eta^*) \frac{\partial^{\ell_2} f}{\partial h^{\ell_2}}(Y|h(X, \eta^*), \nu^*)$$

$$+ \sum_{|\gamma|=0}^{1} \omega_\gamma \cdot X^\gamma \exp((\beta^*)^\top X) p_{G_*}(Y|X) = 0,$$

for almost surely $(X,Y)$. Furthermore, by Lemma 1, the collection

$$\mathcal{W}_1 := \left\{ X^{\ell_1} \exp((\beta^*)^\top X) \frac{\partial^{\alpha_2} h}{\partial \eta^{\alpha_2}}(X, \eta^*) \frac{\partial^{\ell_2} f}{\partial h^{\ell_2}}(Y|h(X, \eta^*), \nu^*) : 0 \le \ell_2 \le 2 \right\}$$

$$\cup \left\{ X^\gamma \exp((\beta^*)^\top X) p_{G_*}(Y|X) \right\} \tag{19}$$

is linearly independent with respect to $(X,Y)$. Consequently, it follows that $\eta_{\ell_1,\ell_2} = \omega_\gamma = 0$, for all $0 \le 2|\ell_1| + \ell_2 \le 2, 0 \le |\gamma| \le 1$. But this contradicts that from the definition, at least one among $\eta_{\ell_1,\ell_2}, \omega_\gamma$ is nonzero. Hence, we reach the desired conclusion. $\square$

**Lemma 1.** *Suppose that $f_0$ is distinguishable with $f$, then the set $\mathcal{W}_1$ defined in equation (19) is linearly independent w.r.t. $(X,Y)$.*

*Proof of Lemma 1.* Recall the set

$$\mathcal{W}_1 := \left\{ X^{\ell_1} \exp((\beta^*)^\top X) \frac{\partial^{\alpha_2} h}{\partial \eta^{\alpha_2}}(X, \eta^*) \frac{\partial^{\ell_2} f}{\partial h^{\ell_2}}(Y|h(X, \eta^*), \nu^*) : 0 \le \ell_2 \le 2 \right\}$$

$$\cup \left\{ X^\gamma \exp((\beta^*)^\top X) p_{\lambda^*, G_*}(Y|X) \right\}$$

and the density

$$p_{G_*}(Y|X) := \frac{1}{1 + \exp((\beta^*)^\top X + \tau^*)} \cdot f_0(Y|h(X, \eta_0), \nu_0)$$

$$+ \frac{\exp((\beta^*)^\top X + \tau^*)}{1 + \exp((\beta^*)^\top X + \tau^*)} \cdot f(Y|h((X, \eta^*), \nu^*).$$

In words, $p_{G_*}$ is a convex combination (depending on $X$) of

$$f_0(Y|h(X, \eta_0), \nu_0) \quad \text{and} \quad f(Y|h(X, \eta^*), \nu^*).$$

Noting that the term in set $\mathcal{W}_1$ can be divided as the density function or its first and second derivatives

$$p_{G_*}(Y|X), f(Y|h(X, \eta^*), \frac{\partial f}{\partial h}(Y|h(X, \eta^*)), \frac{\partial^2 f}{\partial h^2}(Y|h(X, \eta^*)),$$

along with the factor involving only $X$.

**Step 1: Distinguishable property with respect to $Y$.**

First, fix $X$. Suppose for contradiction that there exist real numbers $c_0, c_1, c_2, d$ (may depend on $X$), not all zero, such that

$$c_0 \frac{\partial^0 f}{\partial h^0} + c_1 \frac{\partial^1 f}{\partial h} + c_2 \frac{\partial^2 f}{\partial h^2} + dp_{G_*}(Y|X) = 0, \quad \text{for almost every } Y.$$

Note that $\frac{\partial^0 f}{\partial h^0} = f$. Hence we have

$$c_0 f(Y|h(X, \eta^*), \nu^*) + c_1 \frac{\partial f}{\partial h}(Y|h(X, \eta^*), \nu^*) + c_2 \frac{\partial^2 f}{\partial h^2}(Y|h(X, \eta^*), \nu^*) + dp_{G_*}(Y|X) = 0.$$

Since

$$p_{G_*}(Y|X) = \phi(X)f_0(\cdot) + (1 - \phi(X))f(\cdot), \quad \phi(X) := \frac{1}{1 + \exp((\beta^*)^\top X + \tau^*)},$$

the above can be rewritten as

$$\left[c_0 + d(1 - \phi(X))\right]f(\cdot) + c_1 \frac{\partial f}{\partial h}(\cdot) + c_2 \frac{\partial^2 f}{\partial h^2}(\cdot) + d\phi(X)f_0(\cdot) = 0.$$

Using the hypothesis about the distinguishable property of $f_0$ with respect to $f$ as well as the Gaussian property of $f$, which implies $\partial^2 f/\partial h^2 = 1/2 \cdot \partial f/\partial \nu$, we have

$$d\phi(X) = 0 \quad \text{for almost all } X, \quad \text{and} \quad c_0 + d(1 - \phi(X)) = 0 \quad \text{for almost all } X,$$

and simultaneously $c_1 = c_2 = 0$. But $\phi(X) \neq 0$ on a set of $X$-values of positive measure , so $d = 0$. Plugging $d = 0$ into $c_0 + d(1 - \phi(X)) = 0$ yields $c_0 = 0$. Hence $c_0 = c_1 = c_2 = d = 0$. Since no nontrivial linear combination of $\{f, \frac{\partial f}{\partial \sigma}, \frac{\partial^2 f}{\partial \sigma^2}, p_{G_*}\}$ can vanish almost everywhere, these four functions are linearly independent when $X$ is fixed. This completes the proof of step 1.

**Step 2: Distinguishable property with respect to $X$.**

Let us consider coefficients appear in each density factor.

• Term related to $p_{G_*}(Y|X)$: The factor appearing along with $p_{G_*}(Y|X)$ are $\exp((\beta^*)^\top X)$, and $X^{(i)} \exp((\beta^*)^\top X)$, where $1 \leq i \leq d$. Suppose there exists constants $c, a_1, \ldots, a_d$ such that

$$c \exp((\beta^*)^\top X) + \sum_{i=1}^{d} a_i X^{(i)} \exp((\beta^*)^\top X) = 0, \ a.s.$$

This equation means that $c + \sum_{i=1}^{d} a_i X^{(i)} = 0$, a.s. Given that $X$ has non-vanish almost everywhere density function, this relation implies that $c = 0, a_i = 0, 1 \leq i \leq d$.

• Terms related to $f(Y|h(X, \eta^*), \nu^*)$: The factors appearing along with $f(Y|h(X, \eta^*), \nu^*)$ are $\exp((\beta^*)^\top X)$, and $X^{(i)} \exp((\beta^*)^\top X)$, where $1 \leq i \leq d$. The identical argument as in the case for $p_{G_*}(Y|X)$ also gives us the independency.

• Terms related to $\frac{\partial f}{\partial h}(Y|h(X, \eta^*), \nu^*)$: The factors appearing along with $p_{G_*}(Y|X)$ are

$$\frac{\partial h}{\partial \eta^{(i)}}(X, \eta^*) \exp((\beta^*)^\top X), \ 1 \leq i \leq d.$$

Suppose there exists constants $a_1, \ldots, a_d$ not all equal to zero such that

$$\sum_{i=1}^{d} a_i \frac{\partial h}{\partial \eta^{(i)}}(X, \eta^*) \exp((\beta^*)^\top X) = 0, \ a.s.$$

This equation means that

$$\sum_{i=1}^{d} a_i \frac{\partial h}{\partial \eta^{(i)}}(X, \eta^*) = 0, \ a.s., \quad \text{or } \nabla_a h(X, \eta^*) = 0, \ a.s.,$$

where $a = (a_1, \ldots, a_d)$. This is a contradiction.

• Terms related to $\frac{\partial^2 f}{\partial h^2}(Y|h(X, \eta^*))$: There is only one such term is

$$\exp((\beta^*)^\top X) \frac{\partial^2 f}{\partial h^2}(Y|h(X, \eta^*)).$$

Its coefficient obviously vanishes from the independent property with respect to $Y$.

This completes the proof of Lemma 1. □

## A.2 Proof of Theorem 2

As a first step in proving the minimax lower bounds for the distinguishable setting (Theorem 2), we define two distances:

$$d_1(G_1, G_2) = \exp(\tau_1) \|(\beta_1, \eta_1, \nu_1) - (\beta_2, \eta_2, \nu_2)\|,$$
$$d_2(G_1, G_2) = |\exp(\tau_1) - \exp(\tau_2)|^2,$$

for any $G_1 = (\beta_1, \tau_1, \eta_1, \nu_1) \in \Xi$ and $G_2 = (\beta_2, \tau_2, \eta_2, \nu_2) \in \Xi$. Obviously $d_2(G_1, G_2)$ is a proper distance. The structure for $d_1(G_1, G_2)$ tells us that it is not symmetric. Only when $\tau_1 = \tau_2 = \tau$, $d_1(G_1, G_2)$ is symmetric. Also $d_1(G_1, G_2)$ still satisfies a weak triangle inequality:

$$d_1(G_1, G_2) + d_1(G_2, G_3) \geq \min\{d_1(G_1, G_2), d_1(G_2, G_3)\}.$$

Therefore, we will apply the modified Le Cam method for nonsymmetric loss, as outlined in Lemma C.1 of [9], to handle this distance. For $f$ satisfies all assumptions in Theorem 2, based on the Taylor expansion, we have the following results:

**Lemma 2.** *Given $f$ in Theorem 2, we denote*

$$S_1 = (\tau, \beta_1, \eta_1, \nu_1), S_2 = (\tau, \beta_2, \eta_2, \nu_2), \text{ and } S_1' = (\tau_1, \beta, \eta, \nu), S_2' = (\tau_2, \beta, \eta, \nu),$$

*we achieve for any $r < 1$ that*

*(i)* $\displaystyle \lim_{\epsilon \to 0} \inf_{S_1, S_2} \left\{ \frac{\mathbb{E}_X[d_H(p_{S_1}(\cdot|X), p_{S_2}(\cdot|X))]}{d_1^r(S_1, S_2)} : d_1(S_1, S_2) \leq \epsilon \right\} = 0,$

*(ii)* $\displaystyle \lim_{\epsilon \to 0} \inf_{S_1', S_2'} \left\{ \frac{\mathbb{E}_X[d_H(p_{S_1'}(\cdot|X), p_{S_2'}(\cdot|X))]}{d_2^r(S_1', S_2')} : d_2(S_1', S_2') \leq \epsilon \right\} = 0.$

We will prove this lemma later.

*Proof of Theorem 2.* Denote $G_* = (\beta^*, \tau^*, \eta^*, \nu^*)$ and assume $r < 1$. Given Lemma 2 part (i), for any sufficiently small $\epsilon > 0$, there exists $G_*' = (\beta_1^*, \tau^*, \eta_1^*, \nu_1^*)$ such that $d_1(G_*, G_*') = d_1(G_*', G_*) = \epsilon$, there exists a constant $C_0$, s.t.

$$\mathbb{E}_X[d_H(p_{G_*}(\cdot|X), p_{G_*'}](\cdot|X)) \leq C_0 \epsilon^r. \tag{20}$$

Now we will denote $p_{G_*}^n$ as the density of the $n$-i.i.d. sample $(X_1, Y_1), \cdots, (X_n, Y_n)$. Lemma C.1 in [9] tells us that

$$\inf_{\overline{G}_n \in \Xi} \sup_{G \in \Xi} \mathbb{E}_{p_G} \left( \exp^2(\tau) \|(\overline{\beta}_n, \overline{\eta}_n, \overline{\nu}_n) - (\beta, \eta, \nu)\|^2 \right) \geq \frac{\epsilon^2}{2} \left( 1 - \mathbb{E}_X[d_V(p_{G_*}^n(\cdot|X), p_{G_*'}^n(\cdot|X))] \right)$$

$$\geq \frac{\epsilon^2}{2}\sqrt{1-(1-C_0^2\epsilon^{2r})^n}.$$

Last inequality is from the definition of the Total Variation distance and Hellinger distance and equation (20). Let $\epsilon^{2r} = \dfrac{1}{C_0^2 n}$, then for any $r < 1$ we have

$$\inf_{\overline{G}_n \in \Xi} \sup_{G \in \Xi} \mathbb{E}_{p_G}\left(\exp^2(\tau)\|(\overline{\beta}_n, \overline{\eta}_n, \overline{\nu}_n) - (a, b, \nu)\|^2\right) \geq c_1 n^{-1/r},$$

where $c_1$ is some positive constant. Following a similar reasoning and using Lemma 2 part (ii) , we will obtain

$$\inf_{\overline{G}_n \in \Xi} \sup_{G \in \Xi} \mathbb{E}_{p_G}\left(|\exp(\overline{\tau}_n) - \exp(\tau)|^2\right) \geq c_2 n^{-1/r},$$

for some positive constant $c_2$. Consequently, we establish all of the results for Theorem 2.  □

*Proof of Lemma 2 (i) .*  Consider two sequences

$$S_{1,n} = (\tau_n, \beta_{1,n}, \eta_{1,n}, \nu_{1,n}),$$
$$S_{2,n} = (\tau_n, \beta_{2,n}, \eta_{2,n}, \nu_{2,n}),$$

with the same $\tau_n$. By the contaminated MoE model definition, we have

$$p_{S_{j,n}}(Y|X) = \frac{1}{1 + \exp(\beta_{j,n}^\top X + \tau_n)} f_0(Y|h_0(X, \eta_0), \nu_0)$$
$$+ \frac{\exp(\beta_{j,n}^\top X + \tau_n)}{1 + \exp(\beta_{j,n}^\top X + \tau_n)} f(Y|h(X, \eta_{j,n}), \nu_{j,n}),$$

for $j = 1, 2$. Since $(\tau_n, \beta_{j,n})$ lie in a compact set, and both $f_0$ and $f$ are non-negative. Hence, the squared Hellinger distance satisfies

$$\mathbb{E}_X[d_H^2(p_{S_{1,n}}(\cdot|X), p_{S_{2,n}}(\cdot|X))] \leq C \int \left(\frac{p_{S_{1,n}}(Y|X) - p_{S_{2,n}}(Y|X)}{p_{S_{2,n}}(Y|X)}\right)^2 d(X, Y)$$
$$\leq C' \int \left[\frac{\exp(\beta_{1,n}^\top X) f(Y|h(X, \eta_{1,n}), \nu_{1,n}) - \exp(\beta_{2,n}^\top X) f(Y|h(X, \eta_{2,n}), \nu_{2,n})}{\exp(\beta_{2,n}^\top X) f(Y|h(X, \eta_{2,n}), \nu_{2,n})}\right]^2 d(X, Y),$$

for some constants $C, C'$ depending on the compactness bounds.

Consider the Taylor expansion of the map

$$(\beta, \eta, \nu) \mapsto \exp(\beta^\top X) f(Y|h(X, \eta), \nu)$$

at the point $(\beta_{2,n}, \eta_{2,n}, \nu_{2,n})$, expanded up to first order with integral remainder. Let $\alpha = (\alpha_1, \alpha_2, \alpha_3)$ denote a multi-index where $\alpha_1 \in \mathbb{N}^d$, $\alpha_2 \in \mathbb{N}^q$, and $\alpha_3 \in \mathbb{N}$ index components of $\beta$, $\eta$, and $\nu$, respectively. Then we have:

$$\exp(\beta_{1,n}^\top X) f(Y|h(X, \eta_{1,n}), \nu_{1,n}) - \exp(\beta_{2,n}^\top X) f(Y|h(X, \eta_{2,n}), \nu_{2,n})$$
$$= \sum_{|\alpha|=1} \frac{(\beta_{1,n} - \beta_{2,n})^{\alpha_1}(\eta_{1,n} - \eta_{2,n})^{\alpha_2}(\nu_{1,n} - \nu_{2,n})^{\alpha_3}}{\alpha_1! \alpha_2! \alpha_3!}$$
$$\cdot X^{\alpha_1} \exp(\beta_{2,n}^\top X) \frac{\partial^{|\alpha_2|+\alpha_3} f}{\partial \eta^{\alpha_2} \partial \nu^{\alpha_3}}(Y|h(X, \eta_{2,n}), \nu_{2,n})$$
$$+ \sum_{|\alpha|=1} \frac{(\beta_{1,n} - \beta_{2,n})^{\alpha_1}(\eta_{1,n} - \eta_{2,n})^{\alpha_2}(\nu_{1,n} - \nu_{2,n})^{\alpha_3}}{\alpha_1! \alpha_2! \alpha_3!} \int_0^1 X^{\alpha_1} \exp\left((\beta_{2,n} + t(\beta_{1,n} - \beta_{2,n}))^\top X\right)$$
$$\cdot \frac{\partial^{|\alpha_2|+\alpha_3} f}{\partial \eta^{\alpha_2} \partial \nu^{\alpha_3}}(Y|h(X, \eta_{2,n} + t(\eta_{1,n} - \eta_{2,n})), \nu_{2,n} + t(\nu_{1,n} - \nu_{2,n})) dt.$$

So it follows that

$$\frac{\mathbb{E}_X[d_H^2(p_{S_{1,n}}(\cdot|X), p_{S_{2,n}}(\cdot|X))]}{d_1^{2r}(S_{1,n}, S_{2,n})} \to 0.$$

since $\tau_n$ lies in a compact set. This establishes part (i) of the lemma.

$\square$

*Proof of Lemma 2 (ii).* We consider two sequences

$$S'_{1,n} = (\tau_{1,n}, \beta_n, \eta_n, \nu_n),$$
$$S'_{2,n} = (\tau_{2,n}, \beta_n, \eta_n, \nu_n),$$

with different $\tau_{1,n} \neq \tau_{2,n}$ but the same $(\beta_n, \eta_n, \nu_n)$.

Using the contaminated MoE definition, the difference in conditional densities is:

$$p_{S'_{1,n}}(Y|X) - p_{S'_{2,n}}(Y|X) = \frac{e^{\beta_n^\top X}\left(e^{\tau_{2,n}} - e^{\tau_{1,n}}\right)}{\left(1 + e^{\beta_n^\top X + \tau_{1,n}}\right)\left(1 + e^{\beta_n^\top X + \tau_{2,n}}\right)}$$
$$\cdot \left[f(Y|h(X, \eta_n), \nu_n) - f_0(Y|h_0(X, \eta_0), \nu_0)\right].$$

By the standard bound for squared Hellinger distance,

$$\mathbb{E}_X[d_H^2(p_{S'_{1,n}}(\cdot|X), p_{S'_{2,n}}(\cdot|X))] \leq C \int \left(\frac{p_{S'_{1,n}}(Y|X) - p_{S'_{2,n}}(Y|X)}{p_{S'_{2,n}}(Y|X)}\right)^2 d(X, Y).$$

Since $(\beta_n, \eta_n, \nu_n)$ lie in a compact set, and both $f$ and $f_0$ are bounded away from zero, we have $p_{S'_{2,n}}(Y|X) \geq c > 0$. So the denominator is lower bounded.

Then there exists a constant $C'$ such that:

$$\mathbb{E}_X[d_H^2(p_{S'_{1,n}}(\cdot|X), p_{S'_{2,n}}(\cdot|X))] \leq C'\left(e^{\tau_{1,n}} - e^{\tau_{2,n}}\right)^2.$$

Now recall the definition of the distance:

$$d_2((\tau_{1,n}, \beta_n, \eta_n, \nu_n), (\tau_{2,n}, \beta_n, \eta_n, \nu_n)) := |e^{\tau_{1,n}} - e^{\tau_{2,n}}|^2.$$

So we conclude:

$$\frac{\mathbb{E}_X[d_H^2(p_{S'_{1,n}}(\cdot|X), p_{S'_{2,n}}(\cdot|X))]}{d_2((S'_{1,n}, S'_{2,n}))^r} \leq \frac{C'|e^{\tau_{1,n}} - e^{\tau_{2,n}}|^2}{|e^{\tau_{1,n}} - e^{\tau_{2,n}}|^{2r}} = C'|e^{\tau_{1,n}} - e^{\tau_{2,n}}|^{2(1-r)} \to 0$$

as long as $e^{\tau_{1,n}} - e^{\tau_{2,n}} \to 0$, and $r < 1$.

Hence,

$$\frac{\mathbb{E}_X[d_H^2(p_{S'_{1,n}}(\cdot|X), p_{S'_{2,n}}(\cdot|X))]}{d_2^r(S'_{1,n}, S'_{2,n})} \to 0,$$

which proves part (ii).

$\square$

### A.3 Proof of Theorem 3

We proceed to prove Theorem 3 for the non-distinguishable setting.

*Proof.* Let $\overline{G} = (\bar{\beta}, \bar{\tau}, \bar{\eta}, \bar{\nu})$ and $(\bar{\eta}, \bar{\nu})$ can be identical to $(\eta_0, \nu_0)$. Then, we will show that

(i) When $(\eta_0, \nu_0) \neq (\bar{\eta}, \bar{\nu})$,

$$\lim_{\varepsilon \to 0} \inf_{G, G_*} \left\{\frac{\|p_G - p_{G_*}\|_\infty}{D_1(G, G_*)} : D_1(G, \overline{G}) \vee D_1(G_*, \overline{G}) \leq \varepsilon\right\} > 0.$$

(ii) When $(\eta_0, \nu_0) = (\bar{\eta}, \bar{\nu})$,

$$\lim_{\varepsilon \to 0} \inf_{G, G_*} \left\{ \frac{\|p_G - p_{G_*}\|_\infty}{D_2(G, G_*)} : D_2(G, \overline{G}) \vee D_2(G_*, \overline{G}) \leq \varepsilon \right\} > 0. \tag{21}$$

Part (i) can be proved by using the same arguments as in the proof A.1. Thus, we will consider only part (ii) in this section, specifically the most challenging setting that $(\eta_0, \nu_0) = (\bar{\eta}, \bar{\nu})$. Under this assumption, we know that $h_0$ and $h$ are the same expert function, s.t. $f_0(Y|h_0(X, \eta_0), \nu_0) = f(Y|h(X, \eta_0), \nu_0)$ for almost surely $(X, Y) \in \mathcal{X} \times \mathcal{Y}$. Assume that the above claim in equation (21) does not hold, then there exist two sequences $G_n = (\beta_n, \tau_n, \eta_n, \nu_n)$ and $G_{*,n} = (\beta_n^*, \tau_n^*, \eta_n^*, \nu_n^*)$, such that

$$\begin{cases} D_2(G_n, \overline{G}) \to 0, \\ D_2(G_{*,n}, \overline{G}) \to 0, \\ \|p_{G_n} - p_{G_{*,n}}\|_\infty / D_2(G_n, G_{*,n}) \to 0. \end{cases}$$

We now analyze the limiting behavior of the sequences $(\lambda_n, G_n)$ and $(\lambda_n^*, G_n^*)$ as they approach $(\bar{\lambda}, \overline{G})$. In particular, we distinguish between three asymptotic regimes based on how the expert parameters $\varsigma_n = (\eta_n, \nu_n)$ and $\varsigma_n^* = (\eta_n^*, \nu_n^*)$ converge.

First, it may occur that both $\varsigma_n$ and $\varsigma_n^*$ converge to the same limit $\varsigma_0 = (\eta_0, \nu_0)$. Alternatively, both sequences may converge to a common limit $\varsigma' \neq \varsigma_0$, which is distinct from the true expert. Finally, it is also possible that one sequence converges to $\varsigma_0$ while the other converges to a different point $\varsigma' \neq \varsigma_0$.

In the following, we analyze each of these cases and demonstrate that in all scenarios, the assumption that the normalized difference vanishes leads to a contradiction when $f_0 = f$.

**Case 1:**

At first we consider that $(\eta_n, \nu_n)$ and $(\eta_n^*, \nu_n^*)$ share the same limit of $(\eta_0, \nu_0)$. Without loss of generality, we can suppose that $\tau_n^* \geq \tau_n$. Subsequently, we consider $W_n := [p_{G_n}(Y|X) - p_{G_{*,n}}(Y|X)] \cdot [1 + \exp((\beta_n^*)^\top X + \tau_n^*)] \cdot [1 + \exp((\beta_n)^\top X + \tau_n)]$, which can decomposed as

$$\begin{aligned} W_n &= \exp(\tau_n) \cdot [g(Y|X; \beta_n, \eta_n, \nu_n) - g(Y|X; \beta_n^*, \eta_n^*, \nu_n^*)] \\ &\quad - \exp(\tau_n) \cdot [g(Y|X; \beta_n, \eta_0, \nu_0) - g(Y|X; \beta_n^*, \eta_n^*, \nu_n^*)] \\ &\quad + \exp(\tau_n^*) \cdot [g(Y|X; \beta_n^*, \eta_0, \nu_0) - g(Y|X; \beta_n^*, \eta_n^*, \nu_n^*)] \\ &\quad + \exp\left((\beta_n^* + \beta_n)^\top X + \tau_n^* + \tau_n\right) \cdot [f(Y|h(X, \eta_n), \nu_n) - f(Y|h(X, \eta_n^*), \nu_n^*)] \\ &:= \mathrm{I}_n - \mathrm{II}_n + \mathrm{III}_n + \mathrm{IV}_n \end{aligned}$$

where we denote $g(Y|X; \beta, \eta, \nu) = e(X; \beta)f(Y|X; \eta, \nu) = \exp\left(\beta^\top X\right) f\left(Y|h(X, \eta), \nu\right)$.

We expand around the reference parameters $\beta_n^*, \eta_n^*, \nu_n^*$, where the parameter differences are given by $\Delta\eta_n = \eta_n - \eta_0, \Delta\nu_n = \nu_n - \nu_0$, and $\Delta\eta_n^* = \eta_n^* - \eta_0, \Delta\nu_n^* = \nu_n^* - \nu_0$. Applying a second-order Taylor expansion, then we obtain:

$$\begin{aligned} \mathrm{I}_n &= \exp(\tau_n)\Big[ \sum_{|\alpha|=1}^{2} \frac{1}{\alpha!} \prod_{u=1}^{d} [(\beta_n - \beta_n^*)^{(u)}]^{\alpha_{1u}} \prod_{v=1}^{q} [(\Delta\eta_n - \Delta\eta_n^*)^{(v)}]^{\alpha_{2v}} (\Delta\nu_n - \Delta\nu_n^*)^{\alpha_3} \\ &\qquad \cdot \frac{\partial^{|\alpha|} g}{\partial\beta^{\alpha_1} \partial\eta^{\alpha_2} \partial\nu^{\alpha_3}}(Y|X; \beta_n^*, \eta_n^*, \nu_n^*) + R_1(X, Y)\Big] \\ &= \exp(\tau_n)\Big[ \sum_{|\alpha|=1}^{2} \frac{1}{\alpha! 2^{\alpha_3}} \prod_{u=1}^{d} [(\beta_n - \beta_n^*)^{(u)}]^{\alpha_{1u}} \prod_{v=1}^{q} [(\Delta\eta_n - \Delta\eta_n^*)^{(v)}]^{\alpha_{2v}} (\Delta\nu_n - \Delta\nu_n^*)^{\alpha_3} \\ &\qquad \cdot \exp((\beta_n^*)^\top X) \cdot X^{\alpha_1} \frac{\partial^{|\alpha_2|} h}{\partial\eta^{|\alpha_2|}}(X, \eta_n^*) \frac{\partial^{|\alpha_2|+2\alpha_3} f}{\partial h^{|\alpha_2|+2\alpha_3}}(Y|h(X, \eta_n^*), \nu_n^*) + R_1(X, Y)\Big], \end{aligned}$$

$$\tag{22}$$

where $R_1(X,Y)$ is the remainder term containing higher-order terms, and the second equality is due to $\frac{\partial f}{\partial \nu} = \frac{1}{2}\frac{\partial^2 f}{\partial h^2}$. Similarly, we will have that

$$\mathrm{II}_n = \exp(\tau_n)\Big[\sum_{|\alpha|=1}^{2}\frac{1}{\alpha!}\prod_{u=1}^{d}[(\beta_n - \beta_n^*)^{(u)}]^{\alpha_{1u}}\prod_{v=1}^{q}[(\Delta\eta_n^*)^{(v)}]^{\alpha_{2v}}(\Delta\nu_n^*)^{\alpha_3}$$
$$\cdot\frac{\partial^{|\alpha|}g}{\partial\beta^{\alpha_1}\partial\eta^{\alpha_2}\partial\nu^{\alpha_3}}(Y|X;\beta_n^*,\eta_n^*,\nu_n^*) + R_2(X,Y)\Big],$$

$$\mathrm{III}_n = \exp(\tau_n^*)\Big[\sum_{|\alpha|=1}^{2}\frac{1}{\alpha!}\prod_{v=1}^{q}[(\Delta\eta_n^*)^{(v)}]^{\alpha_{2v}}(\Delta\nu_n^*)^{\alpha_3}\frac{\partial^{|\alpha|}g}{\partial\eta^{\alpha_2}\partial\nu^{\alpha_3}}(Y|X;\beta_n^*,\eta_n^*,\nu_n^*) + R_3(X,Y)\Big],$$

$$\mathrm{IV}_n = \exp(\tau_n^* + \tau_n)\exp\big((\beta_n^* + \beta_n)^\top X\big)\Big[\sum_{|\alpha|=1}^{2}\frac{1}{\alpha!}\prod_{v=1}^{q}[(\Delta\eta_n - \Delta\eta_n^*)^{(v)}]^{\alpha_{2v}}(\Delta\nu_n - \Delta\nu_n^*)^{\alpha_3}$$
$$\cdot\frac{\partial^{|\alpha|}f}{\partial\eta^{\alpha_2}\partial\nu^{\alpha_3}}(Y|X;\eta_n^*,\nu_n^*) + R_4(X,Y)\Big].$$

Then, grouping the terms according to the order of derivative $\gamma := |\alpha_2| + 2\alpha_3$ and the monomial degree $\zeta := |\alpha_1|$, we can rewrite the expansion in the compact form:

$$\mathrm{I}_n = \sum_{\zeta=0}^{2}\Big[\sum_{\gamma=0}^{4}\mathrm{I}_{n,\gamma,\zeta}(X)\frac{\partial^\gamma f}{\partial h^\gamma}(Y|h(X,\eta_n^*),\nu_n^*)\exp((\beta_n^*)^\top X)\Big]X^\zeta + R_1(X,Y)$$

where each coefficient $\mathrm{I}_{n,\gamma,\zeta}(X)$ depends on the parameter differences and derivatives of $h$ with respect to $\eta$. More specifically we have that

$$\mathrm{I}_{n,0,1}(X) = \exp(\tau_n)\sum_{1\leq w\leq d}(\beta_n - \beta_n^*)^{(w)}$$

$$\mathrm{I}_{n,0,2}(X) = \exp(\tau_n)\sum_{1\leq w,r\leq d}\frac{(\beta_n - \beta_n^*)^{(w)}(\beta_n - \beta_n^*)^{(r)}}{1 + \mathbf{1}_{w=r}}$$

$$\mathrm{I}_{n,1,0}(X) = \exp(\tau_n)\Big[\sum_{u=1}^{q}\{(\Delta\eta_n - \Delta\eta_n^*)^{(u)}\}\frac{\partial h}{\partial\eta^{(u)}}(X,\eta_n^*)$$
$$+ \sum_{1\leq u,v\leq q}\frac{(\Delta\eta_n - \Delta\eta_n^*)^{(u)}(\Delta\eta_n - \Delta\eta_n^*)^{(v)}}{1 + \mathbf{1}_{u=v}}\frac{\partial^2 h}{\partial\eta^{(u)}\partial\eta^{(v)}}(X,\eta_n^*)\Big],$$

$$\mathrm{I}_{n,1,1}(X) = \exp(\tau_n)\Big[\sum_{1\leq w\leq d, 1\leq u\leq q}[(\beta_n - \beta_n^*)^{(w)}][(\Delta\eta_n - \Delta\eta_n^*)^{(u)}]\frac{\partial h}{\partial\eta^{(u)}}(X,\eta_n^*)\Big],$$

$$\mathrm{I}_{n,2,0}(X) = \exp(\tau_n)\Big[\frac{1}{2}(\Delta\nu_n - \Delta\nu_n^*)+$$
$$\sum_{1\leq u,v\leq q}\frac{(\Delta\eta_n - \Delta\eta_n^*)^{(u)}(\Delta\eta_n - \Delta\eta_n^*)^{(v)}}{1 + \mathbf{1}_{u=v}}\frac{\partial h}{\partial\eta^{(u)}}(X,\eta_n^*)\frac{\partial h}{\partial\eta^{(v)}}(X,\eta_n^*)\Big],$$

$$\mathrm{I}_{n,2,1}(X) = \frac{\exp(\tau_n)}{2}\Big[\sum_{1\leq w\leq d, 1\leq u\leq q}(\beta_n - \beta_n^*)^{(w)}(\Delta\nu_n - \Delta\nu_n^*)^{(u)}\Big],$$

$$\mathrm{I}_{n,3,0}(X) = \frac{\exp(\tau_n)}{2}\Big[\sum_{u=1}^{q}(\Delta\eta_n - \Delta\eta_n^*)^{(u)}(\Delta\nu_n - \Delta\nu_n^*)\frac{\partial h}{\partial\eta^{(u)}}(X,\eta_n^*)\Big],$$

$$\mathrm{I}_{n,4,0}(X) = \frac{\exp(\tau_n)}{8}(\Delta\nu_n - \Delta\nu_n^*)^2.$$

Similarly, we can rewrite $\mathrm{II}_n$ in the same fashion as follows:

$$\mathrm{II}_n = \sum_{\zeta=0}^{2}\Big[\sum_{\gamma=0}^{4}\mathrm{II}_{n,\gamma,\zeta}(X)\frac{\partial^\gamma f}{\partial h^\gamma}(Y|h(X,\eta_n^*),\nu_n^*)\exp((\beta_n^*)^\top X)\Big]X^\zeta + R_2(X,Y)$$

where

$$\mathrm{II}_{n,0,1}(X) = \exp(\tau_n) \sum_{1 \le w \le d} (\beta_n - \beta_n^*)^{(w)}$$

$$\mathrm{II}_{n,0,2}(X) = \exp(\tau_n) \sum_{1 \le w,r \le d} \frac{(\beta_n - \beta_n^*)^{(w)}(\beta_n - \beta_n^*)^{(r)}}{1 + \mathbf{1}_{w=r}}$$

$$\mathrm{II}_{n,1,0}(X) = \exp(\tau_n) \Big[ \sum_{u=1}^{q} \{(-\Delta\eta_n^*)^{(u)}\} \frac{\partial h}{\partial \eta^{(u)}}(X, \eta_n^*)$$

$$+ \sum_{1 \le u,v \le q} \frac{(-\Delta\eta_n^*)^{(u)}(-\Delta\eta_n^*)^{(v)}}{1 + \mathbf{1}_{u=v}} \frac{\partial^2 h}{\partial \eta^{(u)} \partial \eta^{(v)}}(X, \eta_n^*) \Big],$$

$$\mathrm{II}_{n,1,1}(X) = \exp(\tau_n) \Big[ \sum_{1 \le w \le d, 1 \le u \le q} [(\beta_n - \beta_n^*)^{(w)}][(-\Delta\eta_n^*)^{(u)}] \frac{\partial h}{\partial \eta^{(u)}}(X, \eta_n^*) \Big],$$

$$\mathrm{II}_{n,2,0}(X) = \exp(\tau_n) \Big[ \frac{1}{2}(-\Delta\nu_n^*) + \sum_{1 \le u,v \le q} \frac{(-\Delta\eta_n^*)^{(u)}(-\Delta\eta_n^*)^{(v)}}{1 + \mathbf{1}_{u=v}} \frac{\partial h}{\partial \eta^{(u)}}(X, \eta_n^*) \frac{\partial h}{\partial \eta^{(v)}}(X, \eta_n^*) \Big],$$

$$\mathrm{II}_{n,2,1}(X) = \frac{\exp(\tau_n)}{2} \Big[ \sum_{1 \le w \le d, 1 \le u \le q} (\beta_n - \beta_n^*)^{(w)}(-\Delta\nu_n^*)^{(u)} \Big],$$

$$\mathrm{II}_{n,3,0}(X) = \frac{\exp(\tau_n)}{2} \Big[ \sum_{u=1}^{q} (-\Delta\eta_n^*)^{(u)}(-\Delta\nu_n^*) \frac{\partial h}{\partial \eta^{(u)}}(X, \eta_n^*) \Big],$$

$$\mathrm{II}_{n,4,0}(X) = \frac{\exp(\tau_n)}{8} (-\Delta\nu_n^*)^2.$$

In the same way, we can rewrite $\mathrm{III}_n$ in the same fashion as follows, here the difference for $\beta_n^*$ is zero, so all the coefficients with $\zeta \ne 0$ is zero, but in order for the alignment of the expression, we will still express $\mathrm{III}_n$ as follows

$$\mathrm{III}_n = \sum_{\gamma=1}^{4} \mathrm{III}_{n,\gamma,0}(X) \frac{\partial^\gamma f}{\partial h^\gamma}(Y|h(X, \eta_n^*), \nu_n^*) \exp((\beta_n^*)^\top X) + R_2(X, Y)$$

where

$$\mathrm{III}_{n,1,0}(X) = \exp(\tau_n^*) \Big[ \sum_{u=1}^{q} \{(-\Delta\eta_n^*)^{(u)}\} \frac{\partial h}{\partial \eta^{(u)}}(X, \eta_n^*)$$

$$+ \sum_{1 \le u,v \le q} \frac{(-\Delta\eta_n^*)^{(u)}(-\Delta\eta_n^*)^{(v)}}{1 + \mathbf{1}_{u=v}} \frac{\partial^2 h}{\partial \eta^{(u)} \partial \eta^{(v)}}(X, \eta_n^*) \Big],$$

$$\mathrm{III}_{n,2,0}(X) = \exp(\tau_n^*) \Big[ \frac{1}{2}(-\Delta\nu_n^*) + \sum_{1 \le u,v \le q} \frac{(-\Delta\eta_n^*)^{(u)}(-\Delta\eta_n^*)^{(v)}}{1 + \mathbf{1}_{u=v}} \frac{\partial h}{\partial \eta^{(u)}}(X, \eta_n^*) \frac{\partial h}{\partial \eta^{(v)}}(X, \eta_n^*) \Big],$$

$$\mathrm{III}_{n,3,0}(X) = \frac{\exp(\tau_n^*)}{2} \Big[ \sum_{u=1}^{q} (-\Delta\eta_n^*)^{(u)}(-\Delta\nu_n^*) \frac{\partial h}{\partial \eta^{(u)}}(X, \eta_n^*) \Big],$$

$$\mathrm{III}_{n,4,0}(X) = \frac{\exp(\tau_n^*)}{8} (-\Delta\nu_n^*)^2.$$

Now we consider $\mathrm{IV}_n = \exp\left((\beta_n^* + \beta_n)^\top X + \tau_n^* + \tau_n\right) \cdot [f(Y|\sigma(X, \eta_n), \nu_n) - f(Y|\sigma(X, \eta_n^*), \nu_n^*)]$, which is equivalent to

$$\mathrm{IV}_n = \sum_{\gamma=1}^{4} \mathrm{IV}_{n,\gamma,0}(X) \frac{\partial^\gamma f}{\partial h^\gamma}(Y|h(X, \eta_n^*), \nu_n^*) \exp((\beta_n^*)^\top X) \exp((\beta_n)^\top X) + R_4(X, Y)$$

where

$$\mathrm{IV}_{n,1,0}(X) = \exp(\tau_n^* + \tau_n)\Big[\sum_{u=1}^{q}\{(\Delta\eta_n - \Delta\eta_n^*)^{(u)}\}\frac{\partial h}{\partial\eta^{(u)}}(X,\eta_n^*)$$

$$+ \sum_{1\leq u,v\leq q}\frac{(\Delta\eta_n - \Delta\eta_n^*)^{(u)}(\Delta\eta_n - \Delta\eta_n^*)^{(v)}}{1 + \mathbf{1}_{u=v}}\frac{\partial^2 h}{\partial\eta^{(u)}\partial\eta^{(v)}}(X,\eta_n^*)\Big],$$

$$\mathrm{IV}_{n,2,0}(X) = \exp(\tau_n^* + \tau_n)\Big[\frac{1}{2}(\Delta\nu_n - \Delta\nu_n^*)$$

$$+ \sum_{1\leq u,v\leq q}\frac{(\Delta\eta_n - \Delta\eta_n^*)^{(u)}(\Delta\eta_n - \Delta\eta_n^*)^{(v)}}{1 + \mathbf{1}_{u=v}}\frac{\partial h}{\partial\eta^{(u)}}(X,\eta_n^*)\frac{\partial h}{\partial\eta^{(v)}}(X,\eta_n^*)\Big],$$

$$\mathrm{IV}_{n,3,0}(X) = \frac{\exp(\tau_n^* + \tau_n)}{2}\Big[\sum_{u=1}^{q}(\Delta\eta_n - \Delta\eta_n^*)^{(u)}(\Delta\nu_n - \Delta\nu_n^*)\frac{\partial h}{\partial\eta^{(u)}}(X,\eta_n^*)\Big],$$

$$\mathrm{IV}_{n,4,0}(X) = \frac{\exp(\tau_n)}{8}(\Delta\nu_n - \Delta\nu_n^*)^2.$$

Then we could conclude that

$$W_n = \sum_{\gamma=0}^{4}\Bigg[\left(\mathrm{I}_{n,\gamma,0}(X) + \mathrm{II}_{n,\gamma,0}(X) + \mathrm{III}_{n,\gamma,0}(X)\right)$$

$$+ \sum_{\zeta=1}^{2}\left(\mathrm{I}_{n,\gamma,\zeta}(X) + \mathrm{II}_{n,\gamma,\zeta}(X)\right)X^{\zeta} + \mathrm{IV}_{n,\gamma,0}(X)\exp((\beta_n)^{\top}X)\Bigg]$$

$$\cdot\frac{\partial^{\gamma}f}{\partial h^{\gamma}}(Y|h(X,\eta_n^*),\nu_n^*)\cdot\exp((\beta_n^*)^{\top}X).$$

Therefore, we can view the quantity $W_n/D_2(G_n, G_{*,n}))$ as a linear combination of elements of the set $\mathcal{L}\cup\mathcal{K}$, and $\mathcal{L} = \cup_{\gamma=0}^{4}\cup_{\zeta=0}^{2}\mathcal{L}_{\gamma,\zeta}$, $\mathcal{K} = \cup_{\gamma=1}^{4}\mathcal{K}_{\gamma}$, where

$$\mathcal{L}_{0,1} = \left\{Xf(Y|h(X,\eta_n^*),\nu_n^*))\exp((\beta_n^*)^{\top}X)\right\}$$

$$\mathcal{L}_{0,2} = \left\{XX^{\top}f(Y|h(X,\eta_n^*),\nu_n^*))\exp((\beta_n^*)^{\top}X)\right\}$$

$$\mathcal{L}_{1,1} = \left\{\frac{\partial h}{\partial\eta^{(u)}}(X,\eta_n^*)X\frac{\partial f}{\partial h}(Y|h(X,\eta_n^*),\nu_n^*))\exp((\beta_n^*)^{\top}X) : u\in[q]\right\}$$

$$\mathcal{L}_{2,1} = \left\{X\frac{\partial^2 f}{\partial h^2}(Y|h(X,\eta_n^*),\nu_n^*))\exp((\beta_n^*)^{\top}X)\right\}$$

$$\mathcal{L}_{1,0} = \left\{\frac{\partial h}{\partial\eta^{(u)}}(X,\eta_n^*)\frac{\partial f}{\partial h}(Y|h(X,\eta_n^*),\nu_n^*))\exp((\beta_n^*)^{\top}X) : u\in[d]\right\}$$

$$\cup\left\{\frac{\partial^2 h}{\partial\eta^{(u)}\partial\eta^{(v)}}(X,\eta_n^*)\frac{\partial f}{\partial h}(Y|h(X,\eta_n^*),\nu_n^*))\exp((\beta_n^*)^{\top}X) : u,v\in[d]\right\},$$

$$\mathcal{L}_{2,0} = \left\{\frac{\partial^2 f}{\partial h^2}(Y|h(X,\eta_n^*),\nu_n^*))\exp((\beta_n^*)^{\top}X)\right\}$$

$$\cup\left\{\frac{\partial h}{\partial\eta^{(u)}}(X,\eta_n^*)\frac{\partial h}{\partial\eta^{(v)}}(X,\eta_n^*)\frac{\partial^2 f}{\partial h^2}(Y|h(X,\eta_n^*),\nu_n^*))\exp((\beta_n^*)^{\top}X) : u,v\in[q]\right\}$$

$$\mathcal{L}_{3,0} = \left\{\frac{\partial h}{\partial\eta^{(u)}}(X,\eta_n^*)\frac{\partial^3 f}{\partial h^3}(Y|h(X,\eta_n^*),\nu_n^*))\exp((\beta_n^*)^{\top}X) : u\in[d]\right\}$$

$$\mathcal{L}_{4,0} = \left\{\frac{\partial^4 f}{\partial h^4}(Y|h(X,\eta_n^*),\nu_n^*))\exp((\beta_n^*)^{\top}X)\right\},$$

and

$$\mathcal{K}_1 = \left\{\frac{\partial h}{\partial\eta^{(u)}}(X,\eta_n^*)\exp((\beta_n)^{\top}X)\frac{\partial f}{\partial h}(Y|h(X,\eta_n^*),\nu_n^*))\exp((\beta_n^*)^{\top}X) : u\in[d]\right\}$$

$$\cup \left\{ \frac{\partial^2 h}{\partial \eta^{(u)} \partial \eta^{(v)}}(X, \eta_n^*) \exp((\beta_n)^\top X) \frac{\partial f}{\partial h}(Y|h(X, \eta_n^*), \nu_n^*)) \exp((\beta_n^*)^\top X) : u, v \in [d] \right\},$$

$$\mathcal{K}_2 = \left\{ \exp((\beta_n)^\top X) \frac{\partial^2 f}{\partial h^2}(Y|h(X, \eta_n^*), \nu_n^*)) \exp((\beta_n^*)^\top X) \right\}$$

$$\cup \left\{ \frac{\partial h}{\partial \eta^{(u)}}(X, \eta_n^*) \frac{\partial h}{\partial \eta^{(v)}}(X, \eta_n^*) \exp((\beta_n)^\top X) \frac{\partial^2 f}{\partial h^2}(Y|h(X, \eta_n^*), \nu_n^*)) \exp((\beta_n^*)^\top X) : u, v \in [q] \right\},$$

$$\mathcal{K}_3 = \left\{ \frac{\partial h}{\partial \eta^{(u)}}(X, \eta_n^*) \exp((\beta_n)^\top X) \frac{\partial^3 f}{\partial h^3}(Y|h(X, \eta_n^*), \nu_n^*)) \exp((\beta_n^*)^\top X) : u \in [d] \right\},$$

$$\mathcal{K}_4 = \left\{ \exp((\beta_n)^\top X) \frac{\partial^4 f}{\partial h^4}(Y|h(X, \eta_n^*), \nu_n^*)) \exp((\beta_n^*)^\top X) \right\}.$$

Assume by contrary that all the coefficients of these elements vanish when $n \to \infty$. Looking at the coefficients of $\frac{\partial h}{\partial \eta^{(u)}}(X, \eta_n^*) X \frac{\partial f}{\partial h}(Y|h(X, \eta_n^*), \nu_n^*)) \exp((\beta_n^*)^\top X)$, we get for all $w \in [d], u \in [q]$

$$\exp(\tau_n)[(\beta_n - \beta_n^*)^{(w)}][(\Delta \eta_n)^{(u)}]/D_2(G_n, G_{*,n}) \to 0, \tag{23}$$

Looking at the coefficients of $X \frac{\partial^2 f}{\partial h^2}(Y|h(X, \eta_n^*), \nu_n^*)) \exp((\beta_n^*)^\top X)$, we get for all $w \in [d]$

$$\exp(\tau_n)[(\beta_n - \beta_n^*)^{(w)}](\Delta \nu_n)/D_2(G_n, G_{*,n}) \to 0, \tag{24}$$

Looking at the coefficients of $\frac{\partial^2 h}{\partial \eta^{(u)} \partial \eta^{(v)}}(X, \eta_n^*) \frac{\partial f}{\partial h}(Y|h(X, \eta_n^*), \nu_n^*)) \exp((\beta_n^*)^\top X)$ , we get for all $u, v \in [q]$,

$$[\exp(\tau_n)(\Delta \eta_n - \Delta \eta_n^*)^{(u)}(\Delta \eta_n - \Delta \eta_n^*)^{(v)} + [\exp(\tau_n^*) - \exp(\tau_n)](-\Delta \eta_n^*)^{(u)}(-\Delta \eta_n^*)^{(v)}] \\ /D_2(G_n, G_{*,n}) \to 0, \tag{25}$$

Looking at the coefficients of $\frac{\partial h}{\partial \eta^{(u)}}(X, \eta_n^*) \frac{\partial f}{\partial h}(Y|h(X, \eta_n^*), \nu_n^*)) \exp((\beta_n^*)^\top X)$ , we get for all $u \in [q]$,

$$[\exp(\tau_n)(\Delta \eta_n - \Delta \eta_n^*)^{(u)} + [\exp(\tau_n^*) - \exp(\tau_n)](-\Delta \eta_n^*)^{(u)}] \\ /D_2(G_n, G_{*,n}) \to 0, \tag{26}$$

Looking at the coefficients of $\frac{\partial^2 f}{\partial h^2}(Y|h(X, \eta_n^*), \nu_n^*)) \exp((\beta_n^*)^\top X)$ , we get

$$[\exp(\tau_n)(\Delta \nu_n - \Delta \nu_n^*) + [\exp(\tau_n^*) - \exp(\tau_n)](-\Delta \nu_n^*)]/D_2(G_n, G_{*,n}) \to 0, \tag{27}$$

Looking at the coefficients of $\frac{\partial h}{\partial \eta^{(u)}}(X, \eta_n^*) \frac{\partial h}{\partial \eta^{(v)}}(X, \eta_n^*) \frac{\partial^2 f}{\partial h^2}(Y|h(X, \eta_n^*), \nu_n^*)) \exp((\beta_n^*)^\top X)$ , we get for all $u, v \in [q]$,

$$[\exp(\tau_n)(\Delta \eta_n - \Delta \eta_n^*)^{(u)}(\Delta \eta_n - \Delta \eta_n^*)^{(v)} + [\exp(\tau_n^*) - \exp(\tau_n)](-\Delta \eta_n^*)^{(u)}(-\Delta \eta_n^*)^{(v)}] \\ /D_2(G_n, G_{*,n}) \to 0, \tag{28}$$

Looking at the coefficients of $\frac{\partial h}{\partial \eta^{(u)}}(X, \eta_n^*) \frac{\partial^3 f}{\partial h^3}(Y|h(X, \eta_n^*), \nu_n^*)) \exp((\beta_n^*)^\top X)$ , we get for all $u \in [q]$,

$$[\exp(\tau_n)(\Delta \eta_n - \Delta \eta_n^*)^{(u)}(\Delta \nu_n - \Delta \nu_n^*) + [\exp(\tau_n^*) - \exp(\tau_n)](-\Delta \eta_n^*)^{(u)}(-\Delta \nu_n^*)] \\ /D_2(G_n, G_{*,n}) \to 0, \tag{29}$$

Looking at the coefficients of $\frac{\partial^4 f}{\partial h^4}(Y|h(X, \eta_n^*), \nu_n^*)) \exp((\beta_n^*)^\top X)$ , we get

$$[\exp(\tau_n)(\Delta \nu_n - \Delta \nu_n^*)^2 + [\exp(\tau_n^*) - \exp(\tau_n)](-\Delta \nu_n^*)^2]/D_2(G_n, G_{*,n}) \to 0, \tag{30}$$

Looking at the coefficients of $\frac{\partial h}{\partial \eta^{(u)}}(X, \eta_n^*) \exp((\beta_n)^\top X) \frac{\partial f}{\partial h}(Y|h(X, \eta_n^*), \nu_n^*)) \exp((\beta_n^*)^\top X)$, we get for all $u \in [q]$,

$$[\exp(\tau_n^* + \tau_n)(\Delta \eta_n - \Delta \eta_n^*)^{(u)}]/D_2(G_n, G_{*,n}) \to 0, \tag{31}$$

Looking at the coefficients of $\frac{\partial^2 h}{\partial \eta^{(u)} \partial \eta^{(v)}}(X, \eta_n^*) \exp((\beta_n)^\top X) \frac{\partial f}{\partial h}(Y|h(X, \eta_n^*), \nu_n^*)) \exp((\beta_n^*)^\top X)$, we get for all $u, v \in [q]$,

$$[\exp(\tau_n^* + \tau_n)(\Delta \eta_n - \Delta \eta_n^*)^{(u)}(\Delta \eta_n - \Delta \eta_n^*)^{(v)}]/D_2(G_n, G_{*,n}) \to 0, \tag{32}$$

Looking at the coefficients of $\exp((\beta_n)^\top X) \frac{\partial^2 f}{\partial h^2}(Y|h(X, \eta_n^*), \nu_n^*)) \exp((\beta_n^*)^\top X)$, we get

$$[\exp(\tau_n^* + \tau_n)(\Delta \nu_n - \Delta \nu_n^*)]/D_2(G_n, G_{*,n}) \to 0, \tag{33}$$

Looking at the coefficients of $\frac{\partial h}{\partial \eta^{(u)}}(X, \eta_n^*) \frac{\partial h}{\partial \eta^{(v)}}(X, \eta_n^*) \exp((\beta_n)^\top X) \frac{\partial^2 f}{\partial h^2}(Y|h(X, \eta_n^*), \nu_n^*)) \exp((\beta_n^*)^\top X)$, we get for all $u, v \in [q]$,

$$[\exp(\tau_n^* + \tau_n)(\Delta \eta_n - \Delta \eta_n^*)^{(u)}(\Delta \eta_n - \Delta \eta_n^*)^{(v)}]/D_2(G_n, G_{*,n}) \to 0, \tag{34}$$

Looking at the coefficients of $\frac{\partial h}{\partial \eta^{(u)}}(X, \eta_n^*) \exp((\beta_n)^\top X) \frac{\partial^3 f}{\partial h^3}(Y|h(X, \eta_n^*), \nu_n^*)) \exp((\beta_n^*)^\top X)$, we get for all $u \in [q]$,

$$\exp(\tau_n^* + \tau_n)(\Delta \eta_n - \Delta \eta_n^*)^{(u)}(\Delta \nu_n - \Delta \nu_n^*)/D_2(G_n, G_{*,n}) \to 0, \tag{35}$$

Looking at the coefficients of $\exp((\beta_n)^\top X) \frac{\partial^4 f}{\partial h^4}(Y|h(X, \eta_n^*), \nu_n^*)) \exp((\beta_n^*)^\top X)$, we get for all $u \in [q]$,

$$[\exp(\tau_n^* + \tau_n)(\Delta \nu_n - \Delta \nu_n^*)^2]/D_2(G_n, G_{*,n}) \to 0, \tag{36}$$

Now, combining (23) and (24), recall that all the gating parameters are in compact sets, and applying the Cauchy–Schwarz inequality followed by summation over coordinates, we got that

$$\exp(\tau_n)\|\beta_n - \beta_n^*\|\|(\Delta \eta_n, \Delta \nu_n)\|/D_2(G_n, G_{*,n}) \to 0. \tag{37}$$

While it is intuitive that the similar result holds for $\|(\Delta \eta_n^*, \Delta \nu_n^*)\|$, a slightly tricky handle should be employed here. Suppose that

$$\exp(\tau_n^*)\|\beta_n - \beta_n^*\|\|\Delta \eta_n^*\|/D_2(G_n, G_{*,n}) \not\to 0.$$

By combining this assumption with equation (23), we have there are at least one coordinate $u$ such that $|(\Delta \eta_n^*)^{(u)}/(\Delta \eta_n)^{(u)}| \to \infty$, which implies that $(\Delta \eta_n^*)/(\Delta \eta_n^* - \Delta \eta_n)^{(u)} \to 1$. Thus, by multiplying equation (31) with $(\Delta \eta_n^*)/(\Delta \eta_n^* - \Delta \eta_n)^{(u)} \to 1$, we have

$$\exp(\tau_n^*)(\Delta \eta_n^*)^{(u)}/D_2(G_n, G_{*,n}) \to 0.$$

Also noting that $\|\beta_n - \beta_n^*\|$ is bounded as the parameters belongs to a compact set, we have

$$\exp(\tau_n^*)\|\beta_n - \beta_n^*\|(\Delta \eta_n^*)^{(u)}/D_2(G_n, G_{*,n}) \to 0,$$

which is a contradiction here. Thus, we have

$$\exp(\tau_n^*)\|\beta_n - \beta_n^*\|\|\Delta \eta_n^*\|/D_2(G_n, G_{*,n}) \to 0. \tag{38}$$

Similarly, also by combining equation (24) and (33), we have

$$\exp(\tau_n^*)\|\beta_n - \beta_n^*\|\|\Delta \nu_n^*\|/D_2(G_n, G_{*,n}) \to 0. \tag{39}$$

As a result, we have

$$\exp(\tau_n^*)\|\beta_n - \beta_n^*\|\|(\Delta \eta_n^*, \Delta \nu_n^*)\|/D_2(G_n, G_{*,n}) \to 0. \tag{40}$$

In a similar manner, by considering equations (31) through (36), we obtain that

$$\exp(\tau_n + \tau_n^*) \cdot \|(\Delta\eta_n, \Delta\nu_n) - (\Delta\eta_n^*, \Delta\nu_n^*)\|^2 / D_2(G_n, G_{*,n}) \to 0. \tag{41}$$

Let $u = v$ in the first equation in equation (25), we achieve that for all $u \in [d]$,

$$[\exp(\tau_n)[(\Delta\eta_n - \Delta\eta_n^*)^{(u)}]^2 + [\exp(\tau_n^*) - \exp(\tau_n)][(\Delta\eta_n^*)^{(u)}]^2]/D_2(G_n, G_{*,n}) \to 0, \tag{42}$$

which implies that

$$[\exp(\tau_n)\|(\Delta\eta_n - \Delta\eta_n^*)\|^2 + (\exp(\tau_n^*) - \exp(\tau_n))\|\Delta\eta_n^*\|^2]/D_2(G_n, G_{*,n}) \to 0. \tag{43}$$

We also have each term inside equation (43) is non-negative, thus

$$(\exp(\tau_n^*) - \exp(\tau_n))\|\Delta\eta_n^*\|^2/D_2(G_n, G_{*,n}) \to 0,$$
$$\exp(\tau_n)\|\Delta\eta_n - \Delta\eta_n^*\|^2/D_2(G_n, G_{*,n}) \to 0. \tag{44}$$

Applying the AM-GM inequality, we have for all $u, v \in [d]$,

$$\frac{(\exp(\tau_n^*) - \exp(\tau_n))(\Delta\eta_n^*)^{(u)}(\Delta\eta_n^*)^{(v)}}{D_2(G_n, G_{*,n})} \to 0, \quad \frac{\exp(\tau_n)(\Delta\eta_n - \Delta\eta_n^*)^{(u)}(\Delta\eta_n - \Delta\eta_n^*)^{(v)}}{D_2(G_n, G_{*,n})} \to 0, \tag{45}$$

Next, by considering the coefficients of $\dfrac{\partial h}{\partial\eta^{(u)}}(X, \eta_n^*)\dfrac{\partial f}{\partial h}(Y|h(X, \eta_n^*), \nu_n^*))\exp((\beta_n^*)^\top X)$, and

$\dfrac{\partial^2 f}{\partial h^2}(Y|h(X, \eta_n^*), \nu_n^*))\exp((\beta_n^*)^\top X)$, we have

$$[\exp(\tau_n)(\Delta\eta_n)^{(u)} - \exp(\tau_n^*)(\Delta\eta_n^*)^{(u)}]/D_2(G_n, G_{*,n}) \to 0, \quad u \in [d], \tag{46}$$
$$[\exp(\tau_n)(\Delta\nu_n) - \exp(\tau_n^*)(\Delta\nu_n^*)]/D_2(G_n, G_{*,n}) \to 0. \tag{47}$$

Noting that for $u, v \in [d]$,

$$\exp(\tau_n^*)(\Delta\eta_n^*)^{(u)}(\Delta\eta_n - \Delta\eta_n^*)^{(v)}$$
$$= (\exp(\tau_n)(\Delta\eta_n)^{(v)} - \exp(\tau_n^*)(\Delta\eta_n^*)^{(v)})(\Delta\eta_n^*)^{(u)} + (\exp(\tau_n^*) - \exp(\tau_n))(\Delta\eta_n)^{(v)}(\Delta\eta_n^*)^{(u)},$$
$$\exp(\tau_n)(\Delta\eta_n)^{(u)}(\Delta\eta_n - \Delta\eta_n^*)^{(v)}$$
$$= \exp(\tau_n^*)(\Delta\eta_n^*)^{(u)}(\Delta\eta_n - \Delta\eta_n^*)^{(v)} - (\exp(\tau_n)(\Delta\eta_n)^{(u)} - \exp(\tau_n^*)(\Delta\eta_n^*)^{(u)})(\Delta\eta_n - \Delta\eta_n^*)^{(v)}.$$

Thus, from equation (45) and equation (46), we achieve that for $u, v \in [d]$,

$$\exp(\tau_n^*)(\Delta\eta_n^*)^{(u)}(\Delta\eta_n - \Delta\eta_n^*)^{(v)}/D_2(G_n, G_{*,n}) \to 0,$$
$$\exp(\tau_n)(\Delta\eta_n)^{(u)}(\Delta\eta_n - \Delta\eta_n^*)^{(v)}/D_2(G_n, G_{*,n}) \to 0.$$

By using the same arguments we will derive

$$\exp(\tau_n)\|\Delta\eta_n\|.\|\Delta\eta_n - \Delta\eta_n^*\|/D_2(G_n, G_{*,n}) \to 0, \tag{48}$$
$$\exp(\tau_n^*)\|\Delta\eta_n^*\|.\|\Delta\eta_n - \Delta\eta_n^*\|/D_2(G_n, G_{*,n}) \to 0, \tag{49}$$

By using the same arguments to derive equation (42), equation (44) and equation (45), we can point out that

$$[(\exp(\tau_n^*) - \exp(\tau_n))\|\Delta\nu_n^*\|^2 + \exp(\tau_n)\|\Delta\nu_n - \Delta\nu_n^*\|^2]/D_2(G_n, G_{*,n}) \to 0,$$
$$\exp(\tau_n)\|\Delta\nu_n\|.\|\Delta\nu_n - \Delta\nu_n^*\|/D_2(G_n, G_{*,n}) \to 0,$$
$$\exp(\tau_n^*)\|\Delta\nu_n^*\|.\|\Delta\nu_n - \Delta\nu_n^*\|/D_2(G_n, G_{*,n}) \to 0,$$
$$\exp(\tau_n)\|\Delta\eta_n\|.\|\Delta\nu_n - \Delta\nu_n^*\|/D_2(G_n, G_{*,n}) \to 0,$$
$$\exp(\tau_n^*)\|\Delta\eta_n^*\|.\|\Delta\nu_n - \Delta\nu_n^*\|/D_2(G_n, G_{*,n}) \to 0. \tag{50}$$

Collecting results in equation (37), (40) and (41), and equations (44) to (50), we obtain that

$$1 = D_2(G_n, G_{*,n})/D_2(G_n, G_{*,n}) \to 0,$$

which is a contradiction.

Therefore, not all the coefficients in the representation of $W_n/D_2(G_n, G_{*,n})$ tend to 0 as $n \to \infty$. Let us denote by $m_n$ the maximum of the absolute values of those coefficients. Based on the previous result, $1/m_n \not\to \infty$. Additionally, we define

$$\exp(\tau_n)[(\beta_n - \beta_n^*)^{(w)}][(\Delta\eta_n)^{(u)}]/m_n \to \alpha_{11,wu0},$$

$$\exp(\tau_n)[(\beta_n - \beta_n^*)^{(w)}](\Delta\nu_n)/m_n \to \alpha_{21,w00},$$

$$[\exp(\tau_n)(\Delta\eta_n - \Delta\eta_n^*)^{(u)} + [\exp(\tau_n^*) - \exp(\tau_n)](-\Delta\eta_n^*)^{(u)}]/m_n \to \alpha_{10,0u0},$$

$$[\exp(\tau_n)(\Delta\eta_n - \Delta\eta_n^*)^{(u)}(\Delta\eta_n - \Delta\eta_n^*)^{(v)} + [\exp(\tau_n^*) - \exp(\tau_n)](-\Delta\eta_n^*)^{(u)}(-\Delta\eta_n^*)^{(v)}]/m_n$$
$$\to \beta_{10,0uv},$$

$$[\exp(\tau_n)(\Delta\nu_n - \Delta\nu_n^*) + [\exp(\tau_n^*) - \exp(\tau_n)](-\Delta\nu_n^*)]/m_n \to \alpha_{20,000},$$

$$[\exp(\tau_n)(\Delta\eta_n - \Delta\eta_n^*)^{(u)}(\Delta\eta_n - \Delta\eta_n^*)^{(v)} + [\exp(\tau_n^*) - \exp(\tau_n)](-\Delta\eta_n^*)^{(u)}(-\Delta\eta_n^*)^{(v)}]/m_n$$
$$\to \beta_{20,0uv},$$

$$[\exp(\tau_n)(\Delta\eta_n - \Delta\eta_n^*)^{(u)}(\Delta\nu_n - \Delta\nu_n^*) + [\exp(\tau_n^*) - \exp(\tau_n)](-\Delta\eta_n^*)^{(u)}(-\Delta\nu_n^*)]/m_n$$
$$\to \beta_{30,0u0},$$

$$[\exp(\tau_n)(\Delta\nu_n - \Delta\nu_n^*)^2 + [\exp(\tau_n^*) - \exp(\tau_n)](-\Delta\nu_n^*)^2]/m_n \to \beta_{40,000},$$

$$\exp(\tau_n^* + \tau_n)(\Delta\eta_n - \Delta\eta_n^*)^{(u)}/m_n \to \rho_{1,u0},$$

$$\exp(\tau_n^* + \tau_n)(\Delta\eta_n - \Delta\eta_n^*)^{(u)}(\Delta\eta_n - \Delta\eta_n^*)^{(v)}/m_n \to \pi_{1,uv},$$

$$\exp(\tau_n^* + \tau_n)(\Delta\nu_n - \Delta\nu_n^*)/m_n \to \rho_{2,00},$$

$$\exp(\tau_n^* + \tau_n)(\Delta\eta_n - \Delta\eta_n^*)^{(u)}(\Delta\eta_n - \Delta\eta_n^*)^{(v)}/m_n \to \pi_{2,uv},$$

$$\exp(\tau_n^* + \tau_n)(\Delta\eta_n - \Delta\eta_n^*)^{(u)}(\Delta\nu_n - \Delta\nu_n^*)/m_n \to \pi_{3,u0},$$

$$\exp(\tau_n^* + \tau_n)(\Delta\nu_n - \Delta\nu_n^*)^2/m_n \to \pi_{4,00}, \tag{51}$$

when $n \to \infty$ for all $w \in [d], u, v \in [q]$. Note that at least one among $\alpha_{\gamma\zeta,wuv}, \beta_{\gamma\zeta,wuv}$ and $\rho_{\gamma,uv}, \pi_{\gamma,uv}$ where $\gamma \in [4], \zeta \in \{0, 1\}$ must be different from zero. By applying the Fatou's lemma, we get

$$0 = \lim_{n\to\infty} \frac{1}{m_n} \frac{2\mathbb{E}_X[d_V(p_{G_n}(\cdot|X), p_{G_*}(\cdot|X))]}{D_2(G_n, G_{*,n})} \geq \int \liminf_{n\to\infty} \frac{1}{m_n} \frac{|p_{G_n}(Y|X) - p_{G_{*,n}}(Y|X)|}{D_2(G_n, G_{*,n})} d(X, Y).$$

On the other hand,

$$\frac{1}{m_n} \frac{p_{G_n}(Y|X) - p_{G_{*,n}}(Y|X)}{D_2(G_n, G_{*,n})}$$

$$\to \sum_{\gamma=0}^{4} \left[ \sum_{\zeta=0}^{1} E_{\gamma\zeta}(X) X^\zeta + K_\gamma(X) \exp(\beta^\top X) \right] \frac{\partial^\gamma f}{\partial h^\gamma}(Y|h(X, \eta_0), \nu_0) \cdot \exp(\beta^\top X),$$

where

$$E_{11}(X) = \sum_{1 \leq w \leq d, 1 \leq u \leq q} \alpha_{11,wu0} \frac{\partial h}{\partial \eta^{(u)}}(X, \eta_n^*)$$

$$E_{21}(X) = \frac{1}{2} \sum_{1 \leq w \leq d} \alpha_{21,w00}$$

$$E_{10}(X) = \sum_{u=1}^{q} \alpha_{10,0u0} \frac{\partial h}{\partial \eta^{(u)}}(X, \eta_0) + \sum_{1 \leq u,v \leq q} \frac{\beta_{10,0uv}}{1 + \mathbf{1}_{u=v}} \frac{\partial^2 h}{\partial \eta^{(u)} \partial \eta^{(v)}}(X, \eta_0),$$

$$E_{20}(X) = \frac{1}{2} \alpha_{20,000} + \sum_{1 \leq u,v \leq q} \frac{\beta_{20,0uv}}{1 + \mathbf{1}_{u=v}} \frac{\partial h}{\partial \eta^{(u)}}(X, \eta_0) \frac{\partial h}{\partial \eta^{(v)}}(X, \eta_0),$$

$$E_{30}(X) = \frac{1}{2} \sum_{u=1}^{q} \beta_{30,0u0} \frac{\partial h}{\partial \eta^{(u)}}(X, \eta_0),$$

$$E_{40}(X) = \frac{1}{8}\beta_{40,000}.$$

and

$$K_1(X) = \sum_{u=1}^{q} \rho_{1,u0} \frac{\partial h}{\partial \eta^{(u)}}(X, \eta_0) + \sum_{1 \le u,v \le q} \frac{\pi_{1,uv}}{1 + \mathbf{1}_{u=v}} \frac{\partial^2 h}{\partial \eta^{(u)} \partial \eta^{(v)}}(X, \eta_0),$$

$$K_2(X) = \frac{1}{2}\rho_{2,00} + \sum_{1 \le u,v \le q} \frac{\pi_{2,uv}}{1 + \mathbf{1}_{u=v}} \frac{\partial h}{\partial \eta^{(u)}}(X, \eta_0) \frac{\partial h}{\partial \eta^{(v)}}(X, \eta_0),$$

$$K_3(X) = \frac{1}{2}\sum_{u=1}^{q} \pi_{3,u0} \frac{\partial h}{\partial \eta^{(u)}}(X, \eta_0),$$

$$K_4(X) = \frac{1}{8}\pi_{4,00}.$$

It is worth noting that for almost surely $(X, Y)$, the set $\mathcal{L} \cup \mathcal{K}$ is linearly independent under non-distinguishable setting , which leads to the fact that $E_{\tau\zeta}(X) = K_\tau(X) = 0$ for almost surely $X$ for any $\tau \in [4], \zeta \in \{0, 1\}$.

Similar to the proof of Theorem 1, and recalling that the experts are strongly identifiable, we conclude that all the coefficients in Equation (51) must be zero for all $w, u, v$.

This contradicts the fact that not all coefficients vanish. Thus, we obtain the conclusion for this case.

**Case 2:**

In this case, we consider that $(\eta_n, \nu_n)$ and $(\eta_n^*, \nu_n^*)$ share the same limit, but different from $(\eta_0, \nu_0)$.

From the formulation of the metric $D_1$ in the proof A.1, it is clear that $D_2 \lesssim D_1$. Therefore, we get $W_n(X, Y)/D_1(G_n, G_{*,n}) \to 0$ as $n \to \infty$. Noting that $(\eta_n, \nu_n)$ and $(\eta_n^*, \nu_n^*)$ share the limit $(\eta^*, \nu^*) \ne (\eta_0, \nu_0)$, we have $f_0 = f(Y|h(X, \eta_0), \nu_0)$ and $f(Y|h(X, \eta^*), \nu^*)$ satisfying $f_0$ and $f$ independent up to second order as in Lemma 1. Thus, we can process in a similar way as in Theorem 1 to draw a contradiction.

**Case 3:**

Lastly, we consider that one of $G_n$ or $G_n^*$ converges to $G_0$, while the other converges to $G' \ne G_0$. Without loss of generality, suppose that $G_n \to G'$ and $G_n^* \to G_0$. By passing through the limit for

$$\mathbb{E}_X[h_V(p_{G_n}(\cdot|X), p_{G_{*,n}}(\cdot|X))]/D_2(G_n, G_n^*) \to 0,$$

noting that

$$D_2(G_n, G_n^*) \to D_2(G, G_*) \ne 0, \mathbb{E}_X[h_V(p_{G_n}(\cdot|X), p_{G_{*,n}}(\cdot|X))] \to \mathbb{E}_X[h_V(p_G(\cdot|X), p_{G_*}(\cdot|X))],$$

we have

$$\mathbb{E}_X[h_V(p_G(\cdot|X), p_{G_*}(\cdot|X))] = 0, \text{ or } p_G = p_{G_*}, \text{ a.s.}$$

This equation implies that

$$f(Y|h(X, \eta_0), \nu_0) = \frac{1}{1 + \exp(\beta^\top X + \tau^*)} f(Y|h(X, \eta_0), \nu_0) + \frac{\exp(\beta^\top X + \tau^*)}{1 + \exp(\beta^\top X + \tau^*)} f(Y|h(X, \eta), \nu)$$

which further implies that

$$\frac{\exp(\beta^\top X + \tau^*)}{1 + \exp(\beta^\top X + \tau^*)} f(Y|h(X, \eta_0), \nu_0) = \frac{\exp(\beta^\top X + \tau^*)}{1 + \exp(\beta^\top X + \tau^*)} f(Y|h(X, \eta), \nu)$$

and hence

$$f(Y|h(X, \eta_0), \nu_0) = f(Y|h(X, \eta), \nu) \quad (\text{as } \exp(\beta^\top X + \tau^*) \ne 0).$$

This equation means that $G' = G_0$, which is a contradiction. $\qquad\square$

### A.4 Proof of Theorem 4

In what follows, we present the proof of Theorem 4 for the non-distinguishable setting.

*Proof of Theorem 4.* The proof follows similar steps to the arguments in the previous two sections. Concretely, define for $S_1 = (\tau_1, \beta_1, \eta_1, \nu_1)$, $S_2 = (\tau_2, \beta_2, \eta_2, \nu_2)$ :

$$\begin{cases} d_I(S_1, S_2) = \|\Delta\eta_1, \Delta\nu_1\|^2 |\exp(\tau_1) - \exp(\tau_2)|, \\ d_{II}(S_1, S_2) = \exp(\tau_1)\|\Delta\eta_1, \Delta\nu_1\| \|(\beta_1, \eta_1, \nu_1) - (\beta_2, \eta_2, \nu_2)\|. \end{cases}$$

It is straightforward that $d_I$ and $d_{II}$ satisfy the weak triangle inequality. Following the same schema as in Lemma 2, we can demonstrate two subsequent results for any $r > 1$:

(i) Two sequences can be found

$$\begin{cases} S_{1,n} = (\tau_{1,n}, \beta_n, \eta_n, \nu_n) \in \Xi(l_n), \\ S_{2,n} = (\tau_{1,n}, \beta_n, \eta_n, \nu_n) \in \Xi(l_n), \end{cases}$$

such that $d_I(S_{1,n}, S_{2,n}) \to 0$ and $\mathbb{E}_X[h_H(p_{S_{1,n}}(\cdot|X), p_{S_{2,n}}(\cdot|X))]/d_I^r(S_{1,n}, S_{2,n}) \to 0$ as $n \to \infty$.

(ii) Two sequences can be found

$$\begin{cases} S_{1,n} = (\tau_n, \beta_{1,n}, \eta_{1,n}, \nu_{1,n}) \in \Xi(l_n), \\ S_{2,n} = (\tau_n, \beta_{2,n}, \eta_{2,n}, \nu_{2,n}) \in \Xi(l_n), \end{cases}$$

such that $d_{II}(S_{1,n}, S_{2,n}) \to 0$ and $\mathbb{E}_X[h_H(p_{S_{1,n}}(\cdot|X), p_{S_{2,n}}(\cdot|X))]/d_{II}^r(S_{1,n}, S_{2,n}) \to 0$ as $n \to \infty$.

We can omit the justification for the above results as it can follow a similar approach as in Lemma 2. This leads to the conclusion of the theorem. $\square$

## B Proof of Auxiliary Results

### B.1 Proof of Proposition 1

*Proof.* Fix an arbitrary $x \in \mathcal{X}$ and abbreviate

$$g_1(y) := f\big(y|h(x, \eta_1), \nu_1\big), \qquad g_2(y) := f\big(y|h(x, \eta_2), \nu_2\big), \qquad g_0(y) := f_0\big(y|h_0(x, \eta_0), \nu_0\big).$$

Because $f$ is Gaussian in its argument, there exist $\mu_1, \mu_2 \in \mathbb{R}$ and $\sigma_1^2, \sigma_2^2 > 0$ such that $g_j(y) = \dfrac{1}{\sqrt{2\pi\sigma_j^2}} \exp\big(-(y - \mu_j)^2/(2\sigma_j^2)\big)$ for $j = 1, 2$.

Set

$$H_1(y) := \frac{\partial g_2}{\partial h}(y) = \frac{y - \mu_2}{\sigma_2^2} g_2(y), \qquad H_2(y) := \frac{\partial^2 g_2}{\partial h^2}(y) = \frac{(y - \mu_2)^2 - \sigma_2^2}{\sigma_2^4} g_2(y).$$

With these notations the assumed identity becomes

$$b_0(x)g_0(y) + b_1(x)g_1(y) + c_0(x)g_2(y) + c_1(x)H_1(y) + \tfrac{1}{2}c_2(x)H_2(y) = 0 \quad \text{for a.e. } y \in \mathbb{R}. \quad (52)$$

**1. $b_0(x) = 0$.** Because $g_0$ is *not* Gaussian by assumption, while $g_1, g_2, H_1, H_2$ all belong to the finite–dimensional linear span $\mathcal{G} := \text{span}\{y \mapsto g_1(y), y \mapsto (y - \mu_2)^k g_2(y) : k = 0, 1, 2\}$, we have $g_0 \notin \mathcal{G}$. Hence the only way (52) can hold on a set of positive measure is with $b_0(x) = 0$.

**2. Linear independence inside $\mathcal{G}$.** Divide (52) (now with $b_0(x) = 0$) by $g_2(y)$; we obtain the polynomial identity

$$b_1(x)\frac{g_1(y)}{g_2(y)} + c_0(x) + c_1(x)\frac{y - \mu_2}{\sigma_2^2} + \tfrac{1}{2}c_2(x)\frac{(y - \mu_2)^2 - \sigma_2^2}{\sigma_2^4} = 0 \quad \text{for a.e. } y.$$

The ratio $g_1/g_2$ is the analytic (non-polynomial) function

$$\frac{g_1(y)}{g_2(y)} = K \exp\Big(\tfrac{1}{2}\big[(y - \mu_2)^2/\sigma_2^2 - (y - \mu_1)^2/\sigma_1^2\big]\Big),$$

with $K \neq 0$. Since $\mu_1 \neq \mu_2$ or $\sigma_1^2 \neq \sigma_2^2$, this exponential term cannot be expressed as a quadratic polynomial in $y$. Consequently the set of functions $\big\{g_1/g_2, 1, y - \mu_2, (y - \mu_2)^2\big\}$ is linearly independent on any interval. Hence every coefficient in the polynomial identity must vanish:

$$b_1(x) = c_0(x) = c_1(x) = c_2(x) = 0.$$

**3. Conclusion.** We have shown that $b_0(x) = b_1(x) = c_0(x) = c_1(x) = c_2(x) = 0$ for the fixed $x$. Because the same argument works for almost every $x \in \mathcal{X}$, all coefficients vanish almost surely. Thus the unified distinguishability condition of Definition 1 is satisfied, completing the proof. □

### B.2   Proof of Proposition 2

*Proof.* Write the two (single–expert) conditional densities

$$p_G(y|x) = \big[1 - \lambda(x)\big]f_0\big(y|h_0(x, \eta_0), \nu_0\big) + \lambda(x)f\big(y|h(x, \eta), \nu\big),$$
$$p_{G'}(y|x) = \big[1 - \lambda'(x)\big]f_0\big(y|h_0(x, \eta_0), \nu_0\big) + \lambda'(x)f\big(y|h(x, \eta'), \nu'\big),$$

where $\lambda(x) := \frac{\exp\big(\beta^\top x + \tau\big)}{1 + \exp\big(\beta^\top x + \tau\big)}$ and $\lambda'(x) := \frac{\exp\big(\beta'^\top x + \tau'\big)}{1 + \exp\big(\beta'^\top x + \tau'\big)}$.

Assume the identifiability equality $p_G(y|x) = p_{G'}(y|x)$ holds for almost every $(x, y) \in \mathcal{X} \times \mathcal{Y}$. Subtracting the two representations gives

$$\big[\lambda(x) - \lambda'(x)\big]f_0\big(y|h_0(x, \eta_0), \nu_0\big) + \lambda'(x)f\big(y|h(x, \eta'), \nu'\big) - \lambda(x)f\big(y|h(x, \eta), \nu\big) = 0. \quad (53)$$

**Step 1. If $\lambda(x) \neq \lambda'(x)$.** Suppose on a set of positive $x$-measure, $\lambda(x) \neq \lambda'(x)$. Divide (53) by $\lambda(x) - \lambda'(x)$; then for those $x$

$$f_0\big(y|h_0, \nu_0\big) + b(x)f\big(y|h(x, \eta'), \nu'\big) + c(x)f\big(y|h(x, \eta), \nu\big) = 0,$$

where

$$b(x) := \frac{\lambda'(x)}{\lambda'(x) - \lambda(x)}, c(x) := \frac{-\lambda(x)}{\lambda'(x) - \lambda(x)}.$$

Since $f$ is distinguishable from $f_0$, the only possibility is $b(x) = c(x) = 0$, hence $\lambda(x) = \lambda'(x)$ a.e.—contradiction. Therefore

$$\lambda(x) = \lambda'(x) \quad \text{for a.e. } x.$$

Because the soft-max map $(\beta, \tau) \mapsto \lambda(\cdot)$ is injective, we conclude

$$\beta = \beta', \qquad \tau = \tau'.$$

**Step 2. Equality of expert parameters.** With $\lambda(x) = \lambda'(x)$, equation (53) reduces to

$$f\big(y|h(x, \eta), \nu\big) = f\big(y|h(x, \eta'), \nu'\big) \qquad \text{for a.e. } (x, y).$$

Definition 1 forces the situation $(\eta, \nu) \neq (\eta', \nu')$ impossible. Hence the only consistent solution is

$$(\eta, \nu) = (\eta', \nu').$$

**Step 3. Conclusion.** We have shown $\beta = \beta'$, $\tau = \tau'$, $\eta = \eta'$, and $\nu = \nu'$; hence $G = G'$. □

## B.3 Proof of Proposition 3

We begin by introducing several standard notations used throughout this proof. Let $(\mathcal{P}, d)$ be a metric space, where $d$ is a metric on $\mathcal{P}$. An $\epsilon$-net of $(\mathcal{P}, d)$ is a collection of balls of radius $\epsilon$ whose union covers $\mathcal{P}$. The *covering number* $N(\epsilon, \mathcal{P}, d)$ denotes the minimal cardinality of such a covering, and the *entropy number* is defined as $H(\epsilon, \mathcal{P}, d) := \log N(\epsilon, \mathcal{P}, d)$.

The *bracketing number* $N_B(\epsilon, \mathcal{P}, d)$ is the minimal number of pairs $\{(\underline{f}_i, \overline{f}_i)\}_{i=1}^n$ such that $\underline{f}_i < \overline{f}_i$, $d(\underline{f}_i, \overline{f}_i) < \epsilon$, and $\mathcal{P}$ is covered by the union of the brackets. The corresponding *bracketing entropy* is denoted by $H_B(\epsilon, \mathcal{P}, d) := \log N_B(\epsilon, \mathcal{P}, d)$.

When $\mathcal{P}$ is a family of densities, we take $d$ to be the $L^2(m)$ distance, where $m$ denotes the Lebesgue measure.

In particular, let $\mathcal{P}(\Xi) := \{p_\lambda : \lambda \in \Xi\}$, and define the symmetrized density $\bar{p}_\lambda := \frac{1}{2}(p^* + p_\lambda)$, where $p^*$ denotes the true density. We then define the following sets: $\overline{\mathcal{P}}(\Xi) := \{\bar{p}_\lambda : \lambda \in \Xi\}$ and $\overline{\mathcal{P}}^{1/2}(\Xi) := \{\bar{p}_\lambda^{1/2} : \bar{p}_\lambda \in \overline{\mathcal{P}}(\Xi)\}$. To study convergence rates, we consider the localized version of the symmetrized class: $\overline{\mathcal{P}}^{1/2}(\Xi, \epsilon) := \{\bar{p}_\lambda^{1/2} \in \overline{\mathcal{P}}^{1/2}(\Xi) : d_H(\bar{p}_\lambda, p^*) \le \epsilon\}$, where $d_H(\cdot, \cdot)$ denotes the Hellinger distance. Then we assess the complexity of this class via the *bracketing entropy integral* defined in [34]: $\mathcal{J}_B(\epsilon, \overline{\mathcal{P}}^{1/2}(\Xi, \epsilon), m) := \int_{\epsilon^2/2^{13}}^{\epsilon} \sqrt{H_B(u, \overline{\mathcal{P}}^{1/2}(\Xi, \epsilon), m)} du \vee \epsilon$, where $a \vee b := \max\{a, b\}$. For brevity, we may omit the dependence on $m$ when it is clear from context.

For the proof at first we consider a general lemma that provides the desired convergence rate, provided that a bracketing entropy condition is satisfied.

**Lemma 3.** *Assume the following assumption hold: Given a universal constant $J > 0$, there exists $N > 0$, possibly depending on $\Xi$, such that for all $n \ge N$ and all $\epsilon > (\log(n)/n)^{1/2}$, we have*

$$\mathcal{J}_B(\epsilon, \overline{P}^{1/2}(\Xi, \epsilon)) \le J\sqrt{n}\epsilon^2. \tag{54}$$

*Then, there exists a constant $C > 0$ depending only on $\Xi$ such that for all $n \ge 1$,*

$$\sup_{G_* \in \Xi} \mathbb{E}_{p_{G_*}, n} \mathbb{E}_X[d_H(p_{\widehat{G}_n}(\cdot|X), p_{G_*}(\cdot|X))] \le C\sqrt{\log n/n}.$$

This lemma indicates that it suffices to verify the entropy condition in Equation (54) in order to obtain the convergence rate. However, this condition is often technically difficult to establish directly. As a workaround, we may instead prove the following sufficient condition:

**Lemma 4.** *If the distribution satisfies*

$$H_B(\epsilon, \mathcal{P}(\Xi), d_H) \lesssim \log(1/\epsilon), \tag{55}$$

*it will meet the assumption in Equation (54).*

Although we have simplified the condition in Equation (54) to Equation (55), verifying Equation (55) is still nontrivial. Fortunately, for the contaminated model defined in Equation (1),

$$p_G(Y|X) := \frac{1}{1 + \exp(\beta^\top X + \tau)} \cdot f_0(Y|h_0(X, \eta_0), \nu_0) + \frac{\exp(\beta^\top X + \tau)}{1 + \exp(\beta^\top X + \tau)} \cdot f(Y|h(X, \eta), \nu),$$

we assume that $f_0$ is bounded with light tails and that $f$ is a univariate Gaussian density. Under these assumptions, we can verify Equation (55) via the following lemma:

**Lemma 5.** *Let $\Gamma$ be a compact subsets of $\mathbb{R}^d \times \mathbb{R}$ and $\Theta$ be a bounded subsets of $\mathbb{R}^q \times \mathbb{R}^+$, $f$ is a univariate Gaussian density and $f_0$ is bounded with tail $\mathbb{E}_X\left(-\log f_0(Y|h(X, \eta_0), \nu_0)\right) \gtrsim Y^q$ for almost surely $Y \in \mathcal{Y}$ for some $q > 0$. Then, for any $0 < \varepsilon < \frac{1}{2}$, the following results hold:*

*(i) $\log N(\epsilon, \mathcal{P}(\Xi), \|\cdot\|_\infty) \lesssim \log(1/\epsilon)$,*

*(ii) $H_B(\epsilon, \mathcal{P}(\Xi), d_H) \lesssim \log(1/\epsilon)$.*

Combining the above results, we obtain the desired conclusion for Theorem 3.

Now we will prove Lemma 3, Lemma 4 and Lemma 5 in order. At first we need to introduce another Lemma 6 before we prove Lemma 3. Lemma 6 is Theorem 5.11 in [34] and its proof can also be found in [34].

**Lemma 6.** *Let $R > 0$, $k \geq 1$ and $\mathcal{G}$ is a subset in $\Xi$ where $G_* \in \mathcal{G} \subset \Xi$. Given $C_1 < \infty$, for all $C$ sufficiently large, and for $n \in \mathbb{N}$ and $t > 0$ is in the following range*

$$t \leq (8\sqrt{n}R) \wedge (C_1\sqrt{n}R^2/K), \tag{56}$$

$$t \geq C^2(C_1 + 1)\left(R \vee \int_{t/(2^6\sqrt{n})}^{R} H_B^{1/2}\left(\frac{u}{\sqrt{2}}, \overline{\mathcal{P}}^{1/2}(\Xi, R), m\right) du\right), \tag{57}$$

*then we will have*

$$\mathbb{P}_{G_*, n}\left(\sup_{G \in \mathcal{G}, \mathbb{E}_X[h(\bar{p}_G(\cdot|X), p_{G_*}(\cdot|X))] \leq R} |\mu_n(G)| \geq t\right) \leq C\exp\left(-\frac{t^2}{C^2(C_1 + 1)R^2}\right). \tag{58}$$

*Proof of Lemma 3.* Firstly, by Lemma 4.1 and 4.2 in [34], we have

$$\frac{1}{16}\mathbb{E}_X[d_H^2(p_{\widehat{G}_n}(\cdot|X), p_{G_*}(\cdot|X))] \leq \mathbb{E}_X[d_H^2(\bar{p}_{\widehat{G}_n}(\cdot|X), p_{G_*}(\cdot|X))] \leq \frac{1}{\sqrt{n}}\mu_n(\widehat{G}_n),$$

here $\mu_n(\widehat{G}_n)$ is an empirical process defined as

$$\mu_n(\widehat{G}_n) := \sqrt{n}\int_{p_{G_*}>0} \frac{1}{2}\log\left(\frac{\bar{p}_{\widehat{G}_n}}{p_{G_*}}\right)(\bar{p}_{\widehat{G}_n} - p_{G_*})d(X, Y).$$

Thus, for any $\delta > \delta_n := \sqrt{\log n/n}$, we have

$$\mathbb{P}_{G_*, n}(\mathbb{E}_X[d_H(p_{\widehat{G}_n}(\cdot|X), p_{G_*}(\cdot|X))] \geq \delta)$$

$$\leq \mathbb{P}_{G_*, n}\left(\mu_n(\widehat{G}_n) - \sqrt{n}\mathbb{E}_X[d_H^2(p_{\widehat{G}_n}(\cdot|X), p_{G_*}(\cdot|X))] \geq 0, \mathbb{E}_X[d_H(p_{\widehat{G}_n}(\cdot|X), p_{G_*}(\cdot|X))] \geq \frac{\delta}{4}\right)$$

$$\leq \mathbb{P}_{G_*, n}\left(\sup_{G: \mathbb{E}_X[d_H(\bar{p}_G(\cdot|X), p_{G_*}(\cdot|X))] \geq \delta/4} \left[\mu_n(G) - \sqrt{n}\mathbb{E}_X[d_H^2(\bar{p}_G(\cdot|X), p_{G_*}(\cdot|X))]\right] \geq 0\right)$$

$$\leq \sum_{s=0}^{S} \mathbb{P}_{G_*, n}\left(\sup_{G: 2^s\delta/4 \leq \mathbb{E}_X[d_H(\bar{p}_G(\cdot|X), p_{G_*}(\cdot|X))] \leq 2^{s+1}\delta/4} |\mu_n(G)| \geq \sqrt{n}2^{2s}(\frac{\delta}{4})^2\right)$$

$$\leq \sum_{s=0}^{S} \mathbb{P}_{G_*, n}\left(\sup_{G: \mathbb{E}_X[d_H(\bar{p}_G(\cdot|X), p_{G_*}(\cdot|X))] \leq 2^{s+1}\delta/4} |\mu_n(G)| \geq \sqrt{n}2^{2s}(\frac{\delta}{4})^2\right)$$

where $S$ is a smallest number such that $2^S\delta/4 > 1$.

Now we will use Lemma 6: choose $R = 2^{s+1}\delta$, $C_1 = 15$ and $t = \sqrt{n}2^{2s}(\delta/4)^2$. We can confirm that condition (i) in Lemma 3 is met since $2^{s-1}\delta/4 \leq 1$ for all $s \leq S$. For the condition (ii), it is still satisfied since

$$\int_{t/2^6\sqrt{n}}^{R} H_B^{1/2}\left(\frac{u}{\sqrt{2}}, \mathcal{P}^{1/2}(\Xi, R), \mu\right) du \vee 2^{s+1}\delta$$

$$= \sqrt{2}\int_{R^2/2^{13}}^{R/\sqrt{2}} H_B^{1/2}\left(u, \mathcal{P}^{1/2}(\Xi, R), \mu\right) du \vee 2^{s+1}\delta$$

$$\leq 2\mathcal{J}_B\left(R, \mathcal{P}^{1/2}(\Xi, R), \mu\right)$$

$$\leq 2J\sqrt{n}2^{2s+1}\delta^2$$

$$= 2^6 Jt.$$

Now since the two conditions in Lemma 6 are all satisfied, we could conclude that

$$\mathbb{P}_{G_*, n}\left(\mathbb{E}_X[d_H(p_{\widehat{G}_n}(\cdot|X), p_{G_*}(\cdot|X))] > \delta\right) \leq C\sum_{s=0}^{\infty} \exp\left(-\frac{2^{2s}n\delta^2}{2^{14}C^2}\right) \leq c\exp\left(-\frac{n\delta^2}{c}\right), \tag{59}$$

here constant $c$ is a large constant that does not depend on $G_*$. Now we could derive the bound on supremum of expectation:

$$\mathbb{E}_{p_{G_*,n}} \mathbb{E}_X [d_H(p_{\widehat{G}_n}(\cdot|X), p_{G_*}(\cdot|X))] = \int_0^\infty \mathbb{P}\left(\mathbb{E}_X[d_H(p_{\widehat{G}_n}(\cdot|X), p_{G_*}(\cdot|X))] > \delta\right) d\delta$$
$$\leq \delta_n + c \int_{\delta_n}^\infty \exp\left(-\frac{n\delta^2}{c^2}\right) d\delta$$
$$\leq \tilde{c}\delta_n,$$

here $\tilde{c}$ is independent from $G_*$ and $\delta_n := \sqrt{\log n/n}$. So we can conclude that

$$\sup_{G_* \in \Xi} \mathbb{E}_{p_{G_*,n}} \mathbb{E}_X[d_H(p_{\widehat{G}_n}(\cdot|X), p_{G_*}(\cdot|X))] \leq C\sqrt{\log n/n}.$$

$\square$

*Proof of Lemma 4.* Because $\overline{\mathcal{P}}^{1/2}(\Xi, \delta) \subset \overline{\mathcal{P}}^{1/2}(\Xi)$ and from the definition of Hellinger distance, we have

$$H_B(\delta, \overline{\mathcal{P}}^{1/2}(\Xi, \delta), \mu) \leq H_B(\delta, \overline{\mathcal{P}}^{1/2}(\Xi), \mu) = H_B\left(\frac{\delta}{\sqrt{2}}, \overline{\mathcal{P}}(\Xi), h\right).$$

Now, using the fact that for densities $f^*, f_1, f_2$, we have $h^2\left(\frac{f_1+f^*}{2}, \frac{f_2+f^*}{2}\right) \leq \frac{h^2(f_1,f_2)}{2}$, it is easy to verify that $H_B(\delta/\sqrt{2}, \overline{\mathcal{P}}(\Xi), d_H) \leq H_B(\delta, \mathcal{P}(\Xi), d_H)$. Hence, if equation (55) holds true, then

$$H_B(\delta, \overline{\mathcal{P}}^{1/2}(\Xi, \delta), \mu) \leq H_B(\delta, \mathcal{P}(\Xi), d_H) \lesssim \log\left(\frac{1}{\delta}\right).$$

This implies that

$$\mathcal{J}_B\left(\epsilon, \overline{\mathcal{P}}^{1/2}(\Xi, \delta), \mu\right) \lesssim \epsilon \left(\log(\frac{2^{13}}{\epsilon^2})\right)^{\frac{1}{2}} < n\epsilon^2, \quad \text{for all } d\epsilon > \sqrt{\frac{\log n}{n}}.$$

$\square$

*Proof of Lemma 5.* **Proof for (i):** Let $\mathcal{E}_\epsilon(S)$ denote an $\epsilon$-net of a set $S$ under the $\|\cdot\|_\infty$ norm. Then

$$\log|\mathcal{E}_\epsilon(S)| = \log N(\epsilon, S, \|\cdot\|_\infty).$$

Let $\mathcal{P}(\Theta) := \{p_\Upsilon : \Upsilon \in \Theta\}$, where $p_\Upsilon(Y|X) := f(Y|h(X, \eta), \nu)$. By Lemma 6 in [13], we have

$$\log N(\epsilon, \mathcal{P}(\Theta), \|\cdot\|_\infty) \lesssim \log(1/\epsilon).$$

We now consider the contaminated model $p_\Upsilon$ as a composition of smooth components indexed by $(\beta, \tau, \eta, \nu) \in \Xi := \Gamma \times \Theta$, where $\Gamma \subset \mathbb{R}^{d+1}$ and $\Theta \subset \mathbb{R}^q \times \mathbb{R}^+$ are compact.

Since $\sigma(\beta^\top X + \tau) := \exp(\beta^\top X + \tau)/(1 + \exp(\beta^\top X + \tau))$ is infinitely differentiable and Lipschitz over compact $\Gamma$, it follows that for any $\lambda = (\beta, \tau) \in \Gamma$, there exists $\widetilde{\lambda} = (\widetilde{\beta}, \widetilde{\tau}) \in \mathcal{E}_\epsilon(\Gamma)$ such that

$$\|\sigma_\lambda - \sigma_{\widetilde{\lambda}}\|_\infty := \sup_{X \in \mathcal{X}} \left|\frac{\exp(\beta^\top X + \tau)}{1 + \exp(\beta^\top X + \tau)} - \frac{\exp(\widetilde{\beta}^\top X + \widetilde{\tau})}{1 + \exp(\widetilde{\beta}^\top X + \widetilde{\tau})}\right| \leq \epsilon.$$

Likewise, for any $\Upsilon = (\eta, \nu) \in \Theta$, there exists $\widetilde{\Upsilon} \in \mathcal{E}_\epsilon(\Theta)$ such that

$$\|p_\Upsilon - p_{\widetilde{\Upsilon}}\|_\infty \leq \epsilon.$$

Now, consider the difference

$$p_G(Y|X) - p_{\widetilde{G}}(Y|X)$$
$$= (\sigma_\lambda(X) - \sigma_{\widetilde{\lambda}}(X))[f(Y|h(X, \eta), \nu) - f_0(Y|h_0(X, \eta_0), \nu_0)]$$
$$+ \sigma_{\widetilde{\lambda}}(X)[f(Y|h(X, \eta), \nu) - f(Y|h(X, \widetilde{\eta}), \widetilde{\nu})],$$

so that by the triangle inequality and boundedness of $f_0$ and $f$,

$$\|p_G - p_{\widetilde{G}}\|_\infty \leq \|\sigma_\lambda - \sigma_{\widetilde{\lambda}}\|_\infty \cdot (\|f_0\|_\infty + \|f\|_\infty) + \|\sigma_{\widetilde{\lambda}}\|_\infty \cdot \|p_\Upsilon - p_{\widetilde{\Upsilon}}\|_\infty$$
$$\lesssim \epsilon.$$

Hence, the covering number of $\mathcal{P}(\Xi)$ satisfies

$$\log N(\epsilon, \mathcal{P}(\Xi), \|\cdot\|_\infty) \leq \log N(\epsilon, \Gamma, \|\cdot\|_\infty) + \log N(\epsilon, \mathcal{P}(\Theta), \|\cdot\|_\infty) \lesssim \log(1/\epsilon).$$

**Proof for (ii):** First, let $\eta \leq \varepsilon$ be a positive number, which will be chosen later. We consider $f$ is the density function of an univariate Gaussian distribution, so $f$ is light tail: for any $|Y| \geq 2a$ and $X \in \mathcal{X}$,

$$f(Y|h(X, \eta), \nu) \leq \frac{1}{\sqrt{2\pi}\ell} \exp\left(-\frac{Y^2}{8u^2}\right).$$

Also $f_0$ is bounded with tail $\log f_0(Y|h(X, \eta_0), \nu_0) \lesssim -Y^q$ and $f_0(Y|h(X, \eta_0)), \nu_0) \leq M$ for almost surely $Y \in \mathcal{Y}$ for some $M, q > 0$. Now let $q = \min\{p, 2\}$ and $C_2 = \max\left\{M, 1/\sqrt{2\pi}\ell\right\}$, we will have

$$H(X, Y) = \begin{cases} C_1 \exp(-Y^q), & |Y| \geq 2a \\ C_2, & |Y| < 2a \end{cases} \tag{60}$$

here $C_1$ is a positive constant depending on $\ell$ and $f_0$. Moreover $H(X, Y)$ is an envelope of $\mathcal{P}(\Xi)$. Next, let $g_1, \ldots, g_N$ represent an $\eta$-net over $\mathcal{P}_k(\Xi)$. Then, we construct the brackets $[p_i^L(X, Y), p_i^U(X, Y)]$ as follows:

$$\begin{cases} p_i^L(X, Y) := \max\{g_i(X, Y) - \eta, 0\} \\ p_i^U(X, Y) := \min\{g_i(X, Y) + \eta, H(X, Y)\} \end{cases}$$

for $i = 1, \cdots, N$. As a result, $\mathcal{P}_k(\Xi) \subset \bigcup_{i=1}^N [p_i^L(X, Y), p_i^U(X, Y)]$ and $p_i^U(X, Y) - p_i^L(X, Y) \leq \min\{2\eta, H(X, Y)\}$. Consequently,

$$\int \left(p_i^U(X, Y) - p_i^L(X, Y)\right) d(X, Y)$$
$$\leq \int_{|Y|<2a} \left(p_i^U(X, Y) - p_i^L(X, Y)\right) d(X, Y) + \int_{|Y|\geq 2a} \left(p_i^U(X, Y) - p_i^L(X, Y)\right) d(X, Y)$$
$$\leq \int_{|Y|<2a} 2\eta \, d(X, Y) + \int_{|Y|\geq 2a} H(X, Y) d(X, Y) \lesssim \eta.$$

This shows that

$$H_B(c\eta, \mathcal{P}(\Xi), \|\cdot\|_1) \leq N \lesssim \log(1/\eta).$$

Setting $\eta = \epsilon/c$, we find

$$H_B(\epsilon, \mathcal{P}(\Xi), \|\cdot\|_1) \lesssim \log(1/\epsilon).$$

Since $h^2 \leq \|\cdot\|_1$ holds between the Hellinger distance and the total variation distance, we conclude the bracketing entropy bound. $\qquad\square$

