# OpenReview forum: "On Minimax Estimation of Parameters in Softmax-Contaminated Mixture of Experts"
_NeurIPS.cc/2025/Conference — NeurIPS 2025 poster_

### Official Review · Reviewer_kSC9 · 2025-06-30

**Clarity:** 3
**Significance:** 4
**Originality:** 4
**Rating:** 5
**Confidence:** 3

**Summary:**

This paper provides a theoretical study of a softmax-contaminated mixture of experts model, which corresponds to a parameter-efficient fine-tuning scenario where a large pre-trained expert is kept fixed and a new prompt expert is learned alongside a softmax gating network. The authors analyze the convergence rates of the maximum likelihood estimator (MLE) for the gating parameters and the prompt’s parameters.
it introduces distinguishability condition that formalizes when the prompt’s learned knowledge does not excessively overlap with the pre-trained model. when the prompt is non-distinguishable (i.e. it effectively learns the same function as the pre-trained model), the authors show the estimation problem becomes much harder.

**Questions:**

The definition of distinguishability (Definition 1) is quite involved. Could the authors provide more intuition or a simpler characterization for this condition?

how do these findings manifest in real fine-tuning tasks? For instance, if one fine-tunes a large language model with a small prompt on a new task that is actually already solvable by the base model, does one observe slower convergence or higher data requirements (as the theory suggests)? The current experiments are synthetic for control, but have the authors thought about testing their ideas on a toy real-world scenario (e.g., fine-tuning a small neural network with a fixed part vs. a learned part)?

**Ethical Concerns:**

["NO or VERY MINOR ethics concerns only"]

**Final Justification:**

The answer is satisfactory. I keep my score.

**Limitations:**

Yes.  (1) the analysis is restricted to a single prompt expert, and (2) the prompt model family is limited (in this paper,

**Paper Formatting Concerns:**

-

**Quality:**

3

**Strengths And Weaknesses:**

Strengths:

This work provides the first rigorous characterization of parameter estimation rates in a softmax-gated fine-tuned mixture model, highlighting the critical role of the prompt’s distinguishability from the base model in achieving optimal rates.

The authors derive minimax lower bounds that match the MLE’s rates under both the distinguishable and non-distinguishable regimes

To the best of my knowledge, this is the first analysis of a softmax-gated mixture of experts with one frozen and one trainable expert (which aptly models modern prompt-based fine tuning). The results of this paper are quite significant for the theory of mixture-of-experts and for understanding fine-tuning with large models.

It extends prior work by handling a more realistic model (softmax gating)

The paper is well-structured trying to explain heavy concepts clearly.

Weaknesses:

The theory’s assumptions, such as strong identifiability and the need to restrict the parameter space in the non-identifiable regime, may limit applicability to certain architectures or require careful interpretation.

The experimental validation, while appreciated, is limited to synthetic data

Minor:
38 through-> thorough
118 challening

---

> ### Author Rebuttal · Authors · 2025-07-30
>
> Dear Reviewer kSC9,
>
> Thanks for your constructive and valuable feedback, and giving **excellent (4) grade** to the significance and originality and **good (3) grade** to the quality and clarity of our paper. We are encouraged by the endorsement that
>
> *(i) Our work provides **the first** rigorous characterization of parameter estimation rates in a softmax-gated;*
>
> *(ii) The results of this paper are quite **significant** for the theory of mixture-of-experts and for understanding fine-tuning with large models.*
>
> *(iii) Our work considers a **more realistic** model than previous works;*
>
> *(iv) The paper is **well-structured** trying to **explain heavy concepts clearly**.*
>
> Below are our responses to your questions. We will also add these changes to the revision of our manuscript. We hope that we have addressed your concerns regarding our paper.
>
> ***
>
> **Q1: The theory’s assumptions, such as strong identifiability and the need to restrict the parameter space in the non-identifiable regime, may limit applicability to certain architectures or require careful interpretation.**
>
> Thank you for your comment. As we noted in our article, any pre-trained model that does not belong to the Gaussian density family is distinguishable from the prompt model (Proposition 1). Moreover, many widely used functions—such as GELU, sigmoid, and tanh—satisfy the strong identifiability assumption (see Examples, line 233-238). Therefore, the assumptions made in our analysis do not significantly restrict its applicability in practice.
>
> ***
> **Q2: Could the authors provide more intuition or a simpler characterization for the distinguishability condition?**
>
> Thank you for your question. Let us provide an intuitive explanation of the concept of distinguishability. This notion captures the potential interaction between the pre-trained model and the prompt model, particularly in terms of their first-order derivatives. When a pair of pre-trained and prompt models satisfies the distinguishability condition, it ensures that such interactions are avoided. This separation is beneficial for parameter estimation, as it prevents interference between the components and leads to more accurate inference.
>
> ***
> **Q3: Experimental validation is limited to synthetic data.**
>
> Thanks for your comment. We have added a few-shot fine-tuning experiment on DINOv2 ViT-S embeddings of CIFAR-10 under a domain shift from vehicles to animals. In this setting, the softmax contaminated MoE consistently outperforms both the input-free contaminated MoE and standard fine-tuning. Please see the additional experiments for details.
>
> **Additional Experiments**
> We study adaptation of a pretrained classifier to a distributionally shifted dataset.
> Let $h_{0}$ be a model trained on a “source” subset biased toward *vehicles*.
> We then adapt the model on a small “target” subset biased toward *animals* and evaluate on the full, balanced CIFAR-10 test split.
>
> ---
>
> ### Setup
> We extract 384-dimensional embeddings from CIFAR-10 images using a DINOv2 “small” ViT.
> The ten classes are partitioned into two groups:
> * **Animals** — bird, cat, deer, dog, frog, horse (6 classes)
> * **Vehicles** — airplane, automobile, ship, truck (4 classes)
>
> ---
>
> ### Data construction (per-class counts)
>
> * **Distinguishable setting**: the prompt model does not share knowledge with the pre-trained model
>   * Pre-train: 524 samples exclusively from vehicle classes (131 per vehicle class)
>   * Fine-tune: 56 samples exclusively from animal classes (≈ 9–10 per animal class)
>
> * **Non-distinguishable setting**: the prompt model partly shares knowledge with the pre-trained model
>   * Pre-train: 512 vehicle (128 per class) + 12 animal (2 per class) = 524
>   * Fine-tune: 48 animal (8 per class) + 8 vehicle (2 per class) = 56
>
> The held-out test set is the full, balanced 10 000-image CIFAR-10 split.
>
> ---
>
> ### Implementation details
>
> | Setting      | Split       | Animals / class | Vehicles / class | Total |
> |-------------|-------------|-----------------|------------------|-------|
> | Distinguishable  | Pre-train   | 0               | 131              | 524   |
> |             | Fine-tune   | 9–10            | 0                | 56    |
> | Non-distinguishable | Pre-train   | 2               | 128              | 524   |
> |             | Fine-tune   | 8               | 2                | 56    |
>
> ---
>
> ### Training
>
> * **Pre-training** — train a linear classifier $h_{0}$ on the pre-training set with cross-entropy, Adam (lr $10^{-3}$, wd $10^{-4}$, batch 32) for 20 epochs, then freeze its parameters.
> * **Fine-tuning** — attach a contaminated MoE that mixes $h_{0}$ with a new trainable linear expert $h_{1}$ with gating parameters $(\beta,\tau)$
>
>   $$
>     \hat{y}(x)=\bigl(1-\sigma(\beta^{\top}x+\tau)\bigr)\,h_{0}(x)
>                +\sigma(\beta^{\top}x+\tau)\,h_{1}(x).
>   $$
>
>   Only the prompt and gating parameters are updated (10 epochs, same optimizer).
>
> *Optimizer:* Adam (lr $1\times10^{-3}$, wd $1\times10^{-4}$, batch 32)
> *Epochs:* 20 (pre-train), 10 (fine-tune)
>
> ---
>
> ### Evaluation and baselines
>
> We report accuracy on the full test set, as well as on the animal-only and vehicle-only subsets.
> Baselines: Single frozen pre-trained model, Fine-tuning pre-trained model (without prompt), Input-free contaminated MoE [2], Softmax-contaminated MoE (Ours).
>
> ---
>
> ### Results
>
> *Columns in each table are accuracies on the indicated test subset.*
>
> ### Table 1a – *Distinguishable* setting
>
> | Model | Animals | Vehicles | Overall |
> |-------|:-------:|:--------:|:-------:|
> | Single frozen pre-trained model             | 0.1 | **94.9** | 38.0 |
> |  Fine-tuning pre-trained model  | 69.4 | 91.1     | 78.1 |
> | Input-free contaminated MoE      | 52.6 | 87.9     | 69.1 |
> | **Softmax-contaminated MoE** | **81.7** | 94.1 | **86.7** |
>
> ### Table 1b – *Non-distinguishable* setting
>
> | Model | Animals | Vehicles | Overall |
> |-------|:-------:|:--------:|:-------:|
> | Single frozen pre-trained model             | 20.7 | **95.4** | 50.5 |
> |  Fine-tuning pre-trained model   | 71.7 | 91.5     | 79.6 |
> | Input-free contaminated MoE      | 50.3 | 87.4     | 68.3 |
> | **Softmax-contaminated MoE** | **72.3** | 94.7 | **82.4** |
>
> ### Conclusion
>
> The above experiments empirically justify our two main theoretical findings:
>
> 1. **Softmax-contaminated MoE outperforms input-free contaminated MoE [2]:** Across both distinguishable and non-distinguishable settings, we observe that the Softmax-contaminated MoE achieves the highest and most balanced performance, improving animal accuracy while maintaining high vehicle accuracy. On the other hand, input-free contaminated MoE underperforms, and fine-tuning pre-trained model sits in between.
>
> 2. **Performance of softmax-contaminated MoE under distinguishable settings is higher than that under non-distinguishable setting:** It can be seen that the performance of softmax-contaminated MoE under the distinguishable setting presented in Table 1a is generally higher than that under the non-distinguishable setting presented in Table 1b.
>
> ***
> **Q4: How do these findings manifest in real fine-tuning tasks? For instance, if one fine-tunes a large language model with a small prompt on a new task that is actually already solvable by the base model, does one observe slower convergence or higher data requirements (as the theory suggests)? The current experiments are synthetic for control, but have the authors thought about testing their ideas on a toy real-world scenario (e.g., fine-tuning a small neural network with a fixed part vs. a learned part)?**
>
> Thanks for your questions. As mentioned in Q3, we include few shot adaptation experiments on DINOv2 ViT-S embeddings of CIFAR-10. We compare two cases: (i) _distinguishable setting_, where the prompt does not share knowledge with the pre-trained model; and (ii) _non-distinguishable setting_, where they share some common knowledge. In both cases, the performance of softmax contaminated MoE under the distinguishable setting (see the above Table 1a) is higher than that under the non-distinguishable setting (see the above Table 1b). This observation aligns with our theoretical findings.
>
> ***
> **References**
>
> [2] Yan et al. "Understanding expert structures on minimax parameter estimation in contaminated mixture of experts." In AISTATS, 2025.

---

### Official Review · Reviewer_at8P · 2025-07-01

**Clarity:** 3
**Significance:** 3
**Originality:** 3
**Rating:** 5
**Confidence:** 3

**Summary:**

This paper provides a theoretical understanding of contaminated mixture-of-expert (MoE) in the presence of input-dependent gating. The analysis implies that input-dependent gating helps finding the optimal solution as compared to the input-free gating. The analysis also implies that having diversity between experts (distinguishability) is beneficial in terms of finding the optimal solution.

**Questions:**

- Clear motivation of the considered setting -- one frozen, one non-frozen expert; optimal model also assumes the one frozen, one non-frozen expert
- What is the significance of finding the optimal parameters given that optimal density is already found?
- Detailed comparison with input-free gating (convergence rate under non-distinguishability, convergence rate in terms of density) and reference [26] (why one cannot restrict the model space of one expert in [26] to get the similar, but more general, result of this work)

**Ethical Concerns:**

["NO or VERY MINOR ethics concerns only"]

**Final Justification:**

My main concern of this work was (i) why one needs to care parameter convergence rate (which is the main focus of this work) given that we already have density (i.e., model output) convergence rate that achieves the same asymptotic rate; (ii) practical motivating example of parameter efficient fine-tuning is not well aligned with the setting of this work (1), since parameter efficient fine-tuning (e.g., prefix-tuning) does not aggregate the final likelihoods but rather aggregates intermediate, non-probabilistic, decisions, which cannot be covered by (1). And during the rebuttal/discussion periods, the authors clarified the first point for its usefulness in understanding the expert specialization, i.e., how fast an expert specializes in some specific task, by pointing out the existing literature on MoE. However, the second point has not been perfectly addressed since the authors agreed that parameter efficient fine-tuning indeed cannot be expressed in the form of (1) -- aggregation of likelihoods -- and requires additional equation that involves conditional expectation -- aggregation of the non-probabilistic outputs.

As a result, for its good theoretical contribution in the MoE community where the results of this work can also be well combined with existing literature, e.g., on expert specialization, I recommend this work's score as 5.

**Limitations:**

yes

**Quality:**

3

**Strengths And Weaknesses:**

_Strengths_

- In-depth theoretical analysis of contaminated MoE
- Near-tightness of the bound (nearly matching the minimax lower bound)

_Weaknesses_

- Single expert is assumed
- Significance of parameter estimation is not clear

_Detailed Comments_

1) _Motivating contaminated MoE from prefix tuning is not so clear_

In prefix tuning, only single decision is made unlike MoE where multiple decisions are separately made, and then aggregated. I think in order for prefix tuning to be related to MoE, it should be something like: one decision made by without prefix, the other made with prefix, and then the final decision is made by combining these two. But since existing prefix tuning is not operating like in this way (at least to my understanding), the current motivation seems to be misleading.

2) _Having one frozen expert and one trainable expert, isn't it a too special case of MoE?_

As far as I am aware of, MoE generally assumes multiple experts with which it achieves higher performance that goes beyond the capacity of individual experts. In this regard, (i) having two experts and (ii) assuming the optimal model (1) is also obtained from these two experts seems to be a too narrow case to call this work as MoE. Also, given that mostly softmax treats multiple inputs and it is more common to call it sigmoid if there is only two inputs, I'm also not sure whether calling this work as softmax-contaminated MoE is the best way one can do. I suggest to provide a clearer, practical, justification of (i) and (ii), followed by theoretical difficulties if one does not assume (i) and (ii).

3) _Proposition 3 already shows that the MoE can find the optimal density in a sublinear manner. Why one needs a further results on the parameter itself?_

Proposition 3 shows that the density of MoE converges to the optimal one with sublinear rate. Given that what we need in the end is the density that works fairly well with the optimal one, it is not clear to me why we need to go further to find the optimal parameters. In other words, why knowing the optimal parameters is more informative than knowing the optimal density? Perhaps this can be related to some sort of explainability of the models, but it is not very clear in the current description.

4) _Can we have minimax lower bound for the density result?_

 Theorem 1 and 2 show the convergence rate of the parameters while Proposition 3 is about the rate for the density. Theorem 1 can be treated as near-optimal as it nearly matches with minimax lower bound, Theorem 2. Given that Theorem 1 is built upon Proposition 3, I guess Proposition 3 is also somewhat near-optimal in the sense of matching the minimax lower bound for the density. But is it really true? Can this be proved?

5) _Under non-distinguishability, input-dependent gating would be still better than input-free gating?_

The authors showed that, under distinguishability setting, input-dependent gating helps finding the optimal parameters compared to input-free gating. The authors also showed that, under non-distinguishability, convergence rate of input-dependent gating becomes slower. In such a case, would it be even worse than the input-free gating? Also, how does the convergence rate in terms of density (Proposition 3) compares with that of input-free gating? I guess in terms of density, both input-free and input-dependent gating would admit the same convergence rate (maybe I'm wrong!), and in light of my comment 3), unless there is clear significance of estimating the parameters per se, the advantage of input-dependent gating seems not so clear.

6) _Comparison with [26] is not so clear_

Reference [26] also considered softmax, input-dependent, gating with multiple, more than 2, experts. Of course, they did not consider the frozen expert hence it is not directly applicable to the setting of interest, but it would be nice to clarify the connection of this work with [26]. For instance, why one cannot consider the special case of [26] where one expert has its model space with only single element hence essentially serves the role as the frozen expert.

_Minor Comments_

- Using \$h()$ for both mean expert function and Hellinger distance is confusing.
- In the proof of Lemma 3 (p 37), for the first inequality of line 1105, why a conservative \$\delta/4$ is considered given that considering \$\delta$ seems to have no problem in the upcoming steps?
- In Sec. B. 3, \$\mu$ is not clearly defined, e.g., in (57) and line 1116, \$\mu$ appears which seems to be not defined before.
- In proof of Theorem 2, line 873 -- 875 is not clear how \$\tau_n$ lies in a compact set makes the desired term goes to 0.

---

> ### Author Rebuttal · Authors · 2025-07-30
>
> Dear Reviewer at8P,
>
> Thanks for the wealth of your comments, and giving **good (3) grade** to the quality, clarity, significance and originality of our paper. Below are our responses to your questions. We will also add these changes to the revision of our manuscript. We hope that we have addressed your concerns regarding our paper.
>
> ***
>
> **Q1: Motivating contaminated MoE from prefix tuning is not so clear.**
>
> Thanks for your comment. We clarify that contaminated MoE is a variant of MoE which is inspired by prefix tuning method but does not necessarily share the same concept as prefix tuning. In particular, in prefix tuning [3], the original model parameters are frozen, and then they attend to trainable prefix to learn downstream tasks parameter-efficiently. Inspired from this concept, we consider contaminated MoE, a mixture of a frozen pre-trained model and a trainable prompt expert to learn downstream tasks without having to tune the entire pre-trained model. We will add a clarifying comment.
>
> ***
>
> **Q2: Having one frozen expert and one trainable expert, isn't it a too special case of MoE?**
>
>
> Thank you for raising this point. We view the “one frozen $+$ one trainable” configuration as a contaminated mixture of experts: an instantiation that preserves the strong pretrained behavior as an anchor while also enabling targeted adaptation via a small trainable expert. This is not a degenerate case but a deliberate design choice that trades capacity for stability, sample‑efficiency, and interpretability. The formulation extends naturally to the case of more than two experts, where we keep some frozen anchor experts alongside trainable ones; the theoretical analysis proceeds in the same direction. Finally, the pattern aligns with the widely used “frozen backbone + lightweight adapter” practice—our contribution is to cast it as an MoE, prove guarantees, and show that it is a strong and extensible baseline rather than a restrictive special case.
>
> ***
>
> **Q3:  Proposition 3 already shows that the MoE can find the optimal density in a sublinear manner. Why one needs a further results on the parameter itself?**
>
> Thank you for your valuable comment and suggestions. We believe that parameter estimation provides deeper insight into the model by enabling users to identify which components of the parameters contribute to observed differences. This enhances the explainability of the model. Furthermore, in the context of parametric models, knowing the parameters allows for straightforward data generation, whereas relying solely on the density function is often too general and may require additional information. Therefore, parameter estimation not only improves explainability but also offers practical advantages in terms of data generation and model understanding.
>
> ***
>
> **Q4: Can we have minimax lower bound for the density result?**
>
> Thank you for your suggestions. We believe that a similar result can be established in this setting. One possible approach is to employ Le Cam’s two-point method or its generalization, as presented in Lemma C.1 of [8]. This technique is also utilized in Lemma 2 of our article. The core idea is to construct two sequences of parameters whose distance is sufficiently small to yield a meaningful lower bound. While this construction is relatively straightforward in the context of parameter estimation, it becomes considerably more challenging when extended to density functions, due to the complexity of controlling the Hellinger distance between distributions. For this reason, we defer the derivation of the minimax lower bound for density estimation to future work.
>
>
> ***
>
> **Q5: Under non-distinguishability, input-dependent gating would be still better than input-free gating?**
>
> Thank you for your comment. We would like to reaffirm the advantage of the input-dependent setting over the input-free setting in terms of the asymptotic behavior of parameter estimation. Specifically, under the assumption that $\tau$ belongs to a compact set such that $\exp(\tau) \geq c > 0$, Theorem 3 and Table 1 in our paper (under the non-distinguishable setting) indicate that the convergence rate for the prompt parameters estimation $\eta^\ast$ and $\nu^\ast$ does not depend on the gating parameters. In contrast, as shown in Table 1 and Theorem 8 in [26], the convergence rate in the input-free setting depends on $\lambda^\ast$, in particular, as $\lambda^\ast \to 0$, the prompt component vanishes, which significantly slows the convergence rate of parameter estimation.
>
> Although the convergence rate for density estimation remains the same in both settings, the input-dependent formulation alleviates the slow convergence issue caused by the vanishing prompt phenomenon. In this sense, the use of a sigmoid gating function helps mitigate the adverse effects of prompt vanishing, which could otherwise hinder the reliability and accuracy of parameter estimation.
>
> ***
>
> **Q6: Comparison with [26] is not so clear.**
>
> Thanks for your feedback. There are three main differences between [26] and our work.
>
> *1. More general settings:* We allow ground-truth parameters $G_*$ to change with the sample size $n$, which is closer to practical settings than the assumptions in [26] where $G_*$ does not change with $n$. Furthermore, we consider a general expert function $h$, while the expert function considered in [26] is restricted to be of linear form.
>
> *2. Uniform convergence rates:* Since ground-truth parameters vary with the sample size, the convergence rates of parameter estimations in our work are uniform rather than point-wise as in [26]. Additionally, these rates are able to capture the interaction between the convergences of parameter estimations.
>
> *3. Minimax lower bounds:* Finally, we determine minimax lower bounds under both distinguishable and non-distinguishable settings. Based on these lower bounds, we can claim that our derived convergence rates are optimal. However, no minimax lower bounds are provided in [26].
>
> ***
>
> **Q7: Detailed comparison with input-free gating (convergence rate under non-distinguishability, convergence rate in terms of density) and reference [26] (why one cannot restrict the model space of one expert in [26] to get the similar, but more general, result of this work).**
>
> Thank you for your comment. We would like to clarify the distinctions between this paper and [26], particularly with regard to the role of the softmax gating function.
>
> 1. In terms of technique:  The current paper requires a more delicate analysis to handle the differences between two density functions (e.g., as seen in lines 775 and 991), in contrast to the relatively simpler treatment in the proof of Lemma 2 in [26]. This increased complexity arises from the interaction induced by the input-dependent softmax gating.
>
> 2. In terms of insight:  As discussed in our response to Q5, the softmax gating function is not only more applicable in practice—reflecting realistic dependence on input—but also plays a crucial role in addressing the issue of prompt vanishing. This property improves the stability and accuracy of parameter estimation, representing an important theoretical and practical advantage over the setting considered in [26].
>
> ***
>
> **Minor Suggestions:**
> Thank you very much for your thorough revision.
>
> 1.  Regarding the notation for the expert mean function and the Hellinger distance, we have already updated the notation for the Hellinger distance as $d_H$ to improve the clarity and consistency of our manuscript.
> 2.  As for the use of $\delta/4$ (line 1105), this choice is made purely for technical convenience and does not affect the generality of the results.
> 3. The symbol $\mu$ refers to the Lebesgue measure, and we have now explicitly clarified this in the manuscript.
> 4. Concerning $\tau_n$ (lines 873–875), since $\tau_n$ lies within a compact set, $\exp(\tau_n)$ is bounded above and below by positive constants. As a result, $d_1(S_{1,n},S_{2,n}) = \mathcal{O}(\|(\beta_1,\eta_1,\nu_1) - (\beta_2,\eta_2,\nu_2)\|)$ , which ensures control over the remainder term and the coefficient of the first-order term in the Taylor expansion above.
>
> ***
>
> **References**
>
> [3] Xiang Lisa Li and Percy Liang. Prefix-tuning: Optimizing continuous prompts for generation, 2021.
>
> [8] S. Gadat, J. Kahn, C. Marteau, and C. Maugis-Rabusseau. Parameter recovery in two-component
> contamination mixtures: The lˆ2 strategy. 2020.
>
> [26] H. Nguyen, T. Nguyen, and N. Ho. Demystifying softmax gating function in Gaussian mixture of experts. In NeurIPS, 2023.

---

> > ### Comment · Reviewer_at8P · 2025-08-06
> >
> > Thanks to the authors for their detailed response! Most of my concerns are well addressed, though I would like to confirm a few points (related to my Q1 and Q3) with the authors.
> >
> > Q1. Is prefix tuning a good motivating example?
> >
> > My main concern regarding prefix tuning is that, unlike (1) where both the pretrained model and prompt model make the decision _separately_, with the aggregation being carried out at the decision level, prefix tuning never makes separate decision from the prefix and the backbone model; it rather jointly makes a single decision. I understand that prefix tuning is a timely topic hence a good motivation, but it is still not clear to me how the authors can mitigate such big operational differences. I think for any motivating examples, we need to be able to properly map their operation to (1).
> >
> > Q3. What is the significance of parameter estimation compared to density estimation?
> >
> > The authors' viewpoint that parameter estimation is useful for data generation is interesting, though I'm a bit perplexed. Specifically, I'm confused with the authors' claim that "knowing the parameters allows for straightforward data generation, whereas relying solely on the density function is often too general and may require additional information". Suppose that we have some MoE and have optimized its parameters using available \$n$ number of data samples. It is clear that we should have the optimized MoE model at hand. Now from Proposition 3, the outcome of MoE is near-optimal (suppose that we have sufficient \$n$). It may be not optimal in the sense of distance between the optimal parameters, but, why one would be bothered with the distance in the parameter  space given that it already provides a decent output (density)? Also, given that we have the optimized MoE model at hand, we can generate the data as much as we can. Can the authors please elaborate more on this? It is not clear in what sense knowing parameter makes data generation easier given that, at the end of the day, we just need our MoE to output as best as it can, which is already covered by Proposition 3. Sorry for being picky on this, but given that the most beauty of the authors' theoretical analysis comes from the parameter estimation, I think it should be clarified why one needs parameter convergence, given that we already have good enough density convergence (also given that the model convergence seems not enjoying any faster convergence rate).

---

> ### Author Response · Authors · 2025-08-06
> **Response to Reviewer at8P**
>
> Dear Reviewer at8P,
>
> Thanks for your response. We are glad to hear that most of your concerns have been well addressed. Next, let us clarify the rest as follows.
>
> 1. Is prefix tuning a good motivating example?
>
> Thanks for your question. Firstly, let us provide an example in which contaminated MoE aligns with the concept of prefix tuning. For example, let's say we would like to fine-tune a Transformer architecture through its feed-forward layer. For the sake of parameter efficiency, we keep the original feed-forward network (FFN) frozen and then add a prefix or prompt (a learnable FFN) to it. Then, the decision or output of the Transformer is jointly made by the prefix and the frozen parameters.
>
> Secondly, we would like to note that contaminated MoE is fundamentally an MoE variant inspired from parameter-efficient fine-tuning methods in general as we mentioned in line 29. In addition to prefix tuning, contaminated MoE can also be viewed as an adaptation method when the pre-trained model and the prompt model make the decision separately as described by the reviewer.
>
> In summary, we confirm that contaminated MoE is an MoE variant motivated by parameter-efficient fine-tuning methods, and prefix tuning is a good example.
>
> 2. What is the significance of parameter estimation compared to density estimation?
>
> Thanks for your question. Let us explain the significance of parameter estimation more clearly. In particular, expert specialization, i.e., how fast an expert specializes in some specific tasks, is a problem of interest in the MoE literature [1]. A way to capture the expert specialization from the theoretical perspective is to characterize the expert convergence rates. For that purpose, we need to determine the convergence rates of parameter estimation. Then from the convergence rates of  expert parameter estimation, we can establish the convergence rates of expert estimation. Lower expert convergence rates indicate that experts need less data to specialize in some specific tasks with a given approximation error, and vice versa. Therefore, performing a convergence analysis of parameter estimation might help improve the model sample-efficiency in terms of expert estimation.
>
> Due to the above link between parameter estimation and expert specialization, there were several previous works in the MoE literature investigating the convergence behavior of parameter estimation, namely [2, 3, 4].
>
> We hope that our response successfully addresses your concerns. If the reviewer has further concerns, please feel free to let us know. We are happy to address any additional concerns from you.
>
> Best regards,
>
> The Authors
>
> ***
> **References**
>
> [1] D. Dai, C. Deng, C. Zhao, R. X. Xu, H. Gao, D. Chen, J. Li, W. Zeng, X. Yu, Y. Wu, Z. Xie, Y. K. Li, P. Huang, F. Luo, C. Ruan, Z. Sui, and W. Liang. Deepseekmoe: Towards ultimate expert specialization in mixture-of-experts language models. arXiv preprint arXiv:2401.04088, 2024.
>
> [2] N. Ho, C.-Y. Yang, and M. I. Jordan. Convergence rates for Gaussian mixtures of experts. In Journal of Machine Learning Research, 2022.
>
> [3] D. Do, L. Do, and X. Nguyen. Strong identifiability and parameter learning in regression with heterogeneous response. In Electronic Journal of Statistics, 2025.
>
> [4] H. Nguyen, N. Ho, and A. Rinaldo. Sigmoid gating is more sample efficient than softmax gating in mixture of experts. In NeurIPS, 2024.

---

> > ### Comment · Reviewer_at8P · 2025-08-07
> >
> > Thank you very much again! However I am still a bit confused.
> >
> > 1) Can you please provide how prefix tuning be expressed in the form of (1)? I still cannot understand how one can formulate (1) in prefix tuning. And I believe that the most efficient way of our communication would be using mathematical language. That said, if the authors can tell me what would be \$f_0(y|h_0(x,\eta_0),\nu_0)$ and \$f(y|h_0(x,\eta*),\nu*)$ in prefix tuning, it would be very helpful for me to understand.
> >
> > 2) Thanks for sharing the concept of expert specialization! I need to check the literature, but at least it is not clear to me directly why convergence rate of parameter is more useful than convergence rate of the density, given that again, what we in the end need from the expert is its performance (density) not its intrinsic parameters. That said, it seems to me that nothing prevents us to consider expert specialization in terms of the model output, not in terms of the parameters. In general, I think the reason of studying parameter convergence in the other literature comes from its easier analysis than understanding the model output directly. But it seems opposite to the authors' case -- we already know the convergence rate of the model output, and the convergence rate of the parameter is carried out for the sake of some theoretical fun. Also I'm concerned that the authors' response for the significance of parameter convergence is not consistent and has been changed during the response. Can the authors please provide just one clear reason why density convergence is not enough and we need parameter convergence (which is the main point of this work) which can be understood by the general audience of NeurIPS? Thank you.

---

> ### Author Response · Authors · 2025-08-07
> **Further Response to Reviewer at8P (Part 1)**
>
> Dear Reviewer at8P,
>
> Thanks for your response. Let us clarify your concerns as follows.
>
> **1. Regarding the connection between prefix tuning and contaminated MoE:** Let uss present our example in a mathematical way as suggested by the reviewer. In particular, let’s say we would like to fine-tune a Transformer architecture through its feed-forward layer, where the feed-forward network (FFN) is frozen and plays a role as $f_0(y|h_0(x,\eta_0),\nu_0)$ in contaminated MoE. Next, to learn downstream tasks parameter-efficiently, we add to the frozen FFN a prefix or prompt, that is, a learnable FFN playing a role as $f(y|h(x,\eta^{\ast}),\nu^{\ast})$. Then, the feed-forward layer becomes
>
> $\omega_{learnable}\times FFN_{frozen} + (1-\omega_{learnable})\times FFN_{learnable}$,
>
> where $\omega_{learnable}=\frac{1}{1+\exp((\beta^{\ast})^{\top}x+\tau^*)}$. This is exactly the form of equation (1) in our manuscript. Above, the frozen FFN and the learnable FFN jointly makes a single decision of the feed-forward layer, or more generally, the Transformer architecture.
>
> Lastly, it should be noted that contaminated MoE is fundamentally an MoE variant inspired from parameter-efficient fine-tuning methods. Thus, there might be some ways of interpretation that contaminated MoE is not exactly the same as prefix-tuning. So, we hope that the reviewer will be flexible and open-minded about this concern.
>
> **2. Regarding the parameter estimation problem:** Let us address your concerns, respectively, as follows.
>
> *(2.1) “it is not clear to me directly why convergence rate of parameter is more useful than convergence rate of the density”:* We clarify that we did not say that the convergence rate of parameter estimation is more useful than the convergence rate of density estimation. In our previous response, we only explained why we considered the parameter estimation without making any comparison. In fact, the problems of density estimation and parameter estimation have their own significance, and it is not necessary to make a comparison between them. A notable relation between these two problems is that in our proof arguments, the convergence rate of density estimation is necessary for deriving the the convergence rate of parameter estimation.
>
> *(2.2) “nothing prevents us to consider expert specialization in terms of the model output, not in terms of the parameters”*:  To the best of our knowledge, from the theoretical perspective, the only way to capture the expert specialization in the MoE literature is through the convergence rate of expert estimation, which can be derived from the convergence rate of parameter estimation as in [4]. Since the density estimation rate is necessary for deriving parameter estimation rate, we considered both problems in our work. There might be other approaches without involving parameter estimation which is out of our awareness. However, such approaches lie beyond the scope of our work.
>
> *(2.3) “I think the reason of studying parameter convergence in the other literature comes from its easier analysis than understanding the model output directly.“*: We confirm that this is not true. In fact, in previous works in the literature [2, 3, 4], the convergence rate of density estimation (model output) is necessary for deriving the parameter estimation rate. In other words, to derive parameter estimation rates, the authors in [2, 3, 4] need to derive density estimation rate first. For instance,
>
> *(Please note that the following indices of theorems, propositions, lemmas are from their official publications in journal, conferences but not from their arXiv versions)*
>
> - In [2]: the authors studied the density estimation rate in Proposition 5, and then used it to study the parameter estimation rate in Theorem 7 through Lemma 21;
>
> - In [3]: the authors studied the density estimation rate in Theorem 3, and then used it to study the paramter estimation rate in Theorem 4;
>
> - In [4]: the authors studied the model estimation rate in Theorem 1, and then used it to study the parameter estimation rate in Theorem 2;
>
> - In our work, we also studied the model estimation rate in Proposition 3, and then used it to study the paramter estimation rate in the main theorems (see lines 150-157 for the link between density estimation and parameter estimation).
>
> Hence, we can say that the convergence analysis of parameter estimation is even more challenging than the convergence analysis of density estimation. Therefore, we performed the convergence analysis of parameter estimation was not for theoretical fun but for capturing the expert specialization that we will explain the following thread (due to the character limit).

---

> ### Author Response · Authors · 2025-08-07
> **Further Response to Reviewer at8P (Part 2)**
>
> *(2.4) the authors’ response for the significance of parameter convergence is not consistent and has been changed during the response. Can the authors please provide just one clear reason why density convergence is not enough and we need parameter convergence?*:
>
> Firstly, let us restate the purpose of the analysis of parameter estimation and comment on the consistency of our response towards it. In particular, expert specialization, i.e., how fast an expert specializes in some specific tasks, is a problem of interest in the MoE literature [1]. A way to capture the expert specialization from the theoretical perspective is to characterize the expert convergence rates. For that purpose, we need to determine the convergence rates of parameter estimation. Then from the convergence rates of expert parameter estimation, we can establish the convergence rates of expert estimation. Lower expert convergence rates indicate that experts need less data to specialize in some specific tasks with a given approximation error, and vice versa. Therefore, performing a convergence analysis of parameter estimation helps improve the sample-efficiency in terms of expert estimation.
>
> --> This was the reason why in our rebuttal, we said that the analysis of parameter estimation offers practical advantages in terms of data generation. We agree that this claim is quite confusing. It should have been that the analysis of parameter estimation offers a solution to achieve equivalent expert estimation errors with less data, or obtain smaller expert estimation errors using the same amount of data. We apologize for this inconvenience.
>
> Secondly, as we mentioned in (2.2), the only way to theoretically capture the expert specialization in the MoE literature is through the convergence rate of expert estimation, which can be derived from the convergence rate of parameter estimation. Therefore, we had to conduct the analysis of parameter estimation on top of the analysis of density estimation for capturing the expert specialization, which is an important problem in MoE [1, 5].
>
> ***
> Should the reviewer have additional concerns, please feel free to let us know. We are happy to address all of them.
>
> Thank you,
>
> The Authors
>
> ***
> **References**
>
> [1] D. Dai, C. Deng, C. Zhao, R. X. Xu, H. Gao, D. Chen, J. Li, W. Zeng, X. Yu, Y. Wu, Z. Xie, Y. K. Li, P. Huang, F. Luo, C. Ruan, Z. Sui, and W. Liang. Deepseekmoe: Towards ultimate expert specialization in mixture-of-experts language models. arXiv preprint arXiv:2401.04088, 2024.
>
> [2] N. Ho, C.-Y. Yang, and M. I. Jordan. Convergence rates for Gaussian mixtures of experts. In Journal of Machine Learning Research, 2022.
>
> [3] D. Do, L. Do, and X. Nguyen. Strong identifiability and parameter learning in regression with heterogeneous response. In Electronic Journal of Statistics, 2025.
>
> [4] H. Nguyen, N. Ho, and A. Rinaldo. Sigmoid gating is more sample efficient than softmax gating in mixture of experts. In NeurIPS, 2024.
>
> [5] J. Oldfield et al. Multilinear Mixture of Experts: Scalable Expert Specialization through Factorization. In NeurIPS, 2024.

---

> > ### Comment · Reviewer_at8P · 2025-08-08
> >
> > Thank you very much for the clarification! I hope that the authors can clarify in the revised version about the "expert specialization" and how this is connected with the authors' main result -- convergence rate of parameter estimation -- and why it cannot be addressed by simpler, convergence rate of density estimation.
> >
> > For the prefix tuning motivating example, can the authors please provide me at least one reference on prefix tuning that frozen FFN and learnable FFN separately make decisions and then they are being combined? I think decisions being made jointly in some internal fashion cannot justify this, and we rather need explicit description that, frozen FFN makes its own decision and learnable FFN makes its own decision, and then they are combined at the density level. For instance, if the authors can find single reference on prefix tuning that, one decision is made by "without prefix" transformer and the other is made by "with" prefix transformer, and then they combine these two, then I can be assured by this motivating example.
> >
> > I would really like to be open-minded, but, I think motivating example that only confuses the setting of the work, is worse than having no motivation.

---

> > > ### Author Response · Authors · 2025-08-08
> > >
> > > Dear Reviewer at8P,
> > >
> > > Thanks for your response. Firstly, we are glad to hear that our response addresses your concern regarding the significance of parameter estimation. We will definitely incorporate the connection between expert specialization and parameter estimation into the revision of our manuscript.
> > >
> > > Secondly, let us provide some references showing that contaminated MoE is motivated from prefix tuning. These references considered fine-tuning attention layers rather than feed-forward layers. We hope that the reviewer can be sympathetic to this difference given the limited time of the discussion period.
> > >
> > > In [1], the authors provided a connection between prefix tuning and contaminated MoE in equations (7) and (10). In these equations,
> > >
> > > - $\lambda(\boldsymbol{x})$ played a role as a softmax weight in our paper,
> > >
> > > - $Attn(xW_q,CW_k,CW_v)$ corresponded to $f_0(y|h(x,\eta_0),\nu_0)$,
> > >
> > > - $Attn(xW_q,P_k,P_v)$ corresponded to $f(y|h(x,\eta^{\ast}),\nu^{\ast})$.
> > >
> > > In [2], the authors provided that connection in equation (7). In that equation,
> > >
> > > - $\boldsymbol{A}^{pt}_{io}$ played a role as a softmax weight in our paper,
> > >
> > > - $\boldsymbol{t}_i$ corresponded to $f_0(y|h(x,\eta_0),\nu_0)$,
> > >
> > > - $\boldsymbol{W}_V\boldsymbol{s}_1$ corresponded to $f(y|h(x,\eta^{\ast}),\nu^{\ast})$.
> > >
> > > Should the reviewer have additional concerns, please feel free to let us know. We are happy to address all of them. Otherwise, we hope that the reviewer can consider increasing the rating of our paper. Thanks again for your spending precious time and great effort in reviewing our manuscript, which have helped us improve our paper substantially. We really appreciate it.
> > >
> > > Best regards,
> > >
> > > The Authors
> > >
> > > ***
> > > **References**
> > >
> > > [1] He et al. Towards a Unified View of Parameter-Efficient Transfer Learning. In ICLR, 2022
> > >
> > > [2] Petrov et al. When Do Prompting and Prefix-Tuning Work? A Theory of Capabilities and Limitations. In ICLR, 2024

---

> > > > ### Comment · Reviewer_at8P · 2025-08-08
> > > >
> > > > Thank you for the references! However, I'm afraid that the shared references only increase my concern on misleading motivation of prefix tuning for this work. In your (1), \$f_0(y|h_0(x,\eta_0),\nu_0)$ and \$f(y|h_0(x,\eta*),\nu*)$ both are the final decisions (likelihood of \$y$ given \$x$), not intermediate decisions as in the equations that the authors pointed out in the suggested references. Furthermore, the things that the authors refer to as the \$f_0(y|h_0(x,\eta_0),\nu_0)$ and \$f(y|h_0(x,\eta*),\nu*)$ are the "softmax(attention score) * value", which themselves cannot play the role of final decision, i.e., \$p(y|x)$ (also they are not even in the form of probability density!). I would love to increase my score, but unless we can make this point crystal clear, I cannot take any further actions. Sorry for asking this in the late phase, but I thought that I was making this point quite clear from the beginning -- whether prefix tuning aggregates the "final decisions" as per (1), which was contrast with my understanding of how prefix tuning works --, which has been just assured by the suggested reference by the authors. Hope we can find a good consensus on this in the remaining discussion period, thanks!

---

> ### Author Response · Authors · 2025-08-08
>
> Dear Reviewer at8P,
>
> Thanks for your response. Your concern became clearer for us when you mentioned the likelihood of $y$ given $x$. So, let us clarify your concerns as follows.
>
> In our work, we consider contaminated MoE from a probabilistic view, that is, assuming the data are generated according to a softmax-contaminated MoE model with the conditional density function $p_{G_{\ast}}(y|x)$ of the response $Y$ given the covariate $X$ given in equation (1). We guess that this data generation process might cause the reviewer’s concern about the concepts of “intermediate decisions” and “final decision”. We clarify that the probabilistic model in equation (1) is for the purpose of generating data. To capture the model decision, we should consider the conditional expectation of the response $Y$ given the covariate $X$, that is,
>
> $E[Y|X]=\frac{1}{1+\exp((\beta^{\ast})^{\top}x+\tau^{\ast})}\cdot h_0(x,\eta_0)+\frac{\exp((\beta^{\ast})^{\top}x+\tau^{\ast})}{1+\exp((\beta^{\ast})^{\top}x+\tau^{\ast})}\cdot h(x,\eta^{\ast}).$
>
> Please refer to the equation of expectation on page 5 of [3] as an example for applying hierarchical MoE to healthcare tasks. We think this deterministic version of contaminated MoE aligns better with the concept of prefix tuning, that is, the functions $h_0(x,\eta_0)$ and $h(x,\eta^{\ast})$ make intermediate decisions. For better understanding, let us explain our previous references again here.
>
> In [1], the authors provided a connection between prefix tuning and contaminated MoE in equations (7) and (10). In these equations,
>
> - $\lambda(\boldsymbol{x})$ played a role as a softmax weight in our paper,
>
> - $Attn(xW_q,CW_k,CW_v)$ corresponded to $h_0(x,\eta_0),\nu_0)$,
>
> - $Attn(xW_q,P_k,P_v)$ corresponded to $h(x,\eta^{\ast})$.
>
> In [2], the authors provided that connection in equation (7). In that equation,
>
> - $\boldsymbol{A}^{pt}_{io}$ played a role as a softmax weight in our paper,
>
> - $\boldsymbol{t}_i$ corresponded to $h_0(x,\eta_0)$,
>
> - $\boldsymbol{W}_V\boldsymbol{s}_1$ corresponded to $h(x,\eta^{\ast})$.
>
> We hope that the above explanation clears your concern about the motivation of contaminated MoE. We will incorporate these changes to the revision of our manuscript.
>
> Thanks again for your spending precious time and great effort on engaging in the discussion actively, we really appreciate it. We are looking forward to your response.
>
> Thank you,
>
> The Authors
>
> ***
> **References**
>
> [1] He et al. Towards a Unified View of Parameter-Efficient Transfer Learning. In ICLR, 2022
>
> [2] Petrov et al. When Do Prompting and Prefix-Tuning Work? A Theory of Capabilities and Limitations. In ICLR, 2024
>
> [3] Nguyen et al. On Expert Estimation in Hierarchical Mixture of Experts: Beyond Softmax Gating Functions. arXiv preprint 2410.02935 Version 2.

---

> > ### Comment · Reviewer_at8P · 2025-08-08
> >
> > Thank you very much! I like the new expression using conditional expectation, which makes the connection with prefix tuning clearer. Also thanks for the reference [3], I like it! But now given that the authors use the conditional expectation to do the decision making, rather than using the likelihood approach, what would happen to (2)? It seems that conditional expectation is far less informative than the likelihood (especially given that conditional expectation works bad for multi-modal case, which, in contrast, can be handled nicely by likelihood-based MoE) and I'm afraid that decision-making based on conditional expectation cannot be applied to (2).
> >
> > In short, my concern is: are the authors arguing that all the results in the paper can be applied to the decision making in the form of conditional expectation? If not, how to justify the motivation that can only be applied to the decision making based on conditional expectation (prefix tuning), which is clearly different with decision making based on likelihood (eq. 1)? I think we are getting closer by the way, thanks again!

---

> ### Author Response · Authors · 2025-08-08
>
> Dear Reviewer at8P,
>
> Thanks for your response. We’re happy to hear that we are getting closer to find a consensus on this discussion. Let us clarify the rest concerns as follows.
>
> 1. **Given that the authors use the conditional expectation to do the decision making, rather than using the likelihood approach, what would happen to (2)?**: In equation (2), we use the maximum likelihood method with the data generated from the conditional density function in equation (1) to estimate ground-truth parameters. Therefore, we still need to employ the likelihood function for that purpose. We believe that involving the likelihood function to estimate density and parameters does not affect the decision making formulated by the conditional expectation.
>
> 2. **Can all the results in the paper be applied to the decision making in the form of conditional expectation?** We confirm that our convergence analysis can totally be applied to the case of deterministic MoE under the regression-based framework. In particular, we can assume that the data $(X_i,Y_i)$ are generated from a regression framework
> $$Y_i=f_{G_{\ast}}(X)+\epsilon_i,$$
> for $i=1,2,\ldots,n$, where the regression function takes the form
> $$f_{G_{\ast}}(x)=\frac{1}{1+\exp((\beta^{\ast})^{\top}x+\tau^{\ast})}\cdot h_0(x,\eta_0)+\frac{\exp((\beta^{\ast})^{\top}x+\tau^{\ast})}{1+\exp((\beta^{\ast})^{\top}x+\tau^{\ast})}\cdot h(x,\eta^{\ast}),$$
> and $\epsilon_1,\epsilon_2,\ldots,\epsilon_n$ are independent Gaussian noise variables such that $E[\epsilon_i|X_i]=0$ for all $i=1,2,\ldots,n$.
>
> Next, by applying the least squares method, we can estimate the ground-truth regression function and parameters as
>
> $$\hat{G}n:=\arg\min_{G}\sum_{i=1}^{n}(Y_i-f_{G}(X_i))^2.$$
>
> Then, the analysis of regression function estimation and parameter estimation can be done in a similar way to our work. Please refer to [1] for such analysis for softmax gating MoE. Note that since the noises follow from Gaussian distributions, the above least squares estimator and the maximum likelihood estimator in our equation (2) are equivalent.
>
> We hope that the above explanation clears your remaining concerns. We are looking forward to your response.
>
> Thanks again for actively engaging in the discussion of our submission, we really appreciate it.
>
> ***
> **References**
>
> [1] Nguyen et al. On Least Square Estimation in Softmax Gating Mixture of Experts. In ICML, 2024

---

> > ### Comment · Reviewer_at8P · 2025-08-08
> >
> > Thank you! So my understanding is:
> > - the authors' analysis can be applied to regression case, where the ground-truth data generation is of uni-modality, which can be treated as the special case of this work (from the non-distinguishable setting, I guess)
> > - in the above simplified setting, prefix tuning is a good motivating example
> > - however, when one needs to consider a more complicated setting that allows bi-modality as in (1), i.e., mixture of likelihood, conditional-expectation-type formulation no longer applies (i.e., regression-type analysis no longer applies), which, by the way, nicely analyzed in this work
> > - but, the problem is that, prefix tuning no longer plays the good motivating example, since, it matches only with the simple, regression setting, not with more complicated setting as per (1).
> >
> > From the recent discussions, I think my above understanding is correct (please correct me if I'm wrong!). If so, then my concern is:
> > - prefix-turning is a good motivating example for a very special case of (1) -- regression setting
> > - if the authors think that prefix-tuning is the best example, then why not just focus on regression setting to modify (1), as well as all the analysis?
> >
> > And I believe that at this point, we are on the same page. At least for (1), prefix-tuning is a misleading example as it only serves for a very special case of (1). So, unless the authors want to change all the analysis to the regression (uni-modality), I think we need a better motivating example than prefix-tuning. The reason that I'm insisting on making this point crystal clear is that, although I think I have enough background to understand the setting of this paper, i.e., (1), with enough background on prefix turning, reading this work's motivation only makes me super confused and be perplexed of the theme of this work.

---

> ### Author Response · Authors · 2025-08-08
>
> Dear Reviewer at8P,
>
> Thanks for your response. We would like to clarify that our focus in this paper is on contaminated MoE, not prefix tuning. This MoE variant is inspired from the concept of parameter-efficient fine-tuning methods in general, and prefix tuning is just one of them. To illustrate that inspiration, we have shown the connection between contaminated MoE and prefix tuning through previous references and through the conditional expectation. However, contaminated MoE is fundamentally an MoE variant and, therefore, it cannot be exactly the same as prefix tuning in all aspects. We hope that the reviewer can be open-minded about this.
>
> Next, let us show our plan to make the motivation of contaminated MoE from prefix tuning clearer in the revision of our manuscript. First, we will introduce the conditional expectation after presenting equation (1), and then explain the connection between contaminated MoE and prefix tuning. Regarding the theoretical results, since the noises in the regression framework follows the Gaussian distributions, the least squares estimator in that case and our maximum likelihood estimator in equation (2) are equivalent. This means that there would be no mismatch between the asymptotic properties of these two estimators. With this presentation, we believe that the motivation from prefix tuning becomes clearer, while the theoretical study is still valid. Finally, one of the reasons we consider probabilistic framework rather than regression framework is that the analysis under the probabilistic framework is also a problem of interest in statistics when performing statistical inference on parameters. Moreover, since the probabilistic contaminated MoE has more interesting phenomena, e.g., the distinguishability between frozen model and learnable model, it also captures more attention from the theory community.
>
> We hope that the above explanation clears your remaining concerns. If the reviewer has any suggestion to make the motivation clearer, we really appreciate if you can share with us. We are looking forward to your response.
>
> Thank you,
>
> The Authors

---

> > ### Comment · Reviewer_at8P · 2025-08-09
> >
> > Thank you, so parameter-efficient fine-tuning, including prefix tuning, is well suited for conditional expectation formula (since parameter-efficient fine-tuning also in general never aggregates the likelihoods as in (1)) not for probabilistic formula as in (1). And the authors could not find at the moment good practical machine learning motivating examples that can only be expressed in probabilistic formula, not in conditional expectation formula. However, even without practical motivating examples, the authors think that carrying out analysis on probabilistic formula is more meaningful as (i) it is more general to capture the regression setting; (ii) as it deals with a more fundamental problem that statistics community would like.
> >
> > When I did the first review, I choose my score as 4, as this work has nice theoretical results. And I was willing to raise my score if either my concern on theoretical perspective has been resolved, or my concern on practical perspective has been resolved. The former was about the reason why one needs to care about parameter convergence rather than density convergence, which was well addressed by the authors during the discussion period; while my concern on practical perspective regarding parameter-efficient tuning has not perfectly resolved (see the above reason), though I understand that not all theoretical details can be perfectly motivated by practical examples.
> >
> > For the above reasons, I will raise my score to 5, although it is somewhere around 4.7 in my mind. Hope the authors can clearly apply what we have agreed on during the discussion in the revised version! Thank you very much again for all the efforts during the rebuttal and discussion period!

---

> > > ### Author Response · Authors · 2025-08-09
> > > **Thank You**
> > >
> > > Dear Reviewer at8P,
> > >
> > > Thanks for your response. We are glad to hear that our response addressed your concern regarding the parameter convergence but we also regret that the other concern regarding the motivation from prefix tuning was not perfectly resolved. We will try to address this remaining concern in the revision of our manuscript. Furthermore, we will also incorporate changes in our discussion into the revised manuscript.
> > >
> > > Lastly, we would like to express our gratitude for your huge effort to provide constructive feedback and activaly engage in the discussion of our submission, and for increasing your rating to 5 (Accept). This have helped us improve our manuscript significantly. We really appreciate it!
> > >
> > > Thank you,
> > >
> > > The Authors

---

### Official Review · Reviewer_2SMz · 2025-07-03

**Clarity:** 3
**Significance:** 3
**Originality:** 2
**Rating:** 4
**Confidence:** 3

**Summary:**

This paper provides the first comprehensive statistical analysis of a mixture-of-experts model used in parameter-efficient fine-tuning, where a fixed pre-trained expert and a trainable Gaussian “prompt” expert are combined via a softmax linear gating network. The authors introduce a novel analytic notion of distinguishability between the frozen and prompt experts that guarantees identifiability, and show that, under this condition, the maximum likelihood estimator for both gating and prompt parameters converges at the near-parametric rate $O(n^{-\frac{1}{2}})$, matching minimax lower bounds. Conversely, when distinguishability fails—so that the prompt overlaps with the pre-trained expert—they prove that estimation rates slow to $O(n^{-\frac{1}{4}})$ or $O(n^{-\frac{3}{8}})$, depending on how the prompt parameters approach those of the pre-trained model, again with minimax-optimal lower bounds. This work clarifies how overlap between experts degrades statistical efficiency and lays the groundwork for extensions to multiple prompts and broader model families.

**Questions:**

1. What is the most interesting insight compared to [2], [3]?
2. Can give more empirical results demonstrate the potential of Contaminated MoE that performs better than fully fine-tuning?



**References**

[1] Zadouri, T., Ust¨un, A., Ahmadian, A., Ermis, B., Lo-catelli, A., and Hooker, S. (2024). Pushing mixture of experts to the limit: Extremely parameter efficient moe for instruction tuning. In The Twelfth International Conference on Learning Representations.

[2] Nguyen, Huy, Nhat Ho, and Alessandro Rinaldo. "Sigmoid gating is more sample efficient than softmax gating in mixture of experts." arXiv preprint arXiv:2405.13997 (2024).

[3] Nguyen, Huy, TrungTin Nguyen, and Nhat Ho. "Demystifying softmax gating function in Gaussian mixture of experts." Advances in Neural Information Processing Systems 2023.

[4] Yan, Fanqi, et al. "Understanding expert structures on minimax parameter estimation in contaminated mixture of experts." arXiv preprint arXiv:2410.12258 (2024).

**Ethical Concerns:**

["NO or VERY MINOR ethics concerns only"]

**Final Justification:**

After carefully reviewing the authors' detailed responses to the concerns raised, I am satisfied with the clarifications provided, which address the key points of confusion.

The authors have clearly distinguished "Contaminated MoE" from related parameter-efficient tuning methods like LoRA and EP-MoE [1], emphasizing its unique structure as an input-dependent weighted sum of a frozen pre-trained model and a trainable prompt model. This distinction helps contextualize the work within the broader landscape of efficient fine-tuning, resolving the initial ambiguity about the scope of their framework.

Regarding the comparison to prior work [2], [3], and [4], the authors have effectively highlighted the novelty of their contributions: the focus on a practically motivated contaminated setting (frozen + trainable experts) absent in standard MoE analyses [2], [3]; the consideration of sample-size-dependent ground-truth parameters leading to uniform convergence rates (a more realistic scenario than fixed parameters); and the analysis of input-dependent softmax gating (vs. input-free gating in [4]), with insights into how gating choice impacts estimation efficiency. These points clarify the paper’s theoretical insights and fill a gap in the literature on Gaussian MoE convergence under parameter-efficient fine-tuning regimes.

However, as I am not the expert on this field. I am still concern it's contribution impact on the problem setting. Therefore I maintain my original rating.

**Limitations:**

yes

**Quality:**

4

**Strengths And Weaknesses:**

## Strengths

+ This paper proposes "distinguishability", which means pre-trained model $f_0$ and prompt model $f$ (and the way $f$ shifts when you tweak its parameters) span a sufficiently rich family so they never lie on the same low-dimensional subspace. When distinguishability holds, the softmax gating can cleanly attribute each sample to one expert or the other, yielding near-parametric $O(n^{-\frac{1}{2}})$ estimation. If they did lie on the same subspace, you could hide a change in gating or prompt parameters by making the other expert “absorb” the effect—leading to non-identifiable models and slower estimation rates. This result is insightful and and align with intuition.

+ It delivers the first rigorous minimax-optimal convergence rates for both gating (softmax) and prompt parameters under softmax-contaminated MoE settings—covering both distinguishable and non-distinguishable regimes—thus filling the convergence analysis gap of Gaussian MoE.


## Weaknesses
- Assume that "Contaminated MoE" only refers to a specific setting where frozen experts and a "conditioned" prompt model, excluding efficient tuning such as LoRA, or EP-MoE [1] (Please correct me if I am wrong). Contaminated MoE seems to be a rare setting in application such that few evidences indicate this pipeline has better performance than general supervised fine-tuning. In addition, mean expert function $f_0$ weakens the mutual effect than discussing multiple experts.

- Similar insights seem to be found in [2], [3]. Under the Gaussian distribution assumption, I think the insight is somehow lack of the novelty. Please state the differences (mostly on insights) compared to the similar work. For example, [4] also proved $O(n^{-\frac{1}{2}})$ convergence under linear and non-linear setting. The novelty of the results seems unclear.

---

> ### Author Rebuttal · Authors · 2025-07-30
>
> Dear Reviewer 2SMz,
>
> Thanks for your insightful reviews, and giving **excellent (4) grade** to the quality and **good (3) grade** to the clarity and significance of our paper. We are encouraged by the endorsement that
>
> *(i) Our result is **insightful and and align with intuition**;*
>
> *(ii) Our paper delivers **the first** rigorous minimax-optimal convergence rates for both gating (softmax) and prompt parameters under softmax-contaminated MoE;*
>
> (iii) *filling the convergence analysis gap of Gaussian MoE.*
>
> Thank you for your thoughtful review, which have helped us improved our paper substantially. Below are our responses to your questions. We will also add these changes to the revision of our manuscript. We hope that we have addressed your concerns regarding our paper.
>
> ***
>
> **Q1: Assume that "Contaminated MoE" only refers to a specific setting where frozen experts and a "conditioned" prompt model, excluding efficient tuning such as LoRA, or EP-MoE [1]**
>
> Thanks for your question. We confirm that the concept of contaminated MoE is different from that of LoRA and parameter-efficient MoE in [1]. In particular,
>
> - Contaminated MoE is an input-dependent weighted sum of a frozen pre-trained model and a trainable prompt model without rank control;
>
> - LoRA is a normal sum of a frozen pre-trained weight and a product of two learnable low-rank matrices;
>
> - Parameter-efficient MoE is also an input-dependent gating MoE but all the experts are trainable formulated as adapters.
>
> We will add a remark to ensure that this point is clear.
>
> ***
>
> **Q2: Contaminated MoE seems to be a rare setting in application such that few evidences indicate this pipeline has better performance than general supervised fine-tuning. Can you give more empirical results to demonstrate the potential of Contaminated MoE that performs better than fully fine-tuning?**
>
> Thanks for your question. Let us clarify that we do not aim to compare the performance of contaminated MoE and full fine-tuning method in this paper. Instead, we focus on analyzing the prompt convergence in the contaminated MoE, a model inspired from a parameter-efficient fine-tuning method known as prefix-tuning, where a frozen pre-trained model is combined with a trainable prompt model to learn downstream tasks. We agree with the reviewer that the comparison between these two fine-tuning methods is worth exploring. However, since this direction lies beyond the scope of our work, we leave it for future development. We will clarify this important point.
>
> ***
>
> **Q3: The novelty of our paper compared to previous works [2, 3, 4].**
>
> Thank you for your comment. We would like to clarify the key differences in terms of insights and contributions between our work and previous research, particularly [2], [3], and [4]:
>
> **Compared with [2, 3]:**
> There are four main differences between [2, 3] and our work.
>
> *1. Concepts:* In our work, we consider contaminated MoE, which is a mixture of a frozen pre-trained model and a trainable prompt model motivated from parameter-efficient fine-tuning method, while models considered in [2, 3] are a standard mixture of all trainable sub-models, used for scaling up the model capacity in large-scale AI models.
>
> *2. More practical settings:* We allow ground-truth parameters $G_*$ to change with the sample size $n$, which is closer to practical settings than the assumptions in [2, 3] where $G_*$ does not change with $n$.
>
> *3. Uniform convergence rates:* Since ground-truth parameters vary with the sample size, the convergence rates of parameter estimations in our work are uniform rather than point-wise as in [2, 3]. Furthermore, these rates are able to capture the interaction between the convergences of parameter estimations, while those in [2, 3] cannot.
>
> *4. Minimax lower bounds:* Finally, we determine minimax lower bounds under both distinguishable and non-distinguishable settings. Based on these lower bounds, we can claim that our derived convergence rates are optimal. However, no minimax lower bounds are provided in [2, 3].
>
> **Compared with [4]:**
> There are two main differences between [4] and our work.
>
> *1. More realistic and challenging setting:* Our work considers contaminated MoE with softmax gating (input dependent), whereas the gating function in [4] is input-free. This input-dependent formulation is not only more realistic and applicable in practice—capturing the natural dependence of the gating mechanism on the input—but also presents additional theoretical challenges. As a result, our analysis is both more rigorous and better aligned with practical modeling scenarios.
>
> *2. Insights:* From Section 3, we observe that when using the softmax gating, the convergence rates of prompt parameter estimation are faster than those when using the input-free gating in [4] (see, e.g., lines 175-183). Therefore, our theories encourage the use of softmax gating over input-free gating when tuning contaminated-MoE-based models.
>
> From Section 3.2, we observe that when the prompt model acquires overlapping knowledge with the pre-trained model (the distinguishability condition is violated), the convergence rates of prompt parameter estimation are slow down. Thus, our theories advocate using prompt models with different expertise from the pre-trained model. Meanwhile, Yan et al. [4] focused more on the linearity and non-linearity of expert functions.
>
> ***
>
> **References**
>
> [1] Zadouri, T., Ust¨un, A., Ahmadian, A., Ermis, B., Lo-catelli, A., and Hooker, S. (2024). Pushing mixture of experts to the limit: Extremely parameter efficient moe for instruction tuning. In The Twelfth International Conference on Learning Representations.
>
> [2] Nguyen, Huy, Nhat Ho, and Alessandro Rinaldo. "Sigmoid gating is more sample efficient than softmax gating in mixture of experts." arXiv preprint arXiv:2405.13997 (2024).
>
> [3] Nguyen, Huy, TrungTin Nguyen, and Nhat Ho. "Demystifying softmax gating function in Gaussian mixture of experts." Advances in Neural Information Processing Systems 2023.
>
> [4] Yan, Fanqi, et al. "Understanding expert structures on minimax parameter estimation in contaminated mixture of experts." arXiv preprint arXiv:2410.12258 (2024).

---

> > ### Comment · Reviewer_qXy5 · 2025-08-05
> > **Submission16667**
> >
> > Thanks to the authors for the rebuttal and detailed discussions. Some of my questions are addressed. However, the following points need clarification.
> >
> > 1- The evaluation on CIFAR-10, despite the inclusion of few-shot fine-tuning, is insufficient. CIFAR-10 is a small, well-studied dataset with limited complexity, lacking the diversity and scale. The reliance on synthetic domains and modest sample sizes fails to demonstrate robust generalization. From my perspective, a comprehensive evaluation on larger, more diverse datasets is essential to substantiate the claims.
> >
> > 2- Can you clarify how the non-differentiable points of ReLU impact the convergence rates of the MLE in practice? It would be helpful to see empirical evidence or a more detailed explanation to confirm that ReLU reliably satisfies the condition without affecting the theoretical assumptions.

---

> ### Author Response · Authors · 2025-08-06
> **Response to Reviewer qXy5**
>
> Dear Reviewer qXy5,
>
> Thanks for your response. We are glad to hear that we have addressed some of your questions. Next, let us clarify the rest as follows.
>
> *1. Regarding the CIFAR-10 dataset:* we agree with the reviewer that the CIFAR-10 dataset is small and has limited complexity. However, we would like to recall the theoretical nature of our paper. In particular, our primary contribution is to provide a comprehensive convergence analysis of parameter estimation in the softmax-contaminated MoE model. To empirically justify our theoretical findings, we perform several numerical experiments in the paper. Then, per the reviewers' request to provide empirical results on a real-world dataset, we conduct additional experiments on the CIFAR-10 dataset, which fits our computation budget and the restricted time of the discussion period. We believe that those experiments on both synthetic and real datasets are sufficient to validate our theoretical findings. On the other hand, a comprehensive evaluation on larger and more diverse datasets lies beyond the scope of our paper. Therefore, we hope that the reviewer will evaluate the paper based on its main contribution.
>
> *2. Regarding the impact of the non-differentiable point of ReLU function on the MLE convergence rates:* we assume that the reviewer was talking about the strong identifiability condition. This condition only requires the expert function $h(x,\eta)$ to be twice differentiable with respect to $\eta$ for almost every $x$. Therefore, in general, the non-differentiable point of $ReLU$ would not affect the strong identifiability of an expert function. For example, it can be checked that the expert function $h(x,\eta)=ReLU(x^{\top}\eta\eta^{\top}x)$ is strongly identifiable. As a result, the convergence rates of MLE presented in Theorem 3 are not affected. The numerical experiments provided in Section 4 can totally be reproduced using this choice of expert function. However, due to the NeurIPS rebuttal policy, we cannot show the plots illustrating the convergence rates of parameter estimation (as in Figure 2) here. We hope that the reviewer can sympathize with this issue.
>
> We hope that our response successfully addresses your concerns. If the reviewer has further concerns, please feel free to let us know. We are happy to address any additional concerns from you.
>
> Best regards,
>
> The Authors

---

### Official Review · Reviewer_qXy5 · 2025-07-05

**Clarity:** 2
**Significance:** 2
**Originality:** 2
**Rating:** 3
**Confidence:** 3

**Summary:**

The paper analyzes the softmax-contaminated MOE, in which a fixed pre-trained model and a trainable prompt are combined for fine-tuning. The paper investigates the maximum likelihood estimator convergence for gating and prompt parameters, introducing a distinguishability condition to measure model differences.

**Questions:**

1) I did not understand why in page 6 the identifiability condition requires twice-differentiable expert functions. How does it apply to non-smooth functions like ReLU?

2) What is the network design for the experiments and the proposed model.

3) In Fig. 1, what are the data generation process and sample sizes? It is better to show the performances on a public dataset as well.

**Ethical Concerns:**

["NO or VERY MINOR ethics concerns only"]

**Final Justification:**

After reading the authors rebuttal and discussions, some of my comments are addressed. While the authors provided some additional experiments on CIFAR10, still my main concern regarding the experiments on the complex dataset is not addressed. By the way, I raised my rate.

**Limitations:**

In the main paper, limitations are not discussed.

**Quality:**

2

**Strengths And Weaknesses:**

Strengths:

The paper provides mathematical proofs and performs evaluations on two case studies.

:Weaknesses:

1- lack of experimental analysis. While mathematical proof was provided, the paper was only evaluated on a synthetic dataset.

2- The presentation of the paper needs improvement. For example, on page 2, the parameters in the table are not defined.

3- in theorem 2 some parameters are not defined, such as $\langle \langle \Delta \eta, \Delta \nu \rangle \rangle$

---

> ### Author Rebuttal · Authors · 2025-07-30
>
> Dear Reviewer qXy5,
>
> Thanks for your constructive and valuable feedback. Below are our responses to your questions. We will also add these changes to the revision of our manuscript. **We hope that we have addressed your concerns regarding our paper and will ultimately convince you to raise your score.**
>
> ***
>
> **Q1: Lack of experimental analysis, with sole emphasis on synthetic data.**
>
> Thanks for your suggestion. We have added a few-shot fine-tuning experiment on DINOv2 ViT-S embeddings of CIFAR-10 under a domain shift from vehicles to animals. In this setting, the softmax contaminated MoE consistently outperforms both the input-free contaminated MoE and standard fine-tuning. Please see the additional experiments for details.
>
> **Additional Experiments**
> We study adaptation of a pretrained classifier to a distributionally shifted dataset.
> Let $h_{0}$ be a model trained on a “source” subset biased toward *vehicles*.
> We then adapt the model on a small “target” subset biased toward *animals* and evaluate on the full, balanced CIFAR-10 test split.
>
> ---
>
> ### Setup
> We extract 384-dimensional embeddings from CIFAR-10 images using a DINOv2 “small” ViT.
> The ten classes are partitioned into two groups:
> * **Animals** — bird, cat, deer, dog, frog, horse (6 classes)
> * **Vehicles** — airplane, automobile, ship, truck (4 classes)
>
> ---
>
> ### Data construction (per-class counts)
>
> * **Distinguishable setting**: the prompt model does not share knowledge with the pre-trained model
>   * Pre-train: 524 samples exclusively from vehicle classes (131 per vehicle class)
>   * Fine-tune: 56 samples exclusively from animal classes (≈ 9–10 per animal class)
>
> * **Non-distinguishable setting**: the prompt model partly shares knowledge with the pre-trained model
>   * Pre-train: 512 vehicle (128 per class) + 12 animal (2 per class) = 524
>   * Fine-tune: 48 animal (8 per class) + 8 vehicle (2 per class) = 56
>
> The held-out test set is the full, balanced 10 000-image CIFAR-10 split.
>
> ---
>
> ### Implementation details
>
> | Setting      | Split       | Animals / class | Vehicles / class | Total |
> |-------------|-------------|-----------------|------------------|-------|
> | Distinguishable  | Pre-train   | 0               | 131              | 524   |
> |             | Fine-tune   | 9–10            | 0                | 56    |
> | Non-distinguishable | Pre-train   | 2               | 128              | 524   |
> |             | Fine-tune   | 8               | 2                | 56    |
>
> ---
>
> ### Training
>
> * **Pre-training** — train a linear classifier $h_{0}$ on the pre-training set with cross-entropy, Adam (lr $10^{-3}$, wd $10^{-4}$, batch 32) for 20 epochs, then freeze its parameters.
> * **Fine-tuning** — attach a contaminated MoE that mixes $h_{0}$ with a new trainable linear expert $h_{1}$ with gating parameters $(\beta,\tau)$
>
>   $$
>     \hat{y}(x)=\bigl(1-\sigma(\beta^{\top}x+\tau)\bigr)\,h_{0}(x)
>                +\sigma(\beta^{\top}x+\tau)\,h_{1}(x).
>   $$
>
>   Only the prompt and gating parameters are updated (10 epochs, same optimizer).
>
> *Optimizer:* Adam (lr $1\times10^{-3}$, wd $1\times10^{-4}$, batch 32)
> *Epochs:* 20 (pre-train), 10 (fine-tune)
>
> ---
>
> ### Evaluation and baselines
>
> We report accuracy on the full test set, as well as on the animal-only and vehicle-only subsets.
> Baselines: Single frozen pre-trained model, Fine-tuning pre-trained model (without prompt), Input-free contaminated MoE [2], Softmax-contaminated MoE (Ours).
>
> ---
>
> ### Results
>
> *Columns in each table are accuracies on the indicated test subset.*
>
> ### Table 1a – *Distinguishable* setting
>
> | Model | Animals | Vehicles | Overall |
> |-------|:-------:|:--------:|:-------:|
> | Single frozen pre-trained model             | 0.1 | **94.9** | 38.0 |
> |  Fine-tuning pre-trained model  | 69.4 | 91.1     | 78.1 |
> | Input-free contaminated MoE      | 52.6 | 87.9     | 69.1 |
> | **Softmax-contaminated MoE** | **81.7** | 94.1 | **86.7** |
>
> ### Table 1b – *Non-distinguishable* setting
>
> | Model | Animals | Vehicles | Overall |
> |-------|:-------:|:--------:|:-------:|
> | Single frozen pre-trained model             | 20.7 | **95.4** | 50.5 |
> |  Fine-tuning pre-trained model   | 71.7 | 91.5     | 79.6 |
> | Input-free contaminated MoE      | 50.3 | 87.4     | 68.3 |
> | **Softmax-contaminated MoE** | **72.3** | 94.7 | **82.4** |
>
> ### Conclusion
>
> The above experiments empirically justify our two main theoretical findings:
>
> 1. **Softmax-contaminated MoE outperforms input-free contaminated MoE [2]:** Across both distinguishable and non-distinguishable settings, we observe that the Softmax-contaminated MoE achieves the highest and most balanced performance, improving animal accuracy while maintaining high vehicle accuracy. On the other hand, input-free contaminated MoE underperforms, and fine-tuning pre-trained model sits in between.
>
> 2. **Performance of softmax-contaminated MoE under distinguishable settings is higher than that under non-distinguishable setting:** It can be seen that the performance of softmax-contaminated MoE under the distinguishable setting presented in Table 1a is generally higher than that under the non-distinguishable setting presented in Table 1b.
>
> ***
> **Q2: The parameters in Table 1 and Theorem 2 are not defined.**
>
> Thanks for your feedback. We have defined all the parameters in Table 1 in Eq. (1), Eq. (2), and line 241, respectively, of our initial submission. However, following your suggestion, we will add the definitions of these parameters to the caption of Table 1.
>
> As for Theorem 2, we assume that the reviewer is referring to the notation $\bar{\beta}_n$, $\bar{\tau}_n$, $\bar{\eta}_n$, $\bar{\nu}_n$. These parameters are the components of $\bar{G}$ given in the infimum operator, that is, $\bar{G}:=(\bar{\beta}_n,\bar{\tau}_n,\bar{\eta}_n,\bar{\nu}_n)$. We will clarify this in the revised manuscript.
>
> ***
> **Q3: Why does the strong identifiability condition requires twice-differentiable expert functions? How does it apply to non-smooth functions like ReLU?**
>
> Thanks for your questions. It should be noted that a key step in obtaining MLE convergence rates is to decompose the density discrepancy $p_{\widehat{G_n}}-p_{G_*}$ into a combination of linearly independent terms through an appropriate Taylor expansion. This process involves high-order derivatives of the expert function $h(\cdot,\eta)$ w.r.t $\eta$, which may not be algebraically independent. Therefore, we introduce the strong identifiability condition, which involves the second-order derivatives of the expert function, to ensure the linear independence of terms in the Taylor expansion.
>
> Next, we clarify that the strong identifiability condition requires expert functions which are twice differentiable almost everywhere. Therefore, non-smooth functions like ReLU still can satisfy this condition as long as they meet the conditions of linear independence in lines 216-218.
>
> ***
> **Q4: What is the network design for the experiments and the proposed model?**
>
> Thanks for your questions. All architectural details are given in Section 4 (Ln 269–278). The frozen expert is $h_{0}(x)=\tanh(\eta_{0}^{\top}x)$, and the prompt expert is $h(x)=\tanh(\eta^{\top}x)$ - both a linear projection followed by $\tanh$. These two experts are mixed by the softmax–contaminated MoE gate in Eq. (1). We emphasize that we are not proposing a new network architecture; instead, we study the parameter estimation rates of the contaminated softmax MoE. Accordingly, we generate data with exactly this model (with additive label noise) and fit the same class via maximum likelihood, ensuring that the experiments directly validate the convergence rates predicted by our theory.
>
> ***
> **Q5: In Fig. 1, what are the data generation process and sample sizes? It is better to show the performances on a public dataset as well.**
>
> Thanks for your questions. The synthetic data in Fig. 1 are generated as described in Section 4 (Ln 279–291), we sample inputs $x\sim\mathcal N(0,I_d)$, draw labels from the softmax contaminated MoE of Eq. (1) using the ground-truth parameters specified there, and then fit the same model class by maximum likelihood.
> In the plot, the x-axis is the training sample size $n$ (log scale), while the y-axis shows the Euclidean distance (log scale) between each learned parameter and its true value; therefore, the slopes illustrate the empirical convergence rates predicted by our theory.
> To complement these controlled simulations, we have added a few shot fine-tuning experiments on real data, DINOv2 ViT-S embeddings of CIFAR-10 under a vehicles $\rightarrow$ animals domain shift, where the softmax contaminated MoE adapter again outperforms static and fine-tuning baselines. Please see the additional experiments for more details.
>
> ***
> **Q6: In the main paper, limitations are not discussed.**
>
> Thanks for your comment. We have implicitly discussed the limitations of our work in the last paragraph of Section 5. Specifically, we mention that our analysis is restricted to the case of a single prompt and focuses solely on the scenario where prompt models belong to the family of Gaussian distributions. We will further elaborate on these points. In the meanwhile, please let us know if you have further or specific concerns.
>
> ***
> **References**
>
> [2] Yan et al. "Understanding expert structures on minimax parameter estimation in contaminated mixture of experts." In AISTATS, 2025.

---

> > ### Author Response · Authors · 2025-08-05
> > **Looking forward to your response**
> >
> > Dear Reviewer qXy5,
> >
> > We would like to express our gratitude for your valuable review, and we hope that our response addresses your previous concerns regarding our paper. However, as the discussion period is expected to end in the next few days, please feel free to let us know if you have any further comments on our work. We would be willing to address any additional concerns from you. Otherwise, we hope that the reviewer will consider increasing the rating of our paper accordingly.
> >
> > Thank you again for spending time and effort on the paper, we really appreciate it!
> >
> > Best regards,
> >
> > Authors

---

> ### Author Response · Authors · 2025-08-08
> **Response to Reviewer qXy5**
>
> Dear Reviewer  qXy5,
>
> We apologize for the oversight — since your comments appeared in the discussion thread of Reviewer 2SMz, we mistakenly responded there and did not realize you were awaiting a direct reply. We have, in fact, already addressed your questions in that earlier message, but for clarity and completeness, we restate our response to you below.
>
> Thanks for your response. We are glad to hear that we have addressed some of your questions. Next, let us clarify the rest as follows.
>
> *1. Regarding the CIFAR-10 dataset:* we agree with the reviewer that the CIFAR-10 dataset is small and has limited complexity. However, we would like to recall the theoretical nature of our paper. In particular, our primary contribution is to provide a comprehensive convergence analysis of parameter estimation in the softmax-contaminated MoE model. To empirically justify our theoretical findings, we perform several numerical experiments in the paper. Then, per the reviewers' request to provide empirical results on a real-world dataset, we conduct additional experiments on the CIFAR-10 dataset, which fits our computation budget and the restricted time of the discussion period. We believe that those experiments on both synthetic and real datasets are sufficient to validate our theoretical findings. On the other hand, a comprehensive evaluation on larger and more diverse datasets lies beyond the scope of our paper. Therefore, we hope that the reviewer will evaluate the paper based on its main contribution.
>
> *2. Regarding the impact of the non-differentiable point of ReLU function on the MLE convergence rates:* we assume that the reviewer was talking about the strong identifiability condition. This condition only requires the expert function $h(x,\eta)$ to be twice differentiable with respect to $\eta$ for almost every $x$. Therefore, in general, the non-differentiable point of $ReLU$ would not affect the strong identifiability of an expert function. For example, it can be checked that the expert function $h(x,\eta)=ReLU(x^{\top}\eta\eta^{\top}x)$ is strongly identifiable. As a result, the convergence rates of MLE presented in Theorem 3 are not affected. The numerical experiments provided in Section 4 can totally be reproduced using this choice of expert function. However, due to the NeurIPS rebuttal policy, we cannot show the plots illustrating the convergence rates of parameter estimation (as in Figure 2) here. We hope that the reviewer can sympathize with this issue.
>
> We hope that our response successfully addresses your concerns. If the reviewer has further concerns, please feel free to let us know. We are happy to address any additional concerns from you. Otherwise, we will really appreciate if the reviewer can consider increase the rating of our submission.
>
> Best regards,
>
> The Authors

---

### Official Review · Reviewer_mgR9 · 2025-07-05

**Clarity:** 3
**Significance:** 3
**Originality:** 3
**Rating:** 4
**Confidence:** 3

**Summary:**

This paper theoretically analyses the softmax-contaminated mixture of experts model in terms of a distinguishability condition, minimax-optimal convergence rates, looking at the fact that estimation rates degrade, and also provide empirical validation with numerics. Importantly it addresses gaps in the MoE literature which historically has been quite empirical/application driven. There are few theoretical works in MoE, so it is a good contribution to close that gap.

**Questions:**

Can you add a discussion section discussing how this theory can inform practical model tuning decisions for users?

Can the analysis be extended to hard-routing? Similar to: Approximation Rates and VC-Dimension Bounds for (P)ReLU
MLP Mixture of Experts, TMLR

**Ethical Concerns:**

["NO or VERY MINOR ethics concerns only"]

**Final Justification:**

I share similar concerns to reviewer 2SMz regarding the practical experiments. Due to this I will keep my original score.

**Limitations:**

No real world experiments. Could you add some more realistic experiments for me to raise my score? Could you apply your experiments to a  more realistic prompt-tuning task.

Even a simple simulation involving real embeddings (e.g., from BERT, GPT, or CLIP) with synthetic labels similar in spirit to the DINOv2-based experiments in 'Approximation Rates and VC-Dimension Bounds for (P)ReLU
MLP Mixture of Experts, TMLR'  would really help. (could be a natural minimal-realism extension of their synthetic setup)

**Quality:**

3

**Strengths And Weaknesses:**

Strength:
- Addresses a novel problem
- they  derive both upper and matching minimax lower bounds in both distinguishable and non-distinguishable regimes
- motivated by modern fine-tuning practices like prefix tuning (although the experiments themselves are not really practical enough as later discussed)
- good paper writing and structure

Weaknesses:

- All experiments are synthetic and low-dimensional
- prompts are gaussian, doesnt cover other cases
- there are no practical benchmarks, real NLP/LLM tasks

---

> ### Author Rebuttal · Authors · 2025-07-30
>
> Dear Reviewer mgR9,
>
> Thanks for your constructive and valuable feedback, and giving **good (3) grade** to quality, clarity, significance and originality of our paper. We are encouraged by the endorsement: (i) the problem we address is novel and (ii) the paper has good writing and structure.
>
> Below are our responses to your questions. We will also add these changes to the revision of our manuscript. We hope that we address your concerns regarding our paper, and eventually convince you to raise your score.
>
> ***
>
> **Q1: Can you add a discussion section discussing how this theory can inform practical model tuning decisions for users?**
>
> Thank you for your suggestion. There are two important practical implications for tuning models from our theories.
>
> 1. From Section 3, we observe that when using the softmax gating, the convergence rates of prompt parameter estimation are faster than those when using the input-free gating in [2]. Therefore, our theories encourage the use of softmax gating over input-free gating when tuning contaminated-MoE-based models.
>
> 2. From Section 3.2, we observe that when the prompt model acquires overlapping knowledge with the pre-trained model (the distinguishability condition is violated), the convergence rates of prompt parameter estimation are slow down. Thus, our theories advocate using prompt models with different expertise from the pre-trained model.
>
> ***
> **Q2: Can the analysis be extended to hard-routing? Similar to [1]**
>
> Thanks for your question. We believe that our analysis can be extended to the setting where only one expert is activated per input as in [1]. For that purpose, we need to employ techniques for analyzing sparsely gated MoE in [3]. In particular, we first partition the input space into two regions corresponding to two experts in the contaminated MoE:
>
> (i) Region 1 contains $x\in\mathcal{X}$ such that $\exp(\beta^{\top}x+\tau)>1$, that is, the weight of the prompt model is larger than that of the pre-trained model;
>
> (ii) Region 2 contains $x\in\mathcal{X}$ such that $\exp(\beta^{\top}x+\tau)\leq 1$, that is, the weight of prompt model is smaller than that of the pre-trained model.
>
> Then, the conditional density function associated with the contaminated MoE is given by
>
> - $p_{G}(y|x)=f(y|h(x,\eta),\nu)$, if the input $x$ belongs to Region 1;
>
> - $p_{G}(y|x)=f_0(y|h_0(x,\eta_0),\nu_0)$, if the input $x$ belongs to Region 2.
>
> Then, we can reuse our proof techniques to establish the convergence rates of prompt and gating parameter estimation. However, since this direction lies beyond the scope of our work, we leave it for future development, though we will  mention it in the main text as an interesting extension.
>
> ***
> **Q3: All experiments are synthetic and low-dimensional without practical benchmarks and real NLP/LLM tasks. Could you apply your experiments to a more realistic prompt-tuning task?**
>
> Thanks for your suggestion. We have added a few-shot fine-tuning experiment on DINOv2 ViT-S embeddings of CIFAR-10 under a domain shift from vehicles to animals. In this setting, the softmax contaminated MoE consistently outperforms both the input-free contaminated MoE and standard fine-tuning. Please see the additional experiments for details.
>
> **Additional Experiments**
> We study adaptation of a pretrained classifier to a distributionally shifted dataset.
> Let $h_{0}$ be a model trained on a “source” subset biased toward *vehicles*.
> We then adapt the model on a small “target” subset biased toward *animals* and evaluate on the full, balanced CIFAR-10 test split.
>
> ---
>
> ### Setup
> We extract 384-dimensional embeddings from CIFAR-10 images using a DINOv2 “small” ViT.
> The ten classes are partitioned into two groups:
> * **Animals** — bird, cat, deer, dog, frog, horse (6 classes)
> * **Vehicles** — airplane, automobile, ship, truck (4 classes)
>
> ---
>
> ### Data construction (per-class counts)
>
> * **Distinguishable setting**: the prompt model does not share knowledge with the pre-trained model
>   * Pre-train: 524 samples exclusively from vehicle classes (131 per vehicle class)
>   * Fine-tune: 56 samples exclusively from animal classes (≈ 9–10 per animal class)
>
> * **Non-distinguishable setting**: the prompt model partly shares knowledge with the pre-trained model
>   * Pre-train: 512 vehicle (128 per class) + 12 animal (2 per class) = 524
>   * Fine-tune: 48 animal (8 per class) + 8 vehicle (2 per class) = 56
>
> The held-out test set is the full, balanced 10 000-image CIFAR-10 split.
>
> ---
>
> ### Implementation details
>
> | Setting      | Split       | Animals / class | Vehicles / class | Total |
> |-------------|-------------|-----------------|------------------|-------|
> | Distinguishable  | Pre-train   | 0               | 131              | 524   |
> |             | Fine-tune   | 9–10            | 0                | 56    |
> | Non-distinguishable | Pre-train   | 2               | 128              | 524   |
> |             | Fine-tune   | 8               | 2                | 56    |
>
> ---
>
> ### Training
>
> * **Pre-training** — train a linear classifier $h_{0}$ on the pre-training set with cross-entropy, Adam (lr $10^{-3}$, wd $10^{-4}$, batch 32) for 20 epochs, then freeze its parameters.
> * **Fine-tuning** — attach a contaminated MoE that mixes $h_{0}$ with a new trainable linear expert $h_{1}$ with gating parameters $(\beta,\tau)$
>
>   $$
>     \hat{y}(x)=\bigl(1-\sigma(\beta^{\top}x+\tau)\bigr)\,h_{0}(x)
>                +\sigma(\beta^{\top}x+\tau)\,h_{1}(x).
>   $$
>
>   Only the prompt and gating parameters are updated (10 epochs, same optimizer).
>
> *Optimizer:* Adam (lr $1\times10^{-3}$, wd $1\times10^{-4}$, batch 32)
> *Epochs:* 20 (pre-train), 10 (fine-tune)
>
> ---
>
> ### Evaluation and baselines
>
> We report accuracy on the full test set, as well as on the animal-only and vehicle-only subsets.
> Baselines: Single frozen pre-trained model, Fine-tuning pre-trained model (without prompt), Input-free contaminated MoE [2], Softmax-contaminated MoE (Ours).
>
> ---
>
> ### Results
>
> *Columns in each table are accuracies on the indicated test subset.*
>
> ### Table 1a – *Distinguishable* setting
>
> | Model | Animals | Vehicles | Overall |
> |-------|:-------:|:--------:|:-------:|
> | Single frozen pre-trained model             | 0.1 | **94.9** | 38.0 |
> |  Fine-tuning pre-trained model  | 69.4 | 91.1     | 78.1 |
> | Input-free contaminated MoE      | 52.6 | 87.9     | 69.1 |
> | **Softmax-contaminated MoE** | **81.7** | 94.1 | **86.7** |
>
> ### Table 1b – *Non-distinguishable* setting
>
> | Model | Animals | Vehicles | Overall |
> |-------|:-------:|:--------:|:-------:|
> | Single frozen pre-trained model             | 20.7 | **95.4** | 50.5 |
> |  Fine-tuning pre-trained model   | 71.7 | 91.5     | 79.6 |
> | Input-free contaminated MoE      | 50.3 | 87.4     | 68.3 |
> | **Softmax-contaminated MoE** | **72.3** | 94.7 | **82.4** |
>
> ***
>
> ### Conclusion
>
> The above experiments empirically justify our two main theoretical findings:
>
> 1. **Softmax-contaminated MoE outperforms input-free contaminated MoE [2]:** Across both distinguishable and non-distinguishable settings, we observe that the Softmax-contaminated MoE achieves the highest and most balanced performance, improving animal accuracy while maintaining high vehicle accuracy. On the other hand, input-free contaminated MoE underperforms, and fine-tuning pre-trained model sits in between.
>
> 2. **Performance of softmax-contaminated MoE under distinguishable settings is higher than that under non-distinguishable setting:** It can be seen that the performance of softmax-contaminated MoE under the distinguishable setting presented in Table 1a is generally higher than that under the non-distinguishable setting presented in Table 1b.
>
> ***
> **References**
>
> [1]  Kratsios et al. Approximation Rates and VC-Dimension Bounds for (P)ReLU MLP Mixture of Experts, TMLR.
>
> [2] Yan et al. "Understanding expert structures on minimax parameter estimation in contaminated mixture of experts." In AISTATS, 2025.
>
> [3] Nguyen et al. Statistical perspective of top-k sparse softmax gating mixture of experts. In ICLR, 2024.

---

### Decision · Program_Chairs · 2025-09-17

**Decision:**

Accept (poster)

**Comment:**

They provide the first theoretical analysis of a technique for fine-tuning, where a pre-trained expert model and a trainable Gaussian expert ("prompt" expert) are combined via a SoftMax linear gating network.
They consider the maximum likelihood estimator (MLE) for gating and prompt parameters.  They find multiple regimes of convergence rates, depending whether various conditions hold (e.g., a "distinguishability" condition), and establish minimax rates (achieved by MLE) in each regime (the strongest of which is a $1/\sqrt{n}$ rate of convergence).

The reviewers are nearly unanimous in favoring acceptance.  They are encouraged to see a theoretical treatment of this technique motivated by popular approaches to fine-tuning.  They praise the thoroughness of the analysis, which identifies multiple regimes of convergence depending on which conditions hold.

Initial reservations raised in the reviews appear to have been addressed in the rebuttals, which the reviewers seem to be satisfied with.

The one major critique noted by all reviewers is that the experimental evaluation is synthetic and more experiments may be called for.  The authors' rebuttals address this by providing additional experiments on real data.  Most of the reviewers seem content with these additional experiments.